# AUTOREGRESSIVE CONDITIONAL NEURAL PROCESSES

**Wessel P. Bruinsma**[*12], **Stratis Markou**[*2], **James Requiema**[*2], **Andrew Y. K. Foong**[*1],
**Tom R. Andersson**[3], **Anna Vaughan**[2], **Anthony Buonomo**[2], **J. Scott Hosking**[34],
**Richard E. Turner**[12]

[*]Equal contribution

[1]Microsoft Research AI4Science, [2]University of Cambridge,
[3]British Antarctic Survey, [4]The Alan Turing Institute

{wbruinsma,andrewfoong}@microsoft.com
{em626,jrr41,av555,ab2707,ret26}@cam.ac.uk
{tomand,jask}@bas.ac.uk

## ABSTRACT

Conditional neural processes (CNPs; Garnelo et al., 2018a) are attractive meta-learning models which produce well-calibrated predictions and are trainable via a simple maximum likelihood procedure. Although CNPs have many advantages, they are unable to model dependencies in their predictions. Various works propose solutions to this, but these come at the cost of either requiring approximations or being limited to Gaussian predictions. In this work, we instead propose to change how CNPs are deployed at test time, *without any modifications to the model or training procedure*. Instead of making predictions independently for every target point, we autoregressively define a joint predictive distribution using the chain rule of probability, taking inspiration from the neural autoregressive density estimator (NADE) literature. We show that this simple procedure allows factorised Gaussian CNPs to model highly dependent, non-Gaussian predictive distributions. Perhaps surprisingly, in an extensive range of tasks with synthetic and real data, we show that CNPs in autoregressive (AR) mode not only significantly outperform non-AR CNPs, but are also competitive with more sophisticated models that are significantly more expensive and challenging to train. This performance is remarkable since AR CNPs are not trained to model joint dependencies. Our work provides an example of how ideas from neural distribution estimation can benefit neural processes, motivating research into the AR deployment of other neural process models.

## 1 INTRODUCTION

Conditional neural processes (CNPs; Garnelo et al., 2018a) are a family of meta-learning models which combine the flexibility of deep learning with the uncertainty awareness of probabilistic models. They are trained to produce well-calibrated predictions via a simple maximum-likelihood procedure, and naturally handle off-the-grid and missing data, making them ideally suited for tasks in climate science and healthcare. Since their introduction, attentive (ACNP; Kim et al., 2019) and convolutional (ConvCNP; Gordon et al., 2020) variants have also been proposed. Unfortunately, existing CNPs do

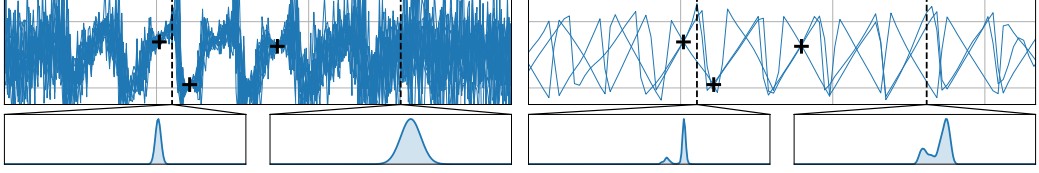

Figure 1: A ConvCNP trained on random sawtooth functions and applied in standard mode (*left*) and in our proposed autoregressive (AR) mode (*right*). The black crosses denote observed data points, the blue lines show model samples, and the bottom plots show the marginal predictive distributions at the locations marked by the dashed vertical lines. In standard mode, the CNP models each output with an independent Gaussian (*left*). However, when run in AR mode, the *same* CNP can produce coherent samples and model multimodality (*right*).

| Class | Consistent | Dependencies | Non-Gaussian | Exact Training |
|---|:---:|:---:|:---:|:---:|
| AR CNPs (this work) | ✗ | ✓ | ✓ | ✓ |
| CNPs (Garnelo et al., 2018a) | ✓ | ✗ | ✓ | ✓ |
| GNPs (Markou et al., 2022) | ✓ | ✓ | ✗ | ✓ |
| LNPs (Garnelo et al., 2018b) | ✓ | ✓ | ✓ | ✗ |

Table 1: Comparison of various classes of neural processes. Shows whether a class produces consistent predictions, models dependencies, can produce non-Gaussian predictions, and can be trained without approximating the objective function. For CNPs, even though the presentation by Garnelo et al. (2018a) assumes Gaussian predictions, it is simple to relax this Gaussianity assumption; this is not the case for GNPs.

not model statistical dependencies (Figure 1; left). This harms their predictive performance and makes it impossible to draw coherent function samples, which are necessary in downstream estimation tasks (Markou et al., 2022). Various approaches have been proposed to address this. Garnelo et al. (2018b) introduced the latent neural process (LNP), which uses a latent variable to induce dependencies and model non-Gaussianity. However, this renders the likelihood intractable, necessitating approximate inference. Another approach is the fully convolutional Gaussian neural process (FullConvGNP; Bruinsma et al., 2021), which maintains tractability at the cost of only allowing Gaussian predictions. It uses a neural network to define the mean and covariance function of a predictive Gaussian process (GP) that models dependencies. However, it uses a much more complex architecture and is only practically applicable to problems with one-dimensional inputs, limiting its adoption compared to the more lightweight CNP. Recently, Markou et al. (2022) proposed the Gaussian neural process (GNP), which is considerably simpler but sacrifices performance relative to the FullConvGNP.

In this paper we propose a much simpler method for modelling dependencies with neural processes that has been largely overlooked: autoregressive (AR) sampling. AR sampling requires *no changes* to the architecture *or* training procedure. Instead, we *change how the CNP is deployed at test time*, extracting predictive information that would ordinarily be ignored. Instead of making predictions at all target points simultaneously, we autoregressively feed samples back into the model. AR CNPs trade the fundamental property of *consistency under marginalisation and permutation*, which is foundational to many neural process models, for non-Gaussian and correlated predictions. In Table 1 we place AR CNPs within the framework of other neural process models. Our key contributions are:

- We show that CNPs used in AR mode capture rich, non-Gaussian predictive distributions and produce coherent samples (Figure 1). This is remarkable, since these CNPs have Gaussian likelihoods, are not trained to model joint dependencies or non-Gaussianity, and are significantly cheaper to train than LNPs and FullConvGNPs (Figure 2).
- We prove that, given sufficient data and model capacity, the performance of AR CNPs is at least as good as that of GNPs, which explicitly model correlations in their predictions.
- Viewing AR CNPs as a type of neural density estimator (Uria et al., 2016), we highlight their connections to a range of existing methods in the deep generative modelling literature.
- In an extensive range of Gaussian and non-Gaussian regression tasks, we show that AR CNPs are consistently competitive with, and often significantly outperform, all other neural process models in terms of predictive log-likelihood.
- We deploy AR CNPs on a range of tasks involving real-world climate data. To handle the high-resolution data in a computationally tractable manner, we introduce a novel multi-scale architecture for ConvCNPs. We also combine AR ConvCNPs with a beta-categorical mixture likelihood, producing strong results compared to other neural processes.

Our work represents a promising first application of this procedure to the simplest class of neural processes, and motivates future work on applications of AR sampling to other neural process models.

## 2 Autoregressive Conditional Neural Processes

**Meta-learning.** We first define the problem setup. Let $\mathcal{X}$ be a compact input space and let $\mathcal{Y}$ be the output space. Let $\mathcal{D}_N = (\mathcal{X} \times \mathcal{Y})^N$ be the collection of all sets of $N$ input–output pairs, and let $\mathcal{D} = \bigcup_{N=0}^{\infty} \mathcal{D}_N$. We call elements $D \in \mathcal{D}$ *data sets* and denote $D = (\mathbf{x}, \mathbf{y})$ where $\mathbf{x} \in \mathcal{X}^N$, $\mathbf{y} \in \mathcal{Y}^N$ are the inputs and outputs respectively. In meta-learning we are given a collection of data sets $(D_m)_{m=1}^M$, called a *meta–data set*, with the individual data sets $D_m$ called *tasks* (Vinyals et al., 2016). Every task $D_m$ is split up $D_m = D_m^{(c)} \cup D_m^{(t)}$ into a *context set* $D_m^{(c)} = (\mathbf{x}_m^{(c)}, \mathbf{y}_m^{(c)})$ and a *target set* $D_m^{(t)} = (\mathbf{x}_m^{(t)}, \mathbf{y}_m^{(t)})$. Here $\mathbf{x}_m^{(c)}$ are called the *context inputs*, $\mathbf{y}_m^{(c)}$ the *context outputs*, $\mathbf{x}_m^{(t)}$ the *target inputs*, and $\mathbf{y}_m^{(t)}$ the *target outputs*. Our goal is to devise an algorithm which takes in a context set $D_m^{(c)}$ and produces the best possible prediction for the target outputs $\mathbf{y}_m^{(t)}$ given target inputs $\mathbf{x}_m^{(t)}$.

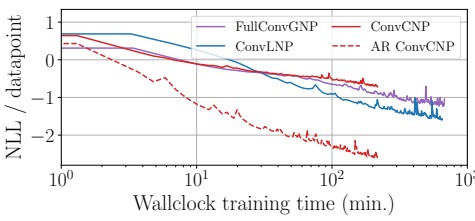

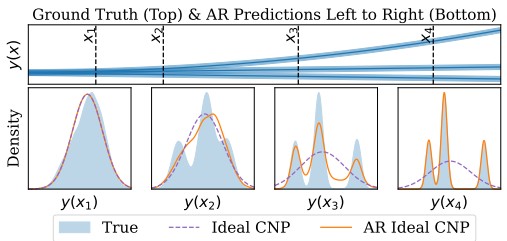

Figure 2: Negative log-likelihoods on non-Gaussian sawtooth data. Deploying the ConvCNP in AR mode dramatically improves performance, and outperforms state-of-the-art NPs with Gaussian (FullConvGNP) and non-Gaussian (ConvLNP) predictive distributions, at a fraction of the training cost.

Figure 3: *Top:* generative process: mixture model of three deterministic functions with additive Gaussian noise. *Bottom:* at the four target locations indicated by dashed lines, the panes show the true distribution and predictions by the ideal CNP and the ideal CNP applied autoregressively at the targets from left to right. See Appendices D.3 and E.

**Neural processes.** Let $\mathcal{P}$ be the set of all $\mathcal{Y}$-valued stochastic processes on $\mathcal{X}$. Neural processes (NPs) directly and flexibly parametrise a map $\pi_\theta\colon \mathcal{D} \to \mathcal{Q}$ where $\mathcal{Q} \subseteq \mathcal{P}$ and where $\theta$ are learnable parameters. CNPs set $\mathcal{Q}$ to be the collection of GPs $f$ such that $\mathrm{cov}(f(x), f(y)) = 0$ for $x \neq y$. GNPs let $\mathcal{Q}$ be the collection of continuous GPs. Latent NPs (LNPs; Garnelo et al., 2018b) let $\mathcal{Q}$ be a collection of non-Gaussian processes by making use of a latent variable. Let $P_{\mathbf{x}_m^{(t)}} \pi(D_m^{(c)})$ denote the finite-dimensional distribution of the process $\pi(D_m^{(c)})$ evaluated at inputs $\mathbf{x}_m^{(t)}$, and denote its density by $q_\theta(\,\cdot\, | \mathbf{x}_m^{(t)}, D_m^{(c)})$. To learn the parameters $\theta$, NPs seek to maximise

$$\mathcal{L}_M(\pi) = \tfrac{1}{M} \sum_{m=1}^M \log q_\theta(\mathbf{y}_m^{(t)} \,|\, \mathbf{x}_m^{(t)}, D_m^{(c)}). \tag{1}$$

For CNPs and GNPs, $\mathcal{L}_M$ can be computed exactly, since $q_\theta$ is Gaussian.[1] However, for LNPs, $\mathcal{L}_M$ must be approximated (Garnelo et al., 2018b; Foong et al., 2020), typically impacting performance.

**Autoregressive CNPs.** Our proposal is to take an existing CNP and run it in an autoregressive fashion, feeding predictions for earlier outputs back into the model. Inspired by the product rule, we define the joint predictive as a product of conditionals, modelling each conditional with a CNP. For example, in the case of three target points, $q_\theta(y_3^{(t)} \,|\, \mathbf{y}_{1:2}^{(t)}, D_m^{(c)}) q_\theta(y_2^{(t)} \,|\, y_1^{(t)}, D_m^{(c)}) q_\theta(y_1^{(t)} \,|\, D_m^{(c)})$. To enable a theoretical analysis of this procedure, we now proceed to set up more formal notation. Suppose that $\pi_\theta\colon \mathcal{D} \to \mathcal{Q}$ is an NP, and we wish to predict at some target inputs $\mathbf{x}^{(t)}$ given a context set $D^{(c)}$. Standard NPs would output the predictive $P_{\mathbf{x}^{(t)}} \pi_\theta(D^{(c)})$ which, for CNPs, would be a factorised Gaussian. We propose to instead roll out the NP autoregressively, as described in Proc. 2.1.

**Procedure 2.1** (Autoregressive application of neural process). For a neural process $\pi_\theta$, context set $D^{(c)} = (\mathbf{x}^{(c)}, \mathbf{y}^{(c)})$, and target inputs $\mathbf{x}^{(t)}$, let $\mathrm{AR}_{\mathbf{x}^{(t)}}(\pi_\theta, D^{(c)})$ be the distribution defined as follows:

$$\text{for } i = 1, \dots, N, \quad y_i^{(t)} \sim P_{x_i^{(t)}} \pi_\theta(\mathbf{x}^{(c)} \oplus \mathbf{x}_{1:(i-1)}^{(t)}, \mathbf{y}^{(c)} \oplus \mathbf{y}_{1:(i-1)}^{(t)}), \tag{2}$$

where $\mathbf{a} \oplus \mathbf{b}$ concatenates two vectors $\mathbf{a}$ and $\mathbf{b}$. See Figure 7 in Appendix C for an illustration.

Since earlier samples $y_i^{(t)}$ feed back into later applications of $\pi_\theta$, the whole sample $\mathbf{y}^{(t)}$ is correlated, *even if $\pi_\theta$ does not model dependencies between target outputs*, as with CNPs. At test time, when evaluating the corresponding the density $q_\theta^{(\mathrm{AR})}$ of $\mathrm{AR}_{\mathbf{x}^{(t)}}(\pi_\theta, D^{(c)})$ at $\mathbf{y}^{(t)}$, we use the formula

$$\log q_\theta^{(\mathrm{AR})}(\mathbf{y}^{(t)} \,|\, \mathbf{x}^{(t)}, D^{(c)}) = \sum_{i=1}^N \log q_\theta(y_i^{(t)} \,|\, x_i^{(t)}, D^{(c)} \oplus (\mathbf{x}_{1:(i-1)}^{(t)}, \mathbf{y}_{1:(i-1)}^{(t)})). \tag{3}$$

Whilst any NP can be used in AR, we focus on CNPs as they are the computationally cheapest class.

**Understanding the infinite data limit.** To better understand why AR CNPs successfully model dependencies, we analyse the idealised case of infinite data and model capacity. Let $p(f)$ be the law of the data-generating stochastic process, and let $p(\varepsilon)$ be the law of a stochastic process representing observation noise, defined by letting $\varepsilon(\mathbf{x})$ be a vector of i.i.d. noise variables for all $\mathbf{x}$. We assume

$$\mathbf{y}_m^{(c)} = y_m(\mathbf{x}_m^{(c)}) \quad \text{and} \quad \mathbf{y}_m^{(t)} = y_m(\mathbf{x}_m^{(t)}) \quad \text{where} \quad y_m(\,\cdot\,) = f_m(\,\cdot\,) + \varepsilon_m(\,\cdot\,), \tag{4}$$

$(f_m)_{m=1}^M$ are i.i.d. draws from $p(f)$, and $(\varepsilon_m)_{m=1}^M$ are i.i.d. draws from $p(\varepsilon)$. Define the *prediction map* $\pi_y\colon \mathcal{D} \to \mathcal{P}$ as the mapping from a data set to the posterior over $y$, $\pi_y(D) = p(y \,|\, D)$. Then $\mathcal{L}_M$ is a Monte Carlo approximation of the following infinite-sample objective (Foong et al., 2020):

$$\mathcal{L}_\infty(\pi) = -\mathbb{E}_{p(D^{(c)})p(\mathbf{x}^{(t)})}[\mathrm{KL}(P_{\mathbf{x}^{(t)}} \pi_y(D^{(c)}), P_{\mathbf{x}^{(t)}} \pi(D^{(c)}))] + \mathrm{const}. \tag{5}$$

---

[1]Unless otherwise specified, we assume CNPs use Gaussian likelihoods, as in Garnelo et al. (2018a). However, it is straightforward to modify them to use non-Gaussian likelihoods, as we do in Section 4.4.

Under appropriate regularity assumptions, $\mathcal{L}_\infty(\pi)$ is maximised over all $\pi$ when the expected KL divergence term is zero, which occurs if and only if $\pi = \pi_y$. In practice, NPs do not maximise $\mathcal{L}_\infty(\pi)$ over all $\pi$, but (i) use a finite-sized meta–data set and (ii) restrict $\mathcal{Q} \subseteq \mathcal{P}$:

$$\overbrace{\underset{\text{all } \pi:\, \mathcal{D} \to \mathcal{Q}}{\pi_M \in \arg\max} \mathcal{L}_M(\pi)}^{\text{what we compute in practice}} \xrightarrow[M \to \infty]{\text{(i)}} \overbrace{\underset{\text{all } \pi:\, \mathcal{D} \to \mathcal{Q}}{\pi_\infty \in \arg\max} \mathcal{L}_\infty(\pi)}^{\text{ideal NP}} \xrightarrow[\mathcal{Q} \to \mathcal{P}]{\text{(ii)}} \overbrace{\underset{\text{all } \pi:\, \mathcal{D} \to \mathcal{P}}{\pi_y = \arg\max} \mathcal{L}_\infty(\pi)}^{\text{exact prediction map}} \quad (6)$$

Here $\pi_M$ is an NP trained on the practical objective (1), which, in the limit of infinite data, approximates the so-called *ideal NP* $\pi_\infty$. The ideal NP depends on the choice of $\mathcal{Q}$, *i.e.* the class of NPs under consideration, and, in turn, approximates $\pi_y$. For CNPs and GNPs, using the fact that minimising $\mathrm{KL}(p, q)$ over $q$ matches moments (Minka, 2001), we can readily determine and even practically compute the ideal NP for these two classes of NPs. The *ideal CNP* predicts a diagonal-covariance GP whose mean function and *marginal variance function* match $\pi_y$: $\pi_\infty(D) = \mathcal{GP}(m, k)$ where $m(x) = \mathbb{E}[y(x) \mid D]$, and $k(x, x') = \mathbb{V}[y(x) \mid D]$ if $x = x'$ and $k(x, x') = 0$ otherwise. On the other hand, the *ideal GNP* predicts a GP whose mean function and *full covariance function* match $\pi_y$: $\pi_\infty(D) = \mathcal{GP}(m, k)$ where $m(x) = \mathbb{E}[y(x) \mid D]$, $k(x, x') = \mathrm{cov}(y(x), y(x') \mid D)$. The main result of this subsection is that the ideal CNP, despite not modelling correlations, becomes superior to the ideal GNP *when deployed in AR mode*:

**Proposition 2.1** (Advantage of AR CNPs over GNPs). Assume appropriate regularity conditions on $y$. Let $\pi_C$ be the ideal CNP and let $\pi_G$ be the ideal GNP. Then, for all inputs $\mathbf{x}$ and data sets $D \in \mathcal{D}$,

$$\mathrm{KL}(P_\mathbf{x} \pi_y(D), \mathrm{AR}_\mathbf{x}(\pi_C, D)) \leq \mathrm{KL}(P_\mathbf{x} \pi_y(D), P_\mathbf{x} \pi_G(D)). \quad (7)$$

We provide a proof in Appendix A. Intuitively, the advantage of AR CNPs comes from their ability to model non-Gaussian dependencies. Proposition 2.1 shows that to outperform the GNP, it suffices to train a CNP to model the *marginals* of $\pi_y$, and rely on the AR procedure to induce dependencies. A visualisation of the ideal CNP and the ideal CNP applied autoregressively can be seen in Figure 3.

**Consistency and the AR design space.** As shown in Table 1, AR CNPs give up the fundamental property of *consistency*, since the distributions $\{\mathrm{AR}_\mathbf{x}(\pi_\theta, D_m^{(c)}) : \mathbf{x} \in \mathcal{X}^N, N \in \mathbb{N}\}$ are not consistent under permutation or marginalisation: permuting $\mathbf{x}$ and introducing or marginalising target points can change the distribution. This violates the conditions of the Kolmogorov extension theorem (Oksendal, 2013), preventing the distributions from defining a consistent stochastic process. There is thus a large design space involved when deploying AR CNPs, where choices that have no effect on the predictions of other NPs can now significantly affect performance.

One such choice is how many points to sample at a time. Sampling one at a time induces dependencies between all points, but requires $N$ forward passes. Alternatively, we could divide the $N$ inputs in $\mathbf{x}^{(t)}$ into blocks of $K$ points each, and sample each block with a single CNP forward pass. This requires $N/K$ forward passes, with points in the same block conditionally independent. If $K = N$, this is the standard CNP prediction; and if $K = 1$, we recover Procedure 2.1. This provides a knob for practitioners to trade off between faster, consistent, but less expressive standard CNP predictions; and slower, less consistent, but more expressive AR predictions. In this paper, we use full AR mode with $K = 1$, and leave an investigation of block AR sampling to future work.

**Obtaining smooth samples.** Due to the lack of consistency in AR mode, the spacing chosen between target points can significantly affect performance. For example, care must be taken so the number of target points is not much greater than the size of the context sets seen during train time, to avoid confronting the model with an out-of-distribution context set size at test time. This raises the question of how to sample functions on a very fine grid. Furthermore, since CNPs do not differentiate between epistemic and aleatoric uncertainty, it is not clear how to obtain smooth, noiseless samples, that is, samples for $f$ uncorrupted by the i.i.d. noise $\varepsilon$ in (4). The following proposition shows that, for a smooth sample corrupted by additive noise, the smooth component can be approximated with the predictive mean conditioned on noisy observations:

**Proposition 2.2** (Recovery of smooth samples). Let $\mathcal{X} \subseteq \mathbb{R}$ be compact, and let $f$ be a stochastic process with surely continuous sample paths and $\sup_{x \in \mathcal{X}} \|f(x)\|_{L^2} < \infty$. Let $(\varepsilon_n)_{n \geq 0}$ be i.i.d. (potentially non-Gaussian) random variables such that $\mathbb{E}[\varepsilon_0] = 0$ and $\mathbb{V}[\varepsilon_0] < \infty$. Consider any sequence $(x_n)_{n \geq 1} \subseteq \mathcal{X}$, and let $x^* \in \mathcal{X}$ be a limit point of $(x_n)_{n \geq 1}$. If $y(x^*) = f(x^*) + \varepsilon_0$ and $y_n = f(x_n) + \varepsilon_n$ are noisy observations of $f$, then

$$\lim_{n \to \infty} \mathbb{E}[y(x^*) \mid y_1, \ldots, y_n] = f(x^*) \quad \text{almost surely.} \quad (8)$$

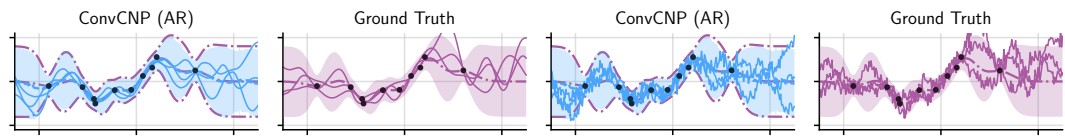

Figure 4: Comparison of noiseless (*left*) and noisy (*right*) samples from an AR ConvCNP trained on data sampled from a GP with an exponentiated-quadratic kernel, and the ground truth GP. The noiseless AR samples were generated from the noisy AR samples using the procedure suggested by Proposition 2.2.

We provide a proof in Appendix B. Equation (8) suggests the following two-step procedure for obtaining smooth samples from AR CNPs. **Step 1:** Let $\mathbf{x}_{1:n}$ be a number of target inputs that does not exceed the number of points seen during training. Sample $\mathbf{y}_{1:n} \sim \text{AR}_{\mathbf{x}_{1:n}}(\pi_\theta, D_m^{(c)})$. This sample includes observation noise. **Step 2:** Remove the noise from the sample by passing it through the model once more: $\mathcal{N}(\boldsymbol{\mu}_{1:n}, \mathbf{D}) = P_{\mathbf{x}_{1:n}}\pi_\theta(D_m^{(c)} \oplus (\mathbf{x}_{1:n}, \mathbf{y}_{1:n}))$. Here the predictive mean $\boldsymbol{\mu}_{1:n}$ forms the noiseless sample. To produce a sample at arbitrarily many inputs, one may also evaluate $\mathcal{N}(\boldsymbol{\mu}'_{1:n}, \mathbf{D}) = P_{\mathbf{x}'_{1:n}}\pi_\theta(D_m^{(c)} \oplus (\mathbf{x}_{1:n}, \mathbf{y}_{1:n}))$ where $\mathbf{x}'_{1:n}$ is arbitrary. This result of this procedure is illustrated in Figure 4, and was used to generate the noiseless samples shown in Figure 1 (right). Figure 7 in Appendix C also illustrates this two-step procedure in a pictorial step-by-step fashion.

## 3 CONNECTIONS TO OTHER NEURAL DISTRIBUTION ESTIMATORS

Various paradigms have been developed for neural distribution estimators (NDEs): normalising flows (Dinh et al., 2015), generative adversarial networks (GANs; Goodfellow et al., 2014), variational autoencoders (VAEs; Kingma & Welling, 2014), autoregressive models (Uria et al., 2016), and diffusion models (Sohl-Dickstein et al., 2015; Ho et al., 2020). Figure 5 visualises the landscape of NDEs. We argue that NPs and AR CNPs should be viewed as neural distribution estimators (NDEs) and be placed in this landscape. AR CNPs inherit the strengths of AR models, such as the ability to model complex dependencies with a tractable likelihood, but also some of their weaknesses, most notably slow test-time sampling. Slow sampling is the main drawback of AR CNPs, though

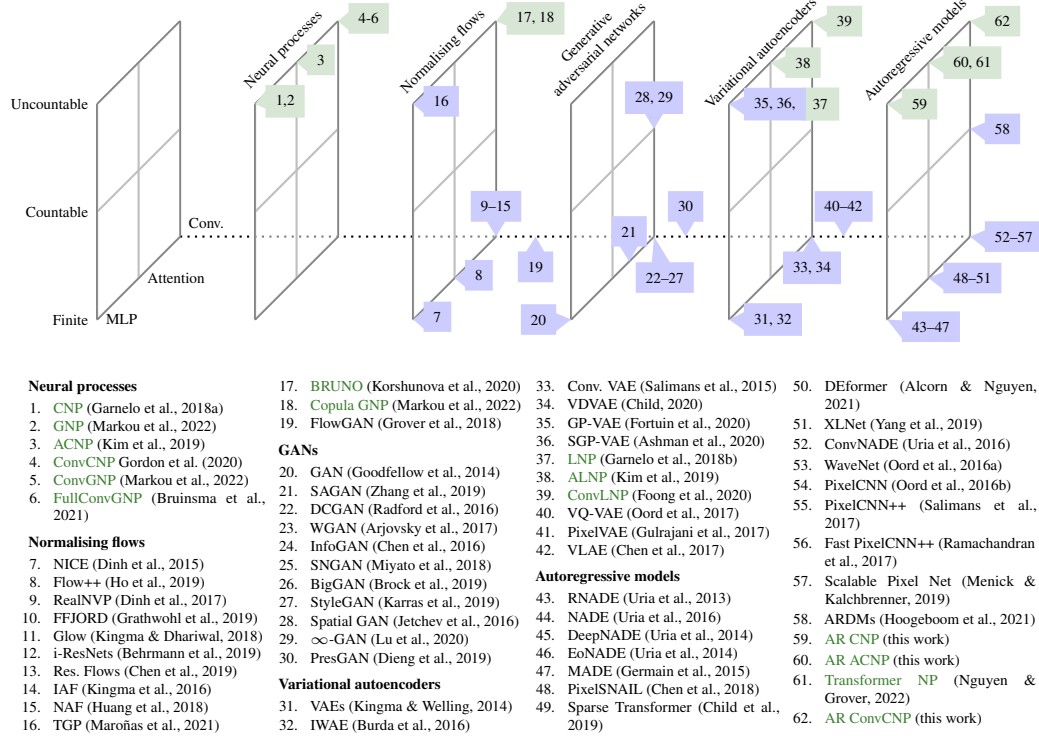

**Neural processes**

1. CNP (Garnelo et al., 2018a)
2. GNP (Markou et al., 2022)
3. ACNP (Kim et al., 2019)
4. ConvCNP Gordon et al. (2020)
5. ConvGNP (Markou et al., 2022)
6. FullConvGNP (Bruinsma et al., 2021)

**Normalising flows**

7. NICE (Dinh et al., 2015)
8. Flow++ (Ho et al., 2019)
9. RealNVP (Dinh et al., 2017)
10. FFJORD (Grathwohl et al., 2019)
11. Glow (Kingma & Dhariwal, 2018)
12. i-ResNets (Behrmann et al., 2019)
13. Res. Flows (Chen et al., 2019)
14. IAF (Kingma et al., 2016)
15. NAF (Huang et al., 2018)
16. TGP (Maroñas et al., 2021)

17. BRUNO (Korshunova et al., 2020)
18. Copula GNP (Markou et al., 2022)
19. FlowGAN (Grover et al., 2018)

**GANs**

20. GAN (Goodfellow et al., 2014)
21. SAGAN (Zhang et al., 2019)
22. DCGAN (Radford et al., 2016)
23. WGAN (Arjovsky et al., 2017)
24. InfoGAN (Chen et al., 2016)
25. SNGAN (Miyato et al., 2018)
26. BigGAN (Brock et al., 2019)
27. StyleGAN (Karras et al., 2019)
28. Spatial GAN (Jetchev et al., 2016)
29. ∞-GAN (Lu et al., 2020)
30. PresGAN (Dieng et al., 2019)

**Variational autoencoders**

31. VAEs (Kingma & Welling, 2014)
32. IWAE (Burda et al., 2016)

33. Conv. VAE (Salimans et al., 2015)
34. VDVAE (Child, 2020)
35. GP-VAE (Fortuin et al., 2020)
36. SGP-VAE (Ashman et al., 2020)
37. LNP (Garnelo et al., 2018b)
38. ALNP (Kim et al., 2019)
39. ConvLNP (Foong et al., 2020)
40. VQ-VAE (Oord et al., 2017)
41. PixelVAE (Gulrajani et al., 2017)
42. VLAE (Chen et al., 2017)

**Autoregressive models**

43. RNADE (Uria et al., 2013)
44. NADE (Uria et al., 2016)
45. DeepNADE (Uria et al., 2014)
46. EoNADE (Uria et al., 2014)
47. MADE (Germain et al., 2015)
48. PixelSNAIL (Chen et al., 2018)
49. Sparse Transformer (Child et al., 2019)

50. DEformer (Alcorn & Nguyen, 2021)
51. XLNet (Yang et al., 2019)
52. ConvNADE (Uria et al., 2016)
53. WaveNet (Oord et al., 2016a)
54. PixelCNN (Oord et al., 2016b)
55. PixelCNN++ (Salimans et al., 2017)
56. Fast PixelCNN++ (Ramachandran et al., 2017)
57. Scalable Pixel Net (Menick & Kalchbrenner, 2019)
58. ARDMs (Hoogeboom et al., 2021)
59. AR CNP (this work)
60. AR ACNP (this work)
61. Transformer NP (Nguyen & Grover, 2022)
62. AR ConvCNP (this work)

Figure 5: Conceptual diagram showing the relationships between AR CNPs and various neural distribution estimators. The vertical axis denotes whether the model learns a distribution over a finite number of random variables, a countably infinite number, or an uncountably infinite number. The axis into the page denotes whether the architecture is MLP-based, or uses attention or convolutions. From left to right, we show different modelling paradigms. Fruitful exchanges occur when NPs (highlighted in green) are introduced into other modelling paradigms. Our proposed AR CNPs can be viewed as introducing NPs to the AR modelling paradigm.

it may be possible to adapt techniques for speeding up AR models (Ramachandran et al., 2017). One major difference between AR CNPs and existing AR models is that AR CNPs decompose the joint distribution of an *uncountably* infinite set of variables, allowing querying at arbitrary input locations (Section 2). Like DEformer (Alcorn & Nguyen, 2021), EoNADE (Uria et al., 2014), and XLnet (Yang et al., 2019), AR CNPs are trained to not prefer any particular order of decomposing the joint distribution into conditionals. To achieve this goal, the AR CNP shares design choices with other AR models: (i) a shared architecture is used to produce each conditional distribution, similar to WaveNet (Oord et al., 2016a) and PixelCNN (Oord et al., 2016b); (ii) the data point index is given as input to the network as in the DEformer model (Alcorn & Nguyen, 2021); and (iii) training maximises a log-likelihood including all decompositions of the joint distribution, similar to EoNADE (Uria et al., 2014) and XLnet (Yang et al., 2019).

Figure 5 also shows the connections between NPs, VAEs and normalising flows (NFs). Like VAEs, LNPs use decoders that parametrise a factorised distribution and rely on the latent variable to induce dependencies. Again, the key difference is that LNPs model a distribution over an *uncountable* set of variables. Models like conditional BRUNO (Korshunova et al., 2020) and copula GNPs (Markou et al., 2022) combine ideas from NPs and NFs, transforming a stochastic process with an invertible transformation. Finally, GAN models such as Spatial GAN (Jetchev et al., 2016) and $\infty-$GAN (Lu et al., 2020) model countable numbers of variables, such as images of arbitrary size. Inspecting Figure 5, we see that GANs are the only class of models depicted that do not currently have an NP version: a version that models an uncountable number of variables. This suggests adversarial training of NPs as an interesting avenue for future investigation.

In recent work, Nguyen & Grover (2022) proposed the Transformer NP (TNPs), which uses a causally-masked transformer architecture with an autoregressive likelihood. In contrast, rather than proposing a new AR architecture, our work focuses on running *existing CNPs* in AR mode to obtain coherent samples and improved likelihoods, without modifying the architecture or training procedure. In prior work, Volpp et al. (2021) used AR sampling in order to visualise samples from CNPs. However, their work focuses on proposing a novel context aggregation mechanism for NPs, and they do not evaluate the likelihood of CNPs autoregressively or investigate any performance gains.

## 4 THE PERFORMANCE OF AUTOREGRESSIVE NEURAL PROCESSES

In this section we investigate the performance of AR CNPs on synthetic and real data. Across a wide range of tasks, the AR CNP is competitive with much more sophisticated approaches. Throughout, we train LNPs with both the ELBO (Garnelo et al., 2018b) and ML objective (Foong et al., 2020). Code for implementations of NPs and reproducing our experiments can be found at `https://github.com/wesselb/neuralprocesses`. For all experiments, we use a random ordering of the target points in Proc. 2.1; see App. D for a justification.

### 4.1 SYNTHETICALLY GENERATED GAUSSIAN AND NON-GAUSSIAN DATA

**Synthetic experiment setup.** We evaluate an extensive collection of NP models on a wide range of Gaussian and non-Gaussian synthetic regression tasks. We consider tasks with functions drawn from (i) different GPs; (ii) a non-Gaussian sawtooth process (as in Figure 1); (iii) a non-Gaussian mixture task, where, with equal probability, we sample the function from one of three possible GPs or the sawtooth process. We also consider various versions of the tasks for different input and output dimension $d_x, d_y$, with dependencies across the output channels. To ensure a fair comparison, we configure the architectures to make the parameter counts comparable between all models.

**Results.** Table 2 highlights the best performing models on some representative tasks; for further results across all twenty synthetic tasks and further experimental details, see Appendix H. The AR procedure dramatically improves the performance of the ConvCNP, with the AR ConvCNP being the best performing model for most tasks, except on the Gaussian EQ task where it performs marginally worse than the FullConvGNP. In particular, the AR ConvCNP outperforms the FullConvGNP and ConvGNP on non-Gaussian tasks, in agreement with Proposition 2.1, while having a faster training time than the other convolutional models (Figure 2). For the sawtooth task, Figure 11 in Appendix H.2 illustrates that predictions by the AR ConvCNP can be multi-modal and non-Gaussian, even when using a Gaussian likelihood. Finally, we note that in tasks with $d_x = 2$, where the FullConvGNP cannot be used (as discussed in Section 1), the AR ConvCNP far outperforms all competing approaches.

### 4.2 SIM-TO-REAL TRANSFER WITH THE LOTKA−VOLTERRA EQUATIONS

**Predator-prey data.** We next investigate sim-to-real transfer, where the models are trained on simulated data and tested on real data. NPs are well-suited to this setting, since a large meta-data set

| | EQ | | Sawtooth | | Mixture | |
| | Norm. KL to truth (↓ better) | | Norm. log-lik. (↑ better) | | Norm. log-lik. (↑ better) | |
| | $d_x, d_y = 1$ | $d_x, d_y = 2$ | $d_x, d_y = 1$ | $d_x, d_y = 2$ | $d_x, d_y = 1$ | $d_x, d_y = 2$ |
|---|---|---|---|---|---|---|
| ConvCNP | $0.41_{\pm 0.01}$ | $0.41_{\pm 0.00}$ | $2.38_{\pm 0.04}$ | $0.12_{\pm 0.01}$ | $-0.23_{\pm 0.04}$ | $-0.85_{\pm 0.01}$ |
| ConvCNP (AR) | $0.01_{\pm 0.00}$ | $\mathbf{0.03}_{\pm 0.00}$ | $\mathbf{3.60}_{\pm 0.01}$ | $\mathbf{0.38}_{\pm 0.00}$ | $\mathbf{0.45}_{\pm 0.04}$ | $\mathbf{-0.62}_{\pm 0.01}$ |
| ConvGNP | $0.01_{\pm 0.00}$ | $0.19_{\pm 0.00}$ | $2.62_{\pm 0.05}$ | $0.26_{\pm 0.01}$ | $-0.24_{\pm 0.02}$ | $-0.74_{\pm 0.01}$ |
| FullConvGNP | $\mathbf{0.00}_{\pm 0.00}$ | | $2.16_{\pm 0.04}$ | | $-0.05_{\pm 0.03}$ | |
| ConvLNP (ML) | $0.25_{\pm 0.01}$ | $0.39_{\pm 0.00}$ | $3.06_{\pm 0.04}$ | $0.31_{\pm 0.01}$ | $-0.06_{\pm 0.03}$ | $-0.78_{\pm 0.02}$ |
| ConvLNP (ELBO) | $0.06_{\pm 0.00}$ | $0.79_{\pm 0.00}$ | $3.51_{\pm 0.02}$ | $0.04_{\pm 0.00}$ | $0.12_{\pm 0.04}$ | $-0.92_{\pm 0.01}$ |
| *Diagonal GP* | $0.40_{\pm 0.01}$ | $0.40_{\pm 0.00}$ | | | | |
| *Trivial* | $1.19_{\pm 0.00}$ | $0.79_{\pm 0.00}$ | $-0.18_{\pm 0.00}$ | $-0.32_{\pm 0.00}$ | $-1.32_{\pm 0.00}$ | $-1.46_{\pm 0.00}$ |

Table 2: Performance of NPs training on the GP EQ task, sawtooth task, and mixture task. *Diagonal GP* denotes the exact GP predictive, but with correlations removed. *Trivial* denotes a model that predicts a Gaussian distribution with the empirical mean and standard deviation of the context outputs. Significantly best models in bold. Note that the FullConvGNP cannot be run on tasks where $d_x > 1$.

| | Int. (S) | For. (S) | Rec. (S) | Int. (R) | For. (R) | Rec. (R) |
|---|---|---|---|---|---|---|
| ConvCNP | $-3.47_{\pm 0.02}$ | $-4.06_{\pm 0.02}$ | $-4.85_{\pm 0.02}$ | $-4.17_{\pm 0.04}$ | $-4.70_{\pm 0.06}$ | $-4.97_{\pm 0.01}$ |
| ConvCNP (AR) | $\mathbf{-3.30}_{\pm 0.02}$ | $\mathbf{-3.47}_{\pm 0.02}$ | $\mathbf{-3.60}_{\pm 0.02}$ | $\mathbf{-4.10}_{\pm 0.03}$ | $\mathbf{-4.27}_{\pm 0.03}$ | $\mathbf{-4.32}_{\pm 0.01}$ |
| ConvGNP | $-3.47_{\pm 0.02}$ | $-3.65_{\pm 0.02}$ | $-4.15_{\pm 0.02}$ | $-4.21_{\pm 0.05}$ | $-4.82_{\pm 0.13}$ | $-4.61_{\pm 0.01}$ |
| FullConvGNP | $\mathbf{-3.29}_{\pm 0.02}$ | $\mathbf{-3.46}_{\pm 0.02}$ | $-3.79_{\pm 0.02}$ | $-4.16_{\pm 0.04}$ | $\mathbf{-4.28}_{\pm 0.04}$ | $-4.45_{\pm 0.00}$ |
| ConvLNP (ML) | $-3.41_{\pm 0.02}$ | $-3.84_{\pm 0.02}$ | $-4.44_{\pm 0.02}$ | $\mathbf{-4.13}_{\pm 0.04}$ | $-4.45_{\pm 0.05}$ | $-4.54_{\pm 0.01}$ |
| ConvLNP (ELBO) | $-3.77_{\pm 0.02}$ | $-3.83_{\pm 0.02}$ | $-4.12_{\pm 0.02}$ | $-5.45_{\pm 0.05}$ | $-5.47_{\pm 0.07}$ | $-6.39_{\pm 0.05}$ |

Table 3: Normalised log-likelihoods in the predator–prey experiments, showing interpolation (int.), forecasting (for.), and reconstruction (rec.) on simulated (S) and real (R) data. Significantly best results in bold.

can be easily generated to train them. We consider the hare–lynx data set, which is a population time series of Snowshoe hares and Canadian lynx (MacLulich, 1937). To generate simulated data, we use a stochastic version of the Lotka–Volterra equations (Lotka, 1910; Volterra, 1926):

$$\mathrm{d}X_t = \alpha X_t\,\mathrm{d}t - \beta Y_t X_t\,\mathrm{d}t + \sigma X_t^\nu\,\mathrm{d}W_t^{(1)}, \quad \mathrm{d}Y_t = -\gamma X_t\,\mathrm{d}t + \delta Y_t X_t\,\mathrm{d}t + \sigma Y_t^\nu\,\mathrm{d}W_t^{(2)}. \quad (9)$$

Under these equations, the prey population $X_t$ grows exponentially with rate $\alpha$, the predator population $Y_t$ decays exponentially with rate $\gamma$, and the predators hunt the prey. $W^{(1)}$ and $W^{(2)}$ are independent Brownian motions introducing noisy behaviour. These equations generate non-Gaussian data with both within-channel as well as cross-channel dependencies. We simulate the Lotka-Volterra equations on a dense grid, and use them to generate meta–data sets in three different ways. *Interpolation*: we randomly divide the data into context and target sets. *Forecasting*: we choose a random time, before which all data are contexts, and all future data are targets. *Reconstruction:* we randomly choose between the $X_t$ or $Y_t$, split the chosen series as in forecasting, and append the other series to the context. In training, for every batch, we choose one of these tasks uniformly at random.

**Results.** Table 3 shows the results of the best performing models. The AR ConvCNP performs best both on the simulated as well as the real data, again demonstrating that running CNPs in AR mode improves performance and can even outperform strong GNP and LNP baselines. For full experimental details and additional results see Appendix I.

### 4.3 ELECTROENCEPHALOGRAM EXPERIMENTS

**Electroencephalogram data.** We next trained various NPs on real time series data consisting of electroencephalogram (EEG) measurements (Zhang et al., 1995), following Markou et al. (2022). Each time series consists of 256 regularly spaced measurements across 7 correlated channels. For each channel, we randomly select a number of the 256 points uniformly at random to be target points, and use the remaining ones as context points, independently across the channels.

**Results.** After training, we test the models on this interpolation task and also on a reconstruction task, where we set a part of a channel as target and the remainder as context. In Table 4, we observe that the AR ConvCNP is competitive with the FullConvGNP, despite having significantly shorter training times and fewer parameters. Both the AR ConvCNP and the FullConvGNP outperform the ConvCNP and the ConvLNP. Full experimental detail are in Appendix J.

### 4.4 ENVIRONMENTAL MODELLING

Environmental datasets bring a range of modelling challenges. One example is fusing spatio-temporal data from disparate sources (Chang & Bai, 2018; Lahat et al., 2015), which arises in diverse

|      | ConvCNP | ConvCNP (AR) | ConvGNP | FullConvGNP | ConvLNP (ML) | ConvLNP (ELBO) |
|------|---------|--------------|---------|-------------|--------------|----------------|
| Int. | $-1.02_{\pm 0.01}$ | $\mathbf{-0.34}_{\pm 0.01}$ | $-0.93_{\pm 0.01}$ | $\mathbf{-0.35}_{\pm 0.01}$ | $-1.04_{\pm 0.01}$ | $-1.20_{\pm 0.01}$ |
| Rec. | $-2.07_{\pm 0.03}$ | $\mathbf{-0.63}_{\pm 0.01}$ | $-1.45_{\pm 0.03}$ | $\mathbf{-0.57}_{\pm 0.01}$ | $-1.53_{\pm 0.02}$ | $-2.00_{\pm 0.06}$ |

Table 4: Normalised log-likelihoods on the EEG experiments. Significantly best results in bold.

|          | GAUSSIAN | | | | BETA-CATEGORICAL | | |
|----------|----------|--------------|---------|--------------|------------------|---------|--------------|
|          | ConvGNP  | ConvLNP (ML) | ConvCNP | ConvCNP (AR) | ConvLNP (ML)     | ConvCNP | ConvCNP (AR) |
| Log-lik. | $0.60_{\pm 0.02}$ | $0.62_{\pm 0.02}$ | $0.58_{\pm 0.02}$ | $\mathbf{0.88}_{\pm 0.02}$ | $1.06_{\pm 0.02}$ | $1.03_{\pm 0.02}$ | $\mathbf{1.27}_{\pm 0.02}$ |
| MAE (%)  | $13.05_{\pm 0.17}$ | $12.98_{\pm 0.16}$ | $13.01_{\pm 0.16}$ | $13.01_{\pm 0.16}$ | $12.99_{\pm 0.16}$ | $13.13_{\pm 0.16}$ | $13.13_{\pm 0.16}$ |

Table 5: Normalised log-likelihoods and mean absolute errors (MAE, in units of cloud cover %), over the 2019-2019 test period for the cloud cover task. Note that log-likelihoods cannot be compared directly across the Gaussian and beta-categorical models. Errors indicate standard errors. Significantly best results in bold.

environmental sciences applications from climate monitoring to hydrology (Gettelman et al., 2022; Ferrer-Cid et al., 2020; Robinson et al., 2021; Lu et al., 2010; Hosseini & Kerachian, 2017). Another challenge involves estimating the probability of events of interest, such as the compound risk of *both* low wind speeds at an offshore wind farm *and* high cloud cover over a solar panel farm. To obtain robust uncertainty estimates for such events, it is essential to model correlations as well as non-Gaussian marginals (such as cloud cover). Current GAN-based approaches (e.g. Ravuri et al. 2021) can capture both joint and non-Gaussian statistics, but they cannot perform data fusion or predict at arbitrary off-grid locations. The AR ConvCNP can fuse data of on-grid and off-grid modalities and make predictions at arbitrary locations while modelling arbitrary non-Gaussian likelihoods and capturing statistical dependencies, thus achieving all the desiderata and filling a gap in the environmental modelling toolbox. Here, we assess the AR ConvCNP on two common environmental modelling tasks, namely *data assimilation* and *statistical downscaling*.

**Data assimilation.** Data assimilation is the task of combining observations of the Earth system to produce predictions on a regular grid, called a *reanalysis*. Reanalyses are typically generated by fitting the trajectories of physics-based climate models to observations (Hersbach et al., 2020; Gettelman et al., 2022), but the potential for improving data assimilation with ML has drawn increasing attention in recent years (Geer, 2021). To explore the AR ConvCNP's data assimilation abilities for a non-Gaussian variable, we train convolutional NP models to predict simulated daily-average cloud cover fraction over Antarctica. We use reanalysis data from ECMWF ERA5 (Hersbach et al., 2020) as ground truth. Cloud cover takes values in the interval $[0, 1]$, with observations frequently taking values of 0 or 1 (Figure 14). We evaluate the performance of NPs using either a Gaussian likelihood or a beta-categorical mixture model with three components: two discrete delta components for values of exactly 0 or 1, and a beta distribution capturing continuous values in $(0, 1)$. This provides a robust way of handling 0 and 1 values, unlike the existing copula GNP model (Markou et al., 2022) which can have its output constrained in $(0, 1)$ but places zero density at the endpoints.

**Data assimilation results.** In Table 5 we see that the AR ConvCNP significantly outperforms competing NPs for both the Gaussian and beta-categorical likelihoods. Figure 6 shows samples drawn from the models, after observing context points on half of the space. The AR ConvCNP displays remarkable ability to extrapolate rich, non-stationary, multi-scale structure, such as sudden changes in cloud cover over the Ross Ice Shelf coastline at the bottom of the figure. By comparison, the ConvLNP and ConvGNP produce blurry, lower frequency samples. Unlike GPs, convolutional NP models have a fixed receptive field induced by the CNN architecture used for the encoder, which is computationally expensive to increase. Away from the context points on the left, samples from the non-AR models will be independent of the observations, reverting to some mean representation of the data (Fig. 6c-e). This highlights a further benefit of AR CNPs: successive AR applications increase the receptive field, enabling rich, conditional sample structure to extrapolate far away from observed data. See Appendix K for further commentary, sample figures, and details.

**Environmental downscaling.** The spatial resolutions of physics-based reanalyses are limited by their vast computational demands, making them unsuitable for capturing local and extreme events (Stocker et al., 2013; Maraun et al., 2017). *Statistical downscaling* addresses this issue by leveraging additional information to produce fine-grained predictions (Maraun & Widmann, 2018). Recently, NPs have been shown to outperform a large ensemble of existing climate downscaling approaches (Vaughan et al., 2022). We compare the AR ConvCNP to the MLP ConvCNP of Vaughan et al. and the MLP ConvGNP of (Markou et al., 2022) in a temperature downscaling task over Germany. In this task, the context data consist of low-resolution ERA-Interim reanalysis data and high-resolution topography, and the target data consist of weather station observations from the ECA&D dataset. We also consider a second setup where we reveal some station observations to aid the downscaling

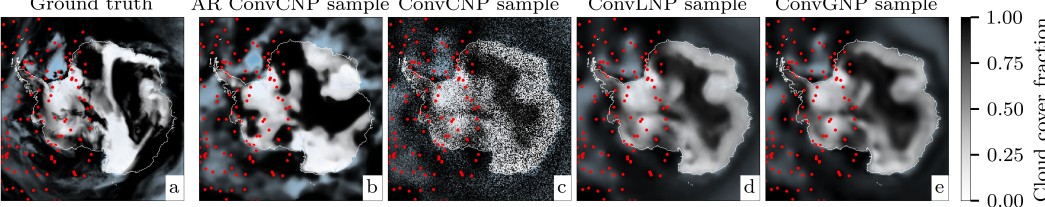

Figure 6: (a) Ground truth simulated cloud cover fraction on 25/06/2018. (b-e), Sample draws from the AR ConvCNP, ConvCNP, ConvLNP and ConvGNP with context points denoted by red dots. Context points were removed from the right hand side of the 2D space to test the models' abilities to extrapolate coherent function samples far away from observations. The ConvCNP and ConvLNP models used a beta-categorical likelihood while the ConvGNP uses a low-rank Gaussian likelihood.

process. As Appendix L.2 explains, the MLP ConvCNP and MLP ConvGNP cannot be extended to include these station observations. We therefore introduce a novel multiscale architecture, which we use to run the ConvCNP in AR mode. See Appendix L for full experimental details.

**Environmental downscaling results.** In Table 6 we observe that the AR ConvCNP matches the performance of the ConvGNP, which is remarkable as the latter has been previously demonstrated to outperform a range of state-of-the-art downscaling approaches (Markou et al., 2022; Vaughan et al., 2022). When additional observations from weather stations are revealed, the AR ConvCNP significantly outperforms the MLP ConvGNP in both metrics.

| Downscaling | Norm. log-lik. | MAE ($^\circ$C) | Down. + stations | Norm. log-lik. | MAE ($^\circ$C) |
|---|---|---|---|---|---|
| ConvCNP (MLP) | $-1.55 \pm 0.01$ | $\mathbf{0.94} \pm 0.03$ | ConvCNP$^*$ (MLP) | $-1.55 \pm 0.01$ | $0.94 \pm 0.03$ |
| ConvGNP (MLP) | $\mathbf{-1.36} \pm 0.01$ | $1.09 \pm 0.09$ | ConvGNP$^*$ (MLP) | $-1.38 \pm 0.01$ | $1.09 \pm 0.09$ |
| ConvCNP (AR) | $\mathbf{-1.36} \pm 0.01$ | $1.04 \pm 0.04$ | ConvCNP (AR) | $\mathbf{-1.31} \pm 0.01$ | $\mathbf{0.85} \pm 0.05$ |

Table 6: Normalised log-likelihoods and mean absolute errors (MAEs) in the downscaling experiments, without (*left*) and with (*right*) assisting weather station observations. Significantly best results in bold. $^*$Cannot use extra weather station observations.

## 5 LIMITATIONS AND CONCLUSION

We have shown that the AR procedure can be readily applied to improve the performance of CNPs, producing coherent samples and dramatically improved likelihoods. Surprisingly, in an extensive range of experiments, this simple approach often outperforms more complicated methods which rely on latent variables or which explicitly model correlations. We demonstrate the effectiveness of our approach on data sets of real-world interest by applying AR CNPs on climate data fusion tasks, modelling $[0, 1]$-constrained data with a beta-categorical likelihood and introducing a novel multiscale architecture. Notably, AR CNPs fill a gap in the climate modelling toolbox by enabling joint, non-Gaussian predictives, which could be used to better estimate the magnitude of compound risks. We also position AR CNPs within the larger neural density estimator literature, showing the fruitfulness of combining NPs with other modelling paradigms.

More generally, AR CNPs equip the NP framework with a new knob where modelling complexity and computational expense at training time can be traded for computational expense at test time. In particular, the higher quality samples and better likelihoods obtained by applying NPs autoregressively come with the additional cost of performing a forward pass for every element in the target set. This can be prohibitively expensive for large target sets, and constitutes the primary practical drawback of using AR CNPs. In addition, since AR CNPs do not define a consistent stochastic process, design choices for the AR procedure may affect the quality of the results. Thus practitioners need to avoid choosing target sets that lead to pathological behaviour, such as when the spatial density of the target inputs is too high. However, the flexibility of this design space also presents an opportunity: as an example, in Appendix M we show that auxiliary target points can be used to further improve predictions. Finally, promising avenues for future work include applying the AR procedure to other NPs besides CNPs, and investigating the efficacy of the block sampling scheme presented in Section 2.

## 6 REPRODUCIBILITY STATEMENT

All our experiments are carried out using either synthetic or publicly available datasets. The EEG data set is available through the UCI database,[2] and the environmental data are also publicly available through the European Climate Data Service.[3]

We make publicly available all code necessary to reproduce our experiments[4] as well as instructions for downloading, preprocessing, and modelling the Antarctic cloud cover data[5]. Proofs for Propositions 2.1 and 2.2 are given in Appendix A and Appendix B respectively. Details on the model architectures and the experimental setup can be found in Appendices F to H for the synthetic datasets, Appendix I the sim-to-real transfer experiments, Appendix J for the EEG experiments, Appendix K for the data assimilation experiment, and Appendix L for the downscaling experiment.

## 7 ETHICS STATEMENT

Training CNPs autoregressively improves their performance dramatically, but we do not foresee adverse societal impacts as a result of this work. That being said, the problem of capturing the statistical trends present in a dataset must be performed with care, especially in safety critical applications, where the stakes of making incorrect and confident predictions can have severe consequences. We view the AR procedure as a useful tool, rather than a panacea, for capturing such behaviours, and hope this work encourages further research into building effective but reliable models to this end.

We also note that while training CNPs is computationally cheaper than alternative NP models, AR sampling itself incurs a substantial computational cost, and thus energy cost, at test time. Running AR sampling on a large scale could lead to greater power demands for these models, resulting in larger carbon footprints which are undesirable. However, we believe the potential benefits for environmental modelling could outweigh this cost, while leveraging methods to make AR CNPs more computationally efficient should help alleviate this issue.

## 8 ACKNOWLEDGEMENTS

This research was conducted while WPB and AYKF were students at the University of Cambridge. During that time, WPB was supported by the Engineering and Physical Research Council (studentship number 10436152), and AYKF was supported by the Trinity Hall Studentship and the George and Lilian Schiff Foundation. SM acknowledges funding from the Vice Chancellor's & George and Marie Vergottis scholarship and the Qualcomm Innovation Fellowship. TRA and JSH are supported by Wave 1 of The UKRI Strategic Priorities Fund under the EPSRC Grant EP/W006022/1, particularly the AI for Science theme within that grant & The Alan Turing Institute. RET is supported by Google, Amazon, ARM, Improbable and EPSRC grant EP/T005386/1.

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

## A    PROOF OF PROPOSITION 2.1

**Additional notation.** If $\mathbf{y}_1 \oplus \mathbf{y}_2 \sim P_{\mathbf{x}_1 \oplus \mathbf{x}_2} \pi(D)$, then denote the distribution of $\mathbf{y}_1 \,|\, \mathbf{y}_2$ by $P_{\mathbf{x}_1 \,|\, \mathbf{x}_2} \pi(D)$. Note that $P_{\mathbf{x}_1 \,|\, \mathbf{x}_2} \pi(D)$ depends on $\mathbf{y}_2$, because it is the distribution of $\mathbf{y}_1 \,|\, \mathbf{y}_2$, even though the notation does not make this dependence explicit.

**The "appropriate regularity conditions".** Let $\mathcal{P}_\lambda^N$ be the collection of distributions on $\mathbb{R}^N$ that (a) have a density with respect to the Lebesgue measure and (b) have a covariance matrix which is strictly positive definite. Let $\mathcal{P}_{\lambda,G}^N \subseteq \mathcal{P}_\lambda^N$ be the subcollection of distributions which are Gaussian. Then, by Corollary B.1 by Bruinsma et al. (2021), for all $\mu \in \mathcal{P}_\lambda^N$ such that $\inf_{\nu \in \mathcal{P}_{\lambda,G}^N} \mathrm{KL}(\mu, \nu) < \infty$,

$$\arg\min_{\nu \in \mathcal{P}_{\lambda,G}^N} \mathrm{KL}(\mu, \nu) = \mathcal{N}(\mu) \tag{10}$$

where $\mathcal{N}(\mu)$ denotes the Gaussian distribution with mean vector and covariance matrix equal to those of $\mu$.

In the proposition, by appropriate regularity conditions on $y$, we mean the assumption that, for all inputs $\mathbf{x}$ and $D \in \mathcal{D}$, $P_{\mathbf{x}} \pi_y(D)$ is in $\mathcal{P}_\lambda^{|\mathbf{x}|}$ and such that $\inf_{\nu \in \mathcal{P}_{\lambda,G}^{|\mathbf{x}|}} \mathrm{KL}(P_{\mathbf{x}} \pi_y(D), \nu) < \infty$.

Assume the appropriate regularity conditions on $y$. We now list three technical observations.

(1) Note that $P_{\mathbf{x}_1 \,|\, \mathbf{x}_2} \pi_y(D)$ is the distribution of $y(\mathbf{x}_1) \,|\, D, (\mathbf{x}_2, \mathbf{y}_2)$, so we have the identity $P_{\mathbf{x}_1 \,|\, \mathbf{x}_2} \pi_y(D) = P_{\mathbf{x}_1} \pi_y(D \oplus (\mathbf{x}_2, \mathbf{y}_2))$. Therefore, for all inputs $\mathbf{x}_1$, inputs $\mathbf{x}_2$, and $D \in \mathcal{D}$, $P_{\mathbf{x}_1 \,|\, \mathbf{x}_2} \pi_y(D)$ is in $\mathcal{P}_\lambda^{|\mathbf{x}_1|}$ and such that $\inf_\nu \mathrm{KL}(P_{\mathbf{x}_1 \,|\, \mathbf{x}_2} \pi_y(D), \nu) < \infty$.

(2) The ideal CNP $\pi_C$ matches the means and marginal variances of the true posterior predictives (Section 2). Hence, for all $x \in \mathcal{X}$ and $D \in \mathcal{D}$, $P_x \pi_C(D)$ is in $\mathcal{P}_{\lambda,G}^1$.

(3) The ideal GNP $\pi_G$ matches the mean vectors and covariance matrices of the true posterior predictives (Section 2). Hence, for all inputs $\mathbf{x}$ and $D \in \mathcal{D}$, $P_{\mathbf{x}} \pi_G(D)$ is in $\mathcal{P}_{\lambda,G}^{|\mathbf{x}|}$; which means that, for all $x_1 \in \mathcal{X}$, inputs $\mathbf{x}_2$, and $D \in \mathcal{D}$, $P_{x_1 \,|\, \mathbf{x}_2} \pi_G(D)$ is in $\mathcal{P}_{\lambda,G}^1$.

In the proof, to apply and use (10), we implicitly use these observations.

**Proposition 2.1** (Advantage of AR CNPs over GNPs). Assume appropriate regularity conditions on $y$. Let $\pi_C$ be the ideal CNP and let $\pi_G$ be the ideal GNP. Then, for all inputs $\mathbf{x}$ and data sets $D \in \mathcal{D}$,

$$\mathrm{KL}(P_{\mathbf{x}} \pi_y(D), \mathrm{AR}_{\mathbf{x}}(\pi_C, D)) \leq \mathrm{KL}(P_{\mathbf{x}} \pi_y(D), P_{\mathbf{x}} \pi_G(D)). \tag{7}$$

*Proof of Proposition 2.1.* Let $\mathbf{x}$ be some inputs and let $D \in \mathcal{D}$ be some data set. We will argue that, for all $n = 1, \dots, |\mathbf{x}|$,

$$\mathrm{KL}(P_{x_n \,|\, \mathbf{x}_{1:(n-1)}} \pi_y(D), P_{x_n} \pi_C(D \oplus (\mathbf{x}_{1:(n-1)}, \mathbf{y}_{1:(n-1)})))$$
$$\leq \mathrm{KL}(P_{x_n \,|\, \mathbf{x}_{1:(n-1)}} \pi_y(D), P_{x_n \,|\, \mathbf{x}_{1:(n-1)}} \pi_G(D)). \tag{11}$$

Assuming this inequality, the result follows directly from the chain rule for the KL divergence in combination with the definition of $\mathrm{AR}_{\mathbf{x}}$ (Procedure 2.1):

$$\mathrm{KL}(P_{\mathbf{x}} \pi_y(D), \mathrm{AR}_{\mathbf{x}}(\pi_C, D))$$

$$= \sum_{n=1}^{|\mathbf{x}|} \mathbb{E}_{\mathbf{y}_{1:(n-1)}}[\mathrm{KL}(P_{x_n \,|\, \mathbf{x}_{1:(n-1)}} \pi_y(D), P_{x_n} \pi_C(D \oplus (\mathbf{x}_{1:(n-1)}, \mathbf{y}_{1:(n-1)})))] \tag{12}$$

$$\leq \sum_{n=1}^{|\mathbf{x}|} \mathbb{E}_{\mathbf{y}_{1:(n-1)}}[\mathrm{KL}(P_{x_n \,|\, \mathbf{x}_{1:(n-1)}} \pi_y(D), P_{x_n \,|\, \mathbf{x}_{1:(n-1)}} \pi_G(D))] \tag{13}$$

$$= \mathrm{KL}(P_{\mathbf{x}} \pi_y(D), P_{\mathbf{x}} \pi_G(D)) \tag{14}$$

where the expectations are over $\mathbf{y}_{1:(n-1)} \sim P_{\mathbf{x}_{1:(n-1)}} \pi_y(D)$. To prove the inequality, note that, conditional on $\mathbf{y}_{1:(n-1)}$, using (10),

$$\arg\min_{\nu \in \mathcal{P}_{\lambda,G}^1} \mathrm{KL}(P_{x_n \,|\, \mathbf{x}_{1:(n-1)}} \pi_y(D), \nu) = \mathcal{N}(P_{x_n \,|\, \mathbf{x}_{1:(n-1)}} \pi_y(D)). \tag{15}$$

By the property of $\pi_C$ that it matches the mean and marginal variance of the true posterior (Section 2),

$$\mathcal{N}(P_{x_n \,|\, \mathbf{x}_{1:(n-1)}} \pi_y(D)) = \mathcal{N}(P_{x_n} \pi_y(D \oplus (\mathbf{x}_{1:(n-1)}, \mathbf{y}_{1:(n-1)}))) \tag{16}$$

$$= P_{x_n} \pi_C(D \oplus (\mathbf{x}_{1:(n-1)}, \mathbf{y}_{1:(n-1)})). \tag{17}$$

Therefore,

$$\arg\min_{\nu \in \mathcal{P}_{\lambda,G}^1} \mathrm{KL}(P_{x_n \,|\, \mathbf{x}_{1:(n-1)}} \pi_y(D), \nu) = P_{x_n} \pi_C(D \oplus (\mathbf{x}_{1:(n-1)}, \mathbf{y}_{1:(n-1)})). \tag{18}$$

Noting that $P_{x_n \,|\, \mathbf{x}_{1:(n-1)}} \pi_G(D) \in \mathcal{P}_{\lambda,G}^1$, we obtain the desired inequality. $\square$

## B  PROOF OF PROPOSITION 2.2

**Proposition 2.2** (Recovery of smooth samples). Let $\mathcal{X} \subseteq \mathbb{R}$ be compact, and let $f$ be a stochastic process with surely continuous sample paths and $\sup_{x \in \mathcal{X}} \|f(x)\|_{L^2} < \infty$. Let $(\varepsilon_n)_{n \geq 0}$ be i.i.d. (potentially non-Gaussian) random variables such that $\mathbb{E}[\varepsilon_0] = 0$ and $\mathbb{V}[\varepsilon_0] < \infty$. Consider any sequence $(x_n)_{n \geq 1} \subseteq \mathcal{X}$, and let $x^* \in \mathcal{X}$ be a limit point of $(x_n)_{n \geq 1}$. If $y(x^*) = f(x^*) + \varepsilon_0$ and $y_n = f(x_n) + \varepsilon_n$ are noisy observations of $f$, then

$$\lim_{n \to \infty} \mathbb{E}[y(x^*) \,|\, y_1, \ldots, y_n] = f(x^*) \quad \text{almost surely.} \tag{8}$$

*Proof of Proposition 2.2.* Consider the increasing filtration $\mathcal{F}_n = \sigma(y_1, \ldots, y_n)$ with limit $\mathcal{F}_\infty = \sigma(\bigcup_{n=1}^{\infty} \mathcal{F}_n)$. Also let $\mathcal{T}_n = \sigma(\varepsilon_{n+1}, \varepsilon_{n+2}, \ldots)$ and consider the tail $\sigma$-algebra $\mathcal{T} = \bigcap_{n=1}^{\infty} \mathcal{T}_n$. Let $(x_{n_i})_{i=1}^{\infty}$ be a subsequence of $(x_n)_{n=1}^{\infty}$ such that $x_{n_i} \to x^*$. Let $g_n = \frac{1}{n} \sum_{i=1}^{n} y_i$. Since $g_n$ is a function of $y_1, \ldots, y_n$, it is $\mathcal{F}_n$–measurable and therefore $\mathcal{F}_\infty$–measurable. Note that

$$g_n = \frac{1}{n} \sum_{i=1}^{n} f(x_{n_i}) + \frac{1}{n} \sum_{i=1}^{n} \varepsilon_i. \tag{19}$$

By sure continuity of $f$, the first term converges to $f(x^*)$ surely. By the strong law of large numbers (Example 5.6.1; Durrett, 2010), the second term converges to zero on a tail event $A \in \mathcal{T}$ of probability one. We conclude that $\mathbb{1}_A f(x^*)$ is $\sigma(\mathcal{F}_\infty, \mathcal{T})$–measurable. Therefore, by almost sure convergence of $L^2$–bounded martingales (Theorem 5.4.5; Durrett, 2010),

$$\lim_{n \to \infty} \mathbb{E}[y(x^*) \,|\, y_1, \ldots, y_n] = \lim_{n \to \infty} \mathbb{E}[f(x^*) \,|\, y_1, \ldots, y_n] \qquad (\mathbb{E}[\varepsilon_0] = 0) \tag{20}$$

$$= \lim_{n \to \infty} \mathbb{E}[f(x^*) \,|\, \mathcal{F}_n] \qquad (\text{definition of } \mathcal{F}_n) \tag{21}$$

$$= \lim_{n \to \infty} \mathbb{E}[f(x^*) \,|\, \mathcal{F}_n, \mathcal{T}] \qquad (\sigma(f(x^*), \mathcal{F}_n) \perp \mathcal{T}) \tag{22}$$

$$= \lim_{n \to \infty} \mathbb{E}[\mathbb{1}_A f(x^*) \,|\, \mathcal{F}_n, \mathcal{T}] \qquad (\mathbb{P}(A) = 1) \tag{23}$$

$$= \mathbb{E}[\mathbb{1}_A f(x^*) \,|\, \mathcal{F}_\infty, \mathcal{T}] \qquad (L^2\text{–mart. convergence}) \tag{24}$$

$$= \mathbb{1}_A f(x^*) \qquad (\mathbb{1}_A f(x^*) \in \sigma(\mathcal{F}_\infty, \mathcal{T})) \tag{25}$$

$$= f(x^*), \qquad (\mathbb{P}(A) = 1) \tag{26}$$

where all equalities hold almost surely. $\qquad \square$

## C    ILLUSTRATION OF THE AR PROCEDURE

Figure 7 depicts the AR sampling procedure (Procedure 2.1) and procedure to produce smooth samples (Proposition 2.2) using the ConvCNP trained on the EQ data process from Section 4.1.

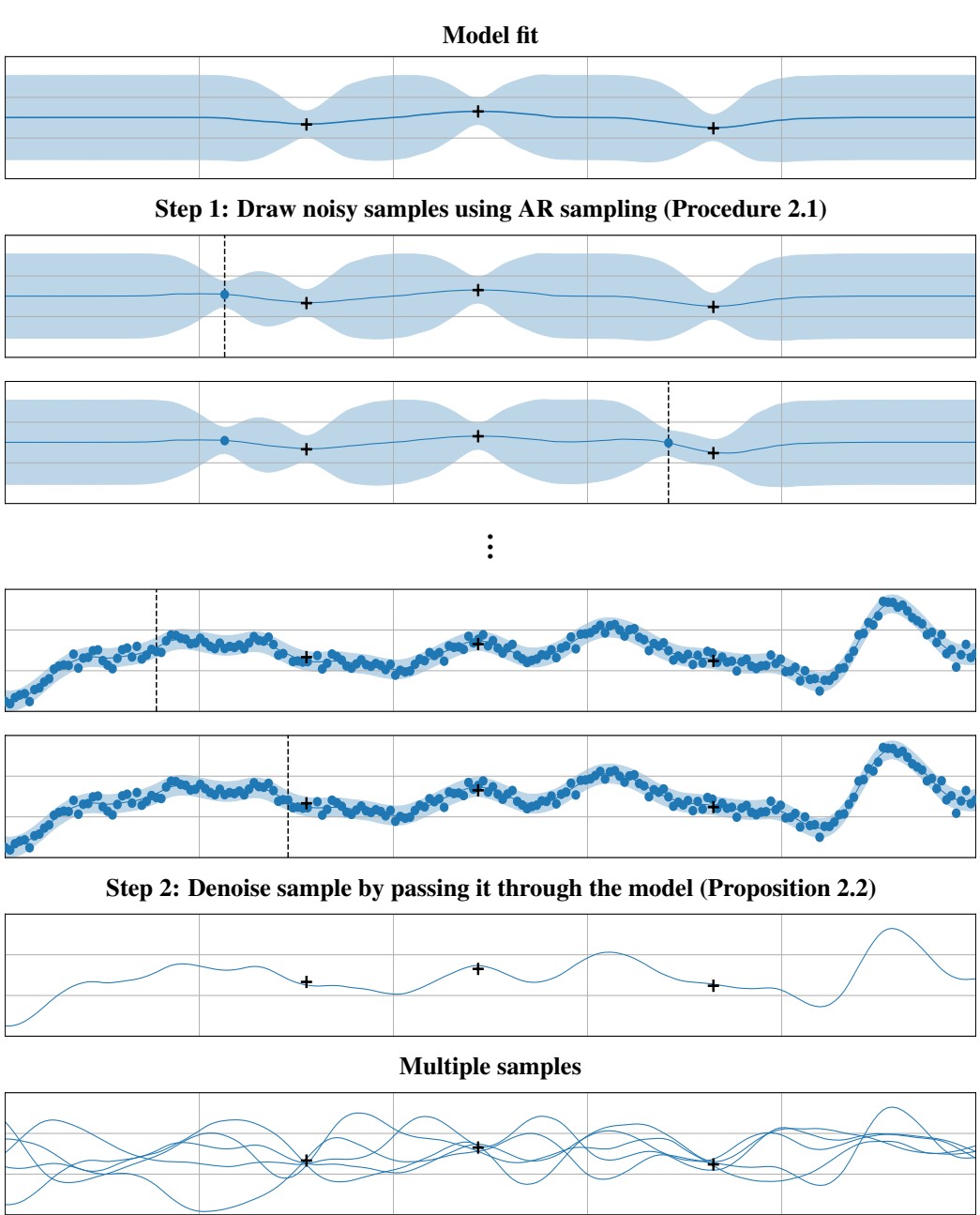

Figure 7: Illustration of the AR procedure with a random AR ordering and the de-noising step (Procedure 2.1 and Proposition 2.2), to produce smooth samples. Given a context set (*black crosses*), we can use the CNP to get marginal predictions at arbitrary input locations (*first figure*). We choose a randomly sampled input location, draw a corresponding output sample from the model's predictive (*blue dots in the second plot*), append this to the context set, and pass the augmented context set through the model again. We repeat this step a number of times (*third and fourth figures*), until all function (epistemic) uncertainty has been removed and all that remains is irreducible noise (aleatoric) uncertainty (*fifth figure*). This procedure yields noisy function samples (*blue dots in the sixth plot*), which we pass one last time through the model to obtained a denoised sample, treating the mean prediction as an approximate noiseless sample (*seventh figure*). Repeating this procedure yields high-quality samples from the model predictive (*eighth figure*).

## D    NUMBER AND ORDER OF TARGET POINTS

When deploying a conditional neural process (CNP) autoregressively (AR; Procedure 2.1), the number and ordering of the target points matters. In this appendix, we describe our observations of the effects of the number and ordering of the target points on the quality of the predictions. In short, our recommendation is to choose a different random ordering for every sample, and to not let the number/density of target points exceed that at training time.

### D.1    EFFECTS OF THE NUMBER OF TARGET POINTS

During the AR sampling procedure, the AR CNP is evaluated at context sets of increasing size. Our experience is that, as long as the sizes of these context sets do not exceed the sizes seen at training time, the predictions should not be significantly affected by changes in the number of target points. However, if the AR sampling procedure evaluates the model at context sets of larger sizes than seen during training time, then that presents the model with an out-of-distribution situation. What happens then comes down to how well the neural networks generalise. Our experience is that the predictions quickly start to break down.

A notable exception of this rule of thumb are convolutional-deep-set–based models, such as the Convolutional Conditional Neural Process (ConvCNP; Gordon et al., 2020). For these models, the *magnitude of the density channel* is what determines whether the models generalises or not. This means that it is not the total number of points that matters, but rather the *density* of the points. Therefore, the AR ConvCNP can be evaluated at arbitrarily many target points, as long as the density of these points does not significantly exceed the density of context points seen at training time. Once the density exceeds the density of the training data, the model is presented with an out-of-distribution situation, and what happens then again comes down to how well neural networks generalise.

Figure Appendix D.1 illustrates this observation. When the density target points does not exceed the training data (50 and 100 points), the predictions look calibrated. However, once the density of target points comes close or exceed the training data (200, 500, and 1000 points), bias starts to creep into the predictions.

Although the number/density of points in the AR sampling procedure should not exceed that at training time, AR CNPs can still produce high-quality samples at arbitrarily many target points by following the trick outlined at the end of the two-step procedure below Proposition 2.2.

### D.2    EFFECTS OF THE ORDERING OF TARGET POINTS

Our experience is that, as long as the number of target points (or density) does not exceed that at training time, the ordering of the target point does not really matter. Appendix D.1 also demonstrates this. When the density of the target points does not exceed the training data (50 and 100 points), sampling randomly or left to right does not really matter. However, once the density of the target points comes close to or exceeds the training data (200, 500, and 1000 points), we observe a difference in performance between sampling randomly and sampling left to right. Across all numbers of target points, a random ordering seems to perform most robustly. Our recommendation is therefore to choose a different random ordering of the target points for every sample.

### D.3    ANALYSIS OF AR CNPS FOR CNPS WITH GAUSSIAN MARGINALS

In this subsection, we argue that, for CNPs with Gaussian marginals, predictions in the first few AR steps might be poor, but predictions in later AR steps tend to be more accurate. Choosing a different random ordering for every sample therefore "averages out" the effects from these first few AR steps.

When evaluating a CNP with Gaussian marginals in AR mode, every conditional prediction in the AR process is Gaussian. Let us consider the process of producing an AR sample. For the first target input $x_1$, we run the CNP forward to obtain a distribution for the corresponding target output $y_1$. In reality, the true posterior most likely is non-Gaussian, which means that the prediction for the first target point may be poor. Nevertheless, we sample this Gaussian, append the sample $(x_1, y_1)$ to the context set, and run the CNP forward again. Because we now feed the earlier sample $y_1$ through the non-linear network, the marginal predictive for the next target output $y_2$ (having integrated out $y_1$)

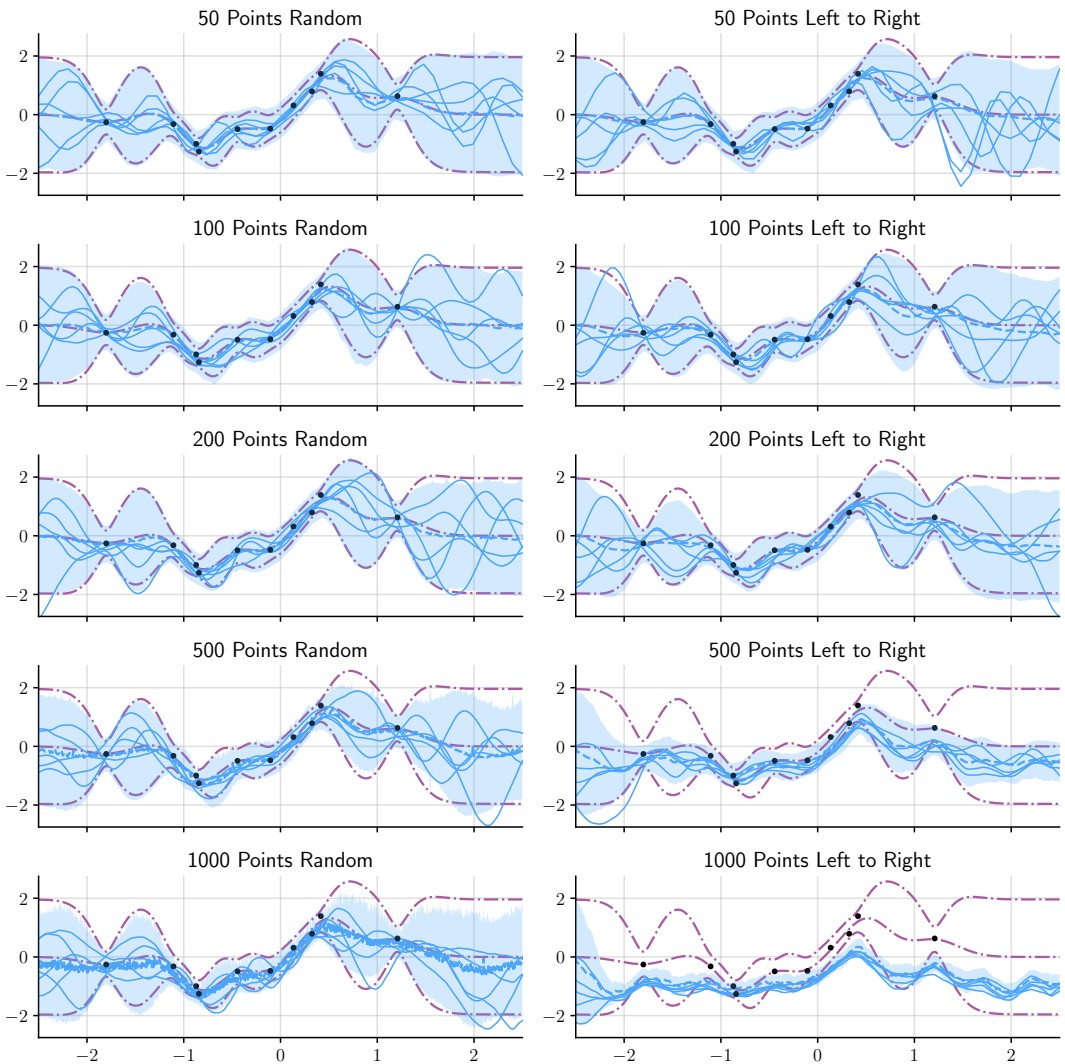

Figure 8: Samples and predictions for an AR ConvCNP with various numbers of target points ordered randomly (*left column*) and ordered left to right (*right column*). When the density of the target points does not exceed the training data (*50 and 100 points*), ordering the target points randomly or left to right does not matter. When the density of the target points comes close to the training data or exceeds it (*200, 500, and 1000 points*), bias creeps into the predictions. The random ordering appears to perform more robustly than left to right. The data is sampled from the EQ data process from the synthetic experiments (Section 4.1), and the trained model is also taken from the synthetic experiments. The predictions by the model are shown in solid blue and the marginals by the ground-truth EQ process are shown in dot-dashed purple.

is non-Gaussian. As we perform more AR steps, the marginal predictions of later points become increasingly non-Gaussian, increasing the model's flexibility.

We see that, for a given ordering of the target inputs, the prediction for the first target input is likely poor (because it is Gaussian), and (in the best case) the predictions become more and more accurate as we take more AR steps (because they become more and more non-Gaussian). This is exactly what is happening in Figure 3: the left prediction is Gaussian and therefore a poor approximation, and, as we go to the right and take more and more AR steps, the prediction becomes more and more non-Gaussian and therefore more accurate. If we were to feed the target inputs in right to left, then the same phenomenon would happen. The right prediction would be a Gaussian and a very poor

approximation, and, as we go to the left and take more AR steps, the prediction would become more non-Gaussian and therefore more accurate.

More generally, for a given ordering of the target points, the ordering will produce high quality predictions if the conditional distributions of the AR factorisation match the corresponding conditional distributions of the true posterior. Since the conditionals of the AR CNP are typically Gaussian by design, this means that the ordering is "good" if the corresponding conditionals of the true posterior are close to Gaussian.

So when is a conditional of the posterior close to Gaussian? Let us assume that the true underlying process is a sum of a non-Gaussian process (constituting epistemic uncertainty) and independent Gaussian noise (constituting aleatoric uncertainty). Generally, a conditional will have both epistemic and aleatoric uncertainty, so a Gaussian will be a bad fit. *However*, as we condition the conditionals of the true generative process on more and more data, the underlying function will be pinned down more and more accurately, meaning that the conditional will consist mostly of aleatoric uncertainty, which is Gaussian. Therefore, as we condition on more and more data, we expect the conditionals to become more and more Gaussian. This again suggests that the samples in the first few AR steps might be a poor fit (because the corresponding conditionals of the true posterior are not yet Gaussian), but that samples in later AR steps should be a better fit (because the corresponding conditionals are then close to Gaussian).

To summarise, an ordering of the target points is "good" if the corresponding conditionals of the true posterior are also close to Gaussian. Under the assumption that the ground-truth process is a non-Gaussian process with additive Gaussian noise, conditionals tend to be close to Gaussian if they are conditioned on many data points. As a consequence, the earlier conditionals in the AR factorisation tend to be poor fits to the ground-truth posterior, whereas later conditionals tend to produce better fits. Choosing a different random ordering for every sample therefore "averages out" the effects from the first few AR steps.

### D.4    EFFECT OF THE RANDOM ORDERING ON THE SPREAD OF THE LOG-LIKELIHOOD

We have thus far argued for the benefit of using random ordering in AR, due to the robustness it provides. However, one issue with random orderings is that, since different random orderings do not in general give rise to the same predictive distribution, we may obtain different predictive log-likelihoods in practice, depending on the exact random ordering that we sample. Ideally, we would like not only the mean predictive log-likelihood (averaged out over orderings) to be high, but also the standard deviation of the log-likelihood (due to, again, different random orderings) to be small. In other words, we would like the model to perform well regardless of the random ordering which we happen to sample.

At this point, note that if the true underlying process is Gaussian, then a sufficiently well-trained AR CNP with Gaussian marginals would have a small such spread in the log-likelihood, because all conditional predictions of the model will be close to the ground truth conditional predictions. Consequently the order with which we make predictions will have a small effect on the log-likelihood, resulting in a small spread of predictive log-likelihood values. Consider for example the case where the conditionals of the CNP exactly match the conditionals of the true process. In this case, there will be zero variance in the predictive log-likelihood of the process under different orderings. However, the situation is different when the ground truth is non-Gaussian. In this case, as we explained in the previous section, the conditionals of the first few target points may be highly non-Gaussian under the true process, while those of the AR CNP are Gaussian. In this case, we may get different log-likelihoods depending on the random order that we happen to sample.

Figure 9 provides a quantitative illustration of the above point. In this figure, we show the standard deviation in the per datapoint predictive log-likelihood of an AR CNP (due to different random orderings) on two variants of a task with sawtooth data. On the first variant, we always pass an empty context set to the model (blue), and on the other task, we pass non-empty context sets with randomly sampled number of context points, uniformly distributed between 0 and 100 (red). We observe that for empty contexts (blue), we get a relatively large standard deviation in predictive log-likelihood for the first few target points. This likely happens because, initially, the model may randomly pick a target input where the conditional of the true process is highly non-Gaussian (making a poor prediction), or it might pick a target input where the true conditional is Gaussian (making a good prediction). This

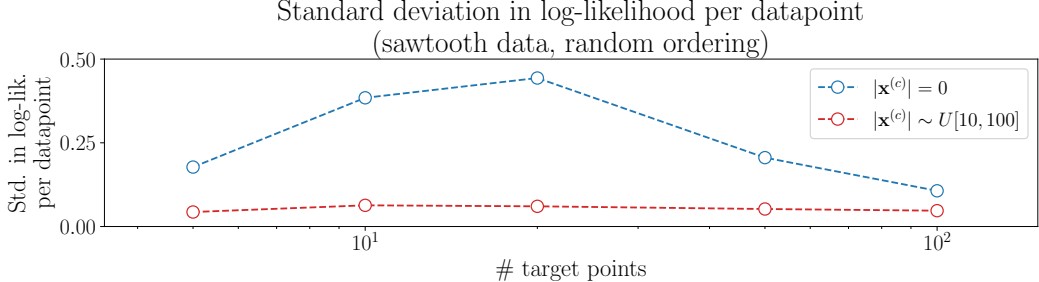

Figure 9: Plot of the standard deviation, due to different random orderings, of the per-data-point predictive log-likelihood (in nats) of an AR ConvCNP on one-dimensional sawtooth data, as a function of the number of target set size. For each point in the plot, we have used $2^{10}$ randomly sampled and fixed tasks, on each of which we apply the AR ConvCNP with 100 different randomly sampled orderings.

results in a larger variance in performance for the first few target points. However, as more target points are introduced, the standard deviation shrinks. This is because the conditionals of the true process become increasingly Gaussian, which means that no matter which target input is picked next, the model will approximate the true conditional accurately using a Gaussian, thereby reducing the impact of the ordering of subsequent points on the variance of the log likelihood. Further, introducing a relatively modest number of initial context points (red) in a second variant of the task, substantially reduces the spread in the predictive log-likelihoods. This is again because conditioning on a context set means that the conditionals of the true process are better approximated by Gaussians, reducing the impact that different random orderings have on the spread of the log-likelihood. In practice, in our experiments, we have found the variance in the log-likelihood to be near-zero for Gaussian or Gaussian-like ground truth processes, and larger, but acceptable, for non-Gaussian tasks.

# E   DETAILS FOR FIGURE 3

The generative process visualised in the top panel of figure 3 is defined by the following mixture distribution:

$$p_{\text{true}}(y \mid x) = a_1 \mathcal{N}(f_1(x), 1) + a_2 \mathcal{N}(f_2(x), 1) + a_3 \mathcal{N}(f_3(x), 1). \tag{27}$$

Given this mixture distribution, the (Gaussian) ideal CNP can be computed in closed form by computing the first two moments of $p_{\text{true}}$:

$$p_{\text{CNP}}(y \mid x) = \mathcal{N}(\mu(x), \sigma^2(x)) \tag{28}$$

where

$$\mu(x) = \sum_{i=1}^{3} a_i f_i(x), \tag{29}$$

$$\sigma^2(x) = \sum_{i=1}^{3} a_i \left(1 + f_i(x)^2\right) - \left(\sum_{i=1}^{3} a_i f_i(x)\right)^2. \tag{30}$$

The updated mixture weights for the posterior distribution $p_{\text{true}}(y \mid x, D^{(c)})$ given a context set $D^{(c)}$ can be computed via Bayes rule and $p_{\text{CNP}}(y \mid x, D^{(c)})$ can be computed given the updated mixture weights. Note that in Figure 3 the prior mixture weights are $a_1 = a_3 = 0.25$ and $a_2 = 0.5$, means are given by

$$f_1(x) = x^2 + 1, \tag{31}$$
$$f_2(x) = x, \tag{32}$$
$$f_3(x) = -x, \tag{33}$$

and the target locations are $x = 1, 2, 4$, and $6$. The bottom four panels of Figure 3 show kernel density estimates (Gaussian kernel) of $30\,000$ samples drawn from the generative distribution $p_{\text{true}}(y_1, y_2, y_4, y_6)$, the ideal CNP $p_{\text{CNP}}(y_1, y_2, y_4, y_6)$, and the ideal CNP applied in AR mode from left to right:

$$p_{\text{AR CNP}}(y_1, y_2, y_4, y_6) = p_{\text{CNP}}(y_1) p_{\text{CNP}}(y_2 \mid y_1) p_{\text{CNP}}(y_4 \mid y_1, y_2) p_{\text{CNP}}(y_6 \mid y_1, y_2, y_4). \tag{34}$$

# F   DESCRIPTION OF MODELS

The architectures follow the descriptions from the respective papers they are introduced. Although these descriptions are for one-dimensional inputs and outputs, the architectures are readily generalised to multidimensional inputs and outputs; we will explicitly mention wherever that generalisation requires extra care. All architectures use ReLU activation functions. All GNPs, in addition to a covariance matrix over the target points, also output heterogeneous observation noise along the marginal means; the total covariance over the target points is thus the sum of the covariance by the model and a diagonal matrix formed from these observation noises.

**Conditional neural process (CNP; Garnelo et al., 2018a).** Set the dimensionality of the encoding to $K = 256$. Parametrise the encoder with a three-hidden-layer multi-layer perceptron (MLP) of width 256; and parametrise the decoder with a six-hidden-layer MLP of width 256. For multidimensional outputs, let the decoder have width 512. For multidimensional outputs where outputs can have context points at different inputs, produce a separate encoding for every output and concatenate these into one big encoding. These encoders may or may not share parameters. In our experiments, for two-dimensional outputs, parametrise separate encoders; for higher-dimensional outputs, apply the same encoder.

**Gaussian neural process (GNP; Markou et al., 2022).** Use the same choices for $K$, the encoder, and the decoder as the CNP. Set the rank of the kernel map to $R = 64$. As mentioned in the introduction, let the decoder produce one extra dimension which forms heterogeneous observation noise. For multidimensional outputs, the same caveats as for the CNP apply.

**Latent neural process (LNP; Garnelo et al., 2018b).** The LNP builds off the CNP. Call the existing encoder the *deterministic encoder*. The NP adds one more encoder called the *stochastic encoder*. The stochastic encoder mimics the deterministic encoder, but outputs a $K$-dimensional vector of means and a $K$-dimensional vector of marginal variances. These are used to sample a $K$-dimensional Gaussian latent variable (the *stochastic encoding*). The decoder now additionally takes in the stochastic encoding. For multidimensional outputs, the same caveats as for the CNP apply.

**Attentive conditional neural process (ACNP; Kim et al., 2019).** The ACNP builds off the CNP. It replaces the deterministic encoder $\mathrm{enc}_\theta \colon \mathcal{D} \to \mathbb{R}^K$ with an eight-head attentive encoder $\mathrm{enc}_\theta^{(\mathrm{att})} \colon \mathcal{D} \times \mathcal{X} \to \mathbb{R}^K$ (Vaswani et al., 2017). Unlike the original deterministic encoder $\mathrm{enc}_\theta$, the new attentive encoder $\mathrm{enc}_\theta^{(\mathrm{att})}$ also takes in the target input. Let $D^{(\mathrm{c})} = (\mathbf{x}^{(\mathrm{c})}, \mathbf{y}^{(\mathrm{c})}) \in \mathcal{D}$ be a context set of size $N$ and let $x^{(\mathrm{t})} \in \mathcal{X}$ be a target input. We now descibe the computation of $\mathrm{enc}_\theta^{(\mathrm{att})}(D^{(\mathrm{c})}, x^{(\mathrm{t})})$. Parametrise $\phi_x \colon \mathcal{X} \to (\mathbb{R}^{32})^8$ and $\phi_{xy} \colon \mathcal{X} \times \mathcal{Y} \to (\mathbb{R}^{32})^8$ both with three-hidden-layer MLPs of width 256. Compute

$$\text{the } keys\colon \quad (\mathbf{k}_{h,n})_{h=1}^8 = \phi_x(x_n^{(\mathrm{c})}) \qquad \text{for } n = 1, \ldots, N, \tag{35}$$

$$\text{the } values\colon \quad (\mathbf{v}_{h,n})_{h=1}^8 = \phi_{xy}(x_n^{(\mathrm{c})}, y_n^{(\mathrm{c})}) \quad \text{for } n = 1, \ldots, N, \tag{36}$$

$$\text{the } query\colon \quad (\mathbf{q}_h)_{h=1}^8 = \phi_x(x^{(\mathrm{t})}). \tag{37}$$

Then compute

$$\mathbf{v}_h^{(\mathrm{q})} = \sum_{n=1}^N \frac{e^{\langle \mathbf{q}_h, \mathbf{k}_{h,n} \rangle}}{\sum_{n'=1}^N e^{\langle \mathbf{q}_h, \mathbf{k}_{h,n'} \rangle}} \mathbf{v}_{h,n} \in \mathbb{R}^{256} \tag{38}$$

Concatenate $\mathbf{v}^{(\mathrm{q})} = (\mathbf{v}_1^{(\mathrm{q})}, \ldots, \mathbf{v}_8^{(\mathrm{q})}) \in \mathbb{R}^{256}$ and $\mathbf{q} = (\mathbf{q}_1, \ldots, \mathbf{q}_8) \in \mathbb{R}^{256}$. Let $\mathbf{L} \colon \mathbb{R}^{256} \to \mathbb{R}^{256}$ be a linear layer; let $\phi^{(\mathrm{res})} \colon \mathbb{R}^{256} \to \mathbb{R}^{256}$ be a one-hidden-layer MLP of width 256; and let $\mathrm{norm}_1$ and $\mathrm{norm}_2$ be two layer normalisation layers with learned pointwise transformations (Ba et al., 2016). Then

$$\mathrm{enc}_\theta^{(\mathrm{att})}(D^{(\mathrm{c})}, x^{(\mathrm{t})}) = \mathrm{norm}_2(\mathbf{z} + \phi^{(\mathrm{res})}(\mathbf{z})) \quad \text{where} \quad \mathbf{z} = \mathrm{norm}_1(\mathbf{v}^{(\mathrm{q})} + \mathbf{L}\mathbf{q}). \tag{39}$$

For multidimensional outputs, the same caveats as for the CNP apply.

**Attentive Gaussian neural process (AGNP).** The AGNP build off the GNP. It replaces the deterministic encoder with the same eight-head attentive deterministic encoder of the ACNP.

**Attentive neural process (ALNP; Kim et al., 2019).** The ALNP build off the LNP. It replaces the deterministic encoder with the same eight-head attentive deterministic encoder of the ACNP.

**Convolutional Conditional Neural Process (ConvCNP; Gordon et al., 2020).** Set the discretisation to an evenly spaced grid at a certain density (the *points per unit*) spanning a bit more (the *margin*) than the most extremal context and target inputs. The points per unit and margin are specified separately for every experiment. Initialise the length scales of all Gaussian kernels to twice the interpoint spacing of the discretisation. Divide the data channel by the density channel. Parametrise $\text{dec}_\theta$ with a U-Net (Ronneberger et al., 2015). Before the U-turn, let the U-Net have six convolutional layers with kernel size five, stride two, and 64 output channels; and six more such layers, but using transposed convolutions, after the U-turn. The layers after the U-turn additionally take in the outputs of the layers before the U-turn in reversed order; this is the U-net structure (Figure 1; Ronneberger et al., 2015). For multidimensional outputs where outputs can have context points at different inputs, produce a separate data and density channel for every output and concatenate these into one big encoding; use separate length scales for every application of $\text{enc}_\theta$.

**Convolutional Gaussian neural process (ConvGNP; Markou et al., 2022).** Use the same choices for the discretisation, length scales, and CNN architecture as for the ConvCNP. Set the rank of the kernel map to $R = 64$. As mentioned in the introduction, let the decoder produce one extra channel which forms heterogeneous observation noise. For multidimensional outputs, the same caveat as for the ConvCNP applies.

**Fully convolutional Gaussian neural process (FullConvGNP; Bruinsma et al., 2021).** For the mean architecture and the kernel architecture, use the same choices for the discretisation, length scales, and CNN architecture as for the ConvCNP. Implement the source channel with the identity matrix and apply the matrix transform $\mathbf{Z} \mapsto \mathbf{Z}\mathbf{Z}^\mathsf{T}$ to ensure positive definiteness. Let the decoder produce one extra channel which forms heterogeneous observation noise. For multidimensional outputs, in addition to the caveat for the ConvCNP, two additional caveats apply. First, for $D_\text{o}$-dimensional outputs, let the decoder produce $D_\text{o}^2$ channels rather than just one. These channels should be interpreted as all covariance and cross-covariance matrices between all outputs. Second, when applying the matrix transform $\mathbf{Z} \mapsto \mathbf{Z}\mathbf{Z}^\mathsf{T}$, these channels should first be assembled into one total covariance matrix.

**Convolutional latent neural process (ConvLNP; Foong et al., 2020).** The ConvLNP builds off the ConvCNP. The ConvLNP replaces the CNN architecture by two copies of this architecture placed in sequence. In between the two architectures, there is a sampling step: the first architecture outputs 32 channels, comprising 16 means and 16 marginal variances, which are used to sample a 16-dimensional Gaussian latent variable; and the second architecture then takes in this sample.

**Autoregressive Conditional Neural Processes (AR CNPs).** The AR CNP, AR ACNP, and AR ConvCNP use the architectures described above. Rolling out an AR CNP according to Procedure 2.1 requires an ordering of the target points. In all experiments, we choose a random ordering of the target points.

## G    TRAINING, CROSS-VALIDATION, AND EVALUATION PROTOCOLS

The following description applies to the synthetic experiments (Section 4.1), the predator–prey experiments (Section 4.2), the EEG experiments (Section 4.3), and the environmental downscaling experiments (Section 4.4). For the environmental data assimilation experiments, a different protocol was used; we refer the reader to Appendix K for full details of the environmental data assimilation experiments.

A *task* consists of a context set and target set. How precisely the context and target sets are generated is specific to an experiment. To train a model, we consider batches of 16 tasks at a time, compute an objective function value, and update the model parameters using ADAM (Kingma & Ba, 2015). The learning rate is specified separately for every experiment. We define an epoch to consist of $2^{14} \approx$ 16 k tasks. We typically train a model for between 100 and 1000 epochs.

For an experiment, we split up the meta–data set into a *training set*, a *cross-validation set*, and an *evaluation set*. The model is trained on the training set. During training, after every epoch, the model is cross-validated on the cross-validation set. Cross-validation uses $2^{12}$ fixed tasks. These $2^{12}$ are fixed, which means that cross-validation always happens with exactly the same data. The cross-validation objective is a confidence bound computed from the model objective. Suppose that model objective over all $2^{12}$ cross-validation tasks has empirical mean $\hat{\mu}$ and empirical variance $\hat{\sigma}^2$. If a higher model objective is better, then the cross-validation objective is given by $\hat{\mu} - 1.96 \cdot \hat{\sigma}/\sqrt{2^{12}}$. The model with the best cross-validation objective is selected and used for evaluation. Evaluation is performed with the evaluation set and also uses $2^{12}$ tasks.

Conditional neural processes and Gaussian neural processes are trained, cross-validated, and evaluated with the neural process ELBO objective proposed by Garnelo et al. (2018a). We normalise the terms in the neural process objective by the target set sizes. Latent-variable neural processes (LNPs) are trained, cross-validated, and evaluated with the ELBO objective proposed by Garnelo et al. (2018b) using five samples, also normalised by the target set size. When training LNPs with the ELBO objective, but not when cross-validating and evaluating, the context set is subsumed in the target set. Additionally, LNPs are trained, cross-validated, and evaluated with the ML objective proposed by Foong et al. (2020), again normalised by the target set size. When training and cross-validating LNPs with the ML objective, we use twenty samples; and when evaluating, we use 512 samples. For completeness, LNPs trained with the ELBO objective are also evaluated with the ML objective using 512 samples.

To stabilise the numerics for GNPs, we increase the regularisation of covariance matrices for one epoch. To encourage LNPs to fit, we fix the variance of the observation noise of the decoder to $10^{-4}$ for the first three epochs.

# H  DETAILS OF SYNTHETIC EXPERIMENTS

## H.1  DESCRIPTION OF EXPERIMENTS

We synthetically generate data sets by randomly sampling from five different choices for the ground-truth stochastic process $f$. Let the inputs be $d_x$-dimensional. Then define the following stochastic processes:

EQ: a Gaussian process with an exponentiated quadratic (EQ) kernel:

$$f \sim \mathcal{GP}(0, \exp(-\tfrac{1}{2\ell^2}\|\mathbf{x}-\mathbf{x}'\|_2^2)) \tag{40}$$

where $\ell > 0$ is a length scale;

Matérn–$\tfrac{5}{2}$: a Gaussian process with a Matérn–$\tfrac{5}{2}$ kernel:

$$f \sim \mathcal{GP}(0, k(\tfrac{1}{\ell}\|\mathbf{x}-\mathbf{x}'\|_2)) \tag{41}$$

where $k(r) = (1 + \sqrt{5}r + \tfrac{5}{3}r^2)e^{-r}$ and $\ell > 0$ is a length scale;

weakly periodic: a Gaussian process with a weakly periodic kernel:

$$f \sim \mathcal{GP}(0, \exp(-\tfrac{1}{2\ell_{\mathrm{d}}^2}\|\mathbf{x}-\mathbf{x}'\|_2^2 - \tfrac{2}{\ell_{\mathrm{p}}^2}\|\sin(\tfrac{\pi}{p}(\mathbf{x}-\mathbf{x}'))\|_2^2)) \tag{42}$$

where $\ell_{\mathrm{d}} > 0$ is a length scale specifying how quickly the periodic pattern changes, $\ell_{\mathrm{p}} > 0$ a length scale of the periodic pattern, and $p > 0$ the period; and where the application of $\sin$ is elementwise;

sawtooth: a sawtooth process with a random frequency, direction, and phase:

$$f = \omega\langle\mathbf{x}, \mathbf{u}\rangle_2 + \phi \mod 1 \tag{43}$$

where $\omega \sim \mathrm{Unif}(\Omega)$ is the frequency of the sawtooth wave, $\mathbf{u} \sim \mathrm{Unif}(\{\mathbf{x} \in \mathbb{R}^{d_x} : \|\mathbf{x}\|_2 = 1\})$ the direction, and $\phi \sim \mathrm{Unif}([0,1])$ the phase;

mixture: with equal probability, sample $f$ from the EQ process, Matérn–$\tfrac{5}{2}$ process, weakly periodic process, or sawtooth process.

We will call these stochastic processes the *data processes*. The data processes are stochastic processes with $d_x$-dimensional inputs and one-dimensional outputs. We will turn them into processes with $d_y$-dimensional outputs according to the following procedure: sample from the one-dimensional-output prior $d_y$ times; and, for these $d_y$ samples, take $d_y$ different linear combinations.

We choose the parameters of the data processes based on the input dimensionality $d_x$:

$$\ell = c \cdot \tfrac{1}{4}, \qquad \ell_{\mathrm{d}} = c \cdot \tfrac{1}{2}, \qquad \ell_{\mathrm{s}} = c, \qquad p = c \cdot \tfrac{1}{4}, \qquad \Omega = [c^{-1} \cdot 2, c^{-1} \cdot 4] \tag{44}$$

with $c = \sqrt{d_x}$. Scaling with the input dimensionality aims to roughly ensure that data with one-dimensional inputs and data with two-dimensional inputs are equally difficult. Figure 10 illustrates the sawtooth data process in all four configurations.

We will construct data sets by sampling inputs uniformly at random from $\mathcal{X} = [-2, 2]^{d_x}$ and then sampling outputs from one of the data processes. We will colloquially call $\mathcal{X}$ the *training range*. For the EQ, Matérn–$\tfrac{5}{2}$, and weakly periodic process, but not for the sawtooth process[6], we also add independent Gaussian noise with variance $0.05$. The numbers of context and target points are as follows. For the EQ, Matérn–$\tfrac{5}{2}$, and weakly periodic process, the number of context points is chosen uniformly at random from $\{0, \ldots, 30 \cdot d_x\}$ and the number of targets points is fixed to $50 \cdot d_x$. For the sawtooth and mixture process, the number of context points is chosen uniformly at random from $\{0, \ldots, 30\}$ if $d_x = 1$ and $\{0, \ldots, 75 \cdot d_x\}$ otherwise; and the number of targets points is fixed to $100 \cdot d_x$. In the case of a multidimensional-output data process, we separately sample the number and positions of the context and target inputs for every output dimension.

For every data process and each of the four configurations, we evaluate every model in three different ways. First, we evaluate the model on data generated exactly like the training data. This task is called *interpolation* and abbreviated "int." in the tables of results. The interpolation task measures how well a model fits the data and is the primary measure of performance. Second, we evaluate the model on data with inputs sampled from $[2, 6]^{d_x}$. This task is called *out-of-input-distribution (OOID)*

---

[6] The sawtooth process is already challenging enough.

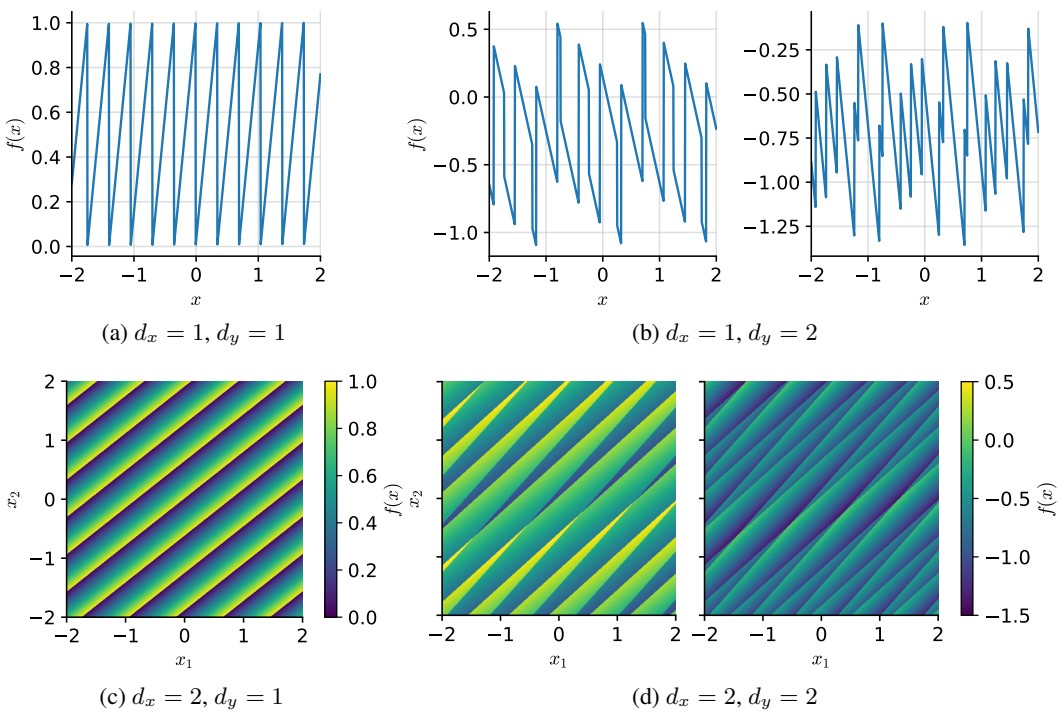

Figure 10: Samples from the sawtooth data process with one and two-dimensional inputs ($d_x = 1$ and $d_x = 2$) and one and two-dimensional outputs ($d_y = 1$ and $d_y = 2$)

*interpolation* and abbreviated "OOID" in the tables of results. OOID interpolation measures how well a model generalises to data sampled from other regions of the input space. Third, we evaluate the model on data with context inputs sampled from $[-2, 2]^{d_x}$ and target inputs sampled from $[2, 6]^{d_x}$. This task is called *extrapolation* and abbreviated "ext." in the tables of results. The extrapolation task measures how well predictions based on data in the training range generalise to other regions of the input space.

For this experiment, the learning rate is $3 \cdot 10^{-4}$, the margin is $0.1$, and the points per unit is $64$. We trained the models for 100 epochs. Due to an error in the cross-validation procedure, we did not use cross-validation, but used the model at epoch 100.

For the kernel architecture of the FullConvGNP, we reduce the points per unit and the number of channels in the U-Net by a factor two. For the ConvLNP with two-dimensional inputs, we reduce the number of outputs channels in the U-Net by a factor $\sqrt{2}$; and, for training and cross-validation, we reduce the number of samples of the ELBO objective to one and the number of samples for the ML objective to five.

## H.2 MULTI-MODALITY OF PREDICTIONS BY AR CONVCNP

Figure 11 demonstrates multi-modality of predictions by the AR ConvCNP trained on the sawtooth process. Note that the prediction is bimodal for one and two observations, and collapses to a single mode upon observing the third observation.

## H.3 FULL RESULTS

We the show the full results for all data sets and tasks in Tables 7 to 18. The AR ConvCNP consistently shows very strong performance compared to other NP models. Note that the FullConvGNP takes much longer to train than the ConvCNP (Figure 2), and cannot be applied to tasks with 2-dimensional input spaces.

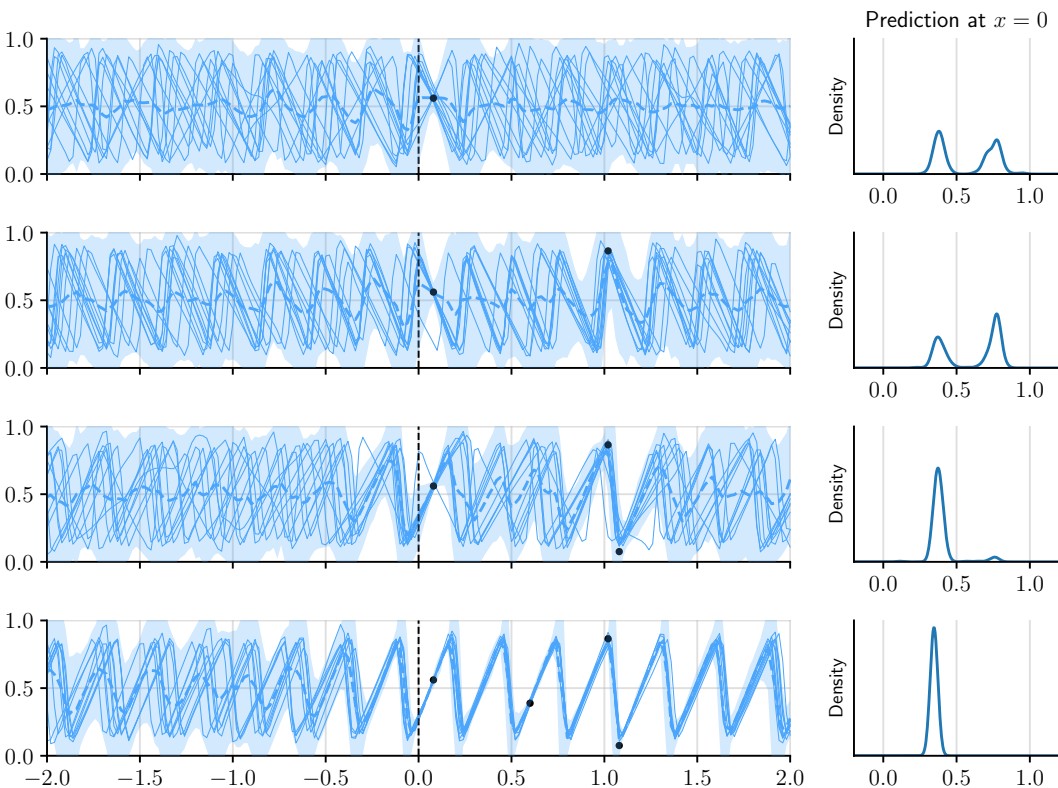

Figure 11: Multi-modality of predictions by the AR ConvCNP. Shows four observations sampled from the sawtooth process. In the four rows, these four observations are revealed one data point at a time. Every row also shows a kernel density estimate of the prediction at $x = 0$. Filled regions are central 95%-credible regions.

Table 7: For the Gaussian experiments, average Kullback–Leibler divergences of the posterior prediction map $\pi_y$ with respect to the model normalised by the number of target points. Shows for one-dimensional inputs (1D; $d_x = 1$) and two-dimensional inputs (2D; $d_y = 2$) the performance for interpolation within the range $[-2, 2]^{d_x}$ where the models were trained ("Int."); interpolation within the range $[2, 6]^{d_x}$ which the models have never seen before ("OOID"); and extrapolation from the range $[-2, 2]^{d_x}$ to the range $[2, 6]^{d_x}$ ("Ext."). Models are ordered by interpolation performance for one-dimensional inputs. The latent variable models are trained and evaluated with the ELBO objective (E); trained and evaluated with the ML objective (M); and trained with the ELBO objective and evaluated with the ML objective (E–M). Diagonal GP refers to predictions by the ground-truth Gaussian processes without correlations. Trivial refers to predicting the empirical means and standard deviation of the test data. Errors indicate the central 95%-confidence interval. Numbers which are significantly best ($p < 0.05$) are boldfaced. Numbers which are very large are marked as failed with "F". Numbers which are missing could not be run.

| Model | Int. (1D) | OOID (1D) | Ext. (1D) | Int. (2D) | OOID (2D) | Ext. (2D) |
|---|---|---|---|---|---|---|
| FullConvGNP | **0.01** ±0.00 | **0.01** ±0.00 | **0.00** ±0.00 | | | |
| ConvCNP (AR) | 0.03 ±0.00 | 0.03 ±0.00 | 0.02 ±0.00 | **0.03** ±0.00 | **0.03** ±0.00 | **0.02** ±0.00 |
| ConvGNP | 0.04 ±0.00 | 0.04 ±0.00 | 1.75 ±0.12 | 0.12 ±0.00 | 0.12 ±0.00 | 0.71 ±0.03 |
| AGNP | 0.10 ±0.00 | 4.34 ±0.17 | 5.45 ±0.23 | 0.17 ±0.00 | 0.62 ±0.01 | 0.39 ±0.01 |
| ConvLNP (E) | 0.19 ±0.01 | 0.19 ±0.01 | 0.29 ±0.03 | 0.39 ±0.01 | 0.39 ±0.01 | 0.36 ±0.01 |
| ACNP (AR) | 0.24 ±0.01 | 1.08 ±0.02 | 0.86 ±0.01 | 0.13 ±0.00 | 0.57 ±0.01 | 0.40 ±0.01 |
| GNP | 0.25 ±0.01 | F | F | 0.25 ±0.01 | 0.75 ±0.01 | 0.57 ±0.00 |
| ConvLNP (M) | 0.31 ±0.01 | 0.31 ±0.01 | 0.64 ±0.01 | 0.28 ±0.01 | 0.28 ±0.01 | 0.36 ±0.01 |
| *Diagonal GP* | 0.42 ±0.02 | 0.42 ±0.02 | 0.84 ±0.01 | 0.29 ±0.01 | 0.29 ±0.01 | 0.40 ±0.01 |
| ALNP (M) | 0.43 ±0.01 | 1.03 ±0.02 | 0.78 ±0.01 | 0.31 ±0.01 | 0.55 ±0.01 | 0.39 ±0.01 |
| ConvCNP | 0.43 ±0.02 | 0.43 ±0.02 | 0.84 ±0.01 | 0.30 ±0.01 | 0.30 ±0.01 | 0.40 ±0.01 |
| CNP (AR) | 0.46 ±0.01 | F | F | 0.36 ±0.01 | F | F |
| LNP (E) | 0.51 ±0.01 | F | 4.34 ±0.76 | 0.40 ±0.01 | 3.03 ±1.68 | 0.60 ±0.01 |
| LNP (E–M) | 0.52 ±0.02 | F | 2.39 ±0.33 | 0.39 ±0.01 | 2.35 ±1.06 | 0.57 ±0.01 |
| ALNP (E–M) | 0.53 ±0.01 | 1.12 ±0.03 | 0.85 ±0.02 | 0.42 ±0.01 | 0.78 ±1.72 | 0.41 ±0.01 |
| ACNP | 0.54 ±0.02 | 1.11 ±0.02 | 0.84 ±0.01 | 0.34 ±0.01 | 0.57 ±0.01 | 0.40 ±0.01 |
| ALNP (E) | 0.54 ±0.01 | 1.60 ±0.06 | 1.25 ±0.03 | 0.43 ±0.01 | 1.06 ±3.04 | 0.42 ±0.01 |
| LNP (M) | 0.59 ±0.01 | F | 1.13 ±0.01 | 0.41 ±0.01 | 0.88 ±0.03 | 0.52 ±0.01 |
| CNP | 0.63 ±0.01 | F | 1.08 ±0.02 | 0.43 ±0.01 | 1.16 ±0.45 | 0.52 ±0.01 |
| *Trivial* | 1.08 ±0.01 | 1.08 ±0.01 | 0.85 ±0.01 | 0.57 ±0.01 | 0.57 ±0.01 | 0.40 ±0.00 |
| ConvLNP (E–M) | 2.01 ±0.11 | 2.01 ±0.11 | 5.95 ±0.16 | 0.44 ±0.01 | 0.44 ±0.01 | 0.47 ±0.01 |

Table 8: For the non-Gaussian experiments, average log-likelihoods normalised by the number of target points. Shows for one-dimensional inputs (1D; $d_x = 1$) and two-dimensional inputs (2D; $d_y = 2$) the performance for interpolation within the range $[-2, 2]^{d_x}$ where the models were trained ("Int."); interpolation within the range $[2, 6]^{d_x}$ which the models have never seen before ("OOID"); and extrapolation from the range $[-2, 2]^{d_x}$ to the range $[2, 6]^{d_x}$ ("Ext."). Models are ordered by interpolation performance for one-dimensional inputs. The latent variable models are trained and evaluated with the ELBO objective (E); trained and evaluated with the ML objective (M); and trained with the ELBO objective and evaluated with the ML objective (E–M). Trivial refers to predicting the empirical means and standard deviation of the test data. Errors indicate the central 95%-confidence interval. Numbers which are significantly best ($p < 0.05$) are boldfaced. Numbers which are very large are marked as failed with "F". Numbers which are missing could not be run.

| Model | Int. (1D) | OOID (1D) | Ext. (1D) | Int. (2D) | OOID (2D) | Ext. (2D) |
|---|---|---|---|---|---|---|
| ConvCNP (AR) | $\mathbf{1.52}_{\pm0.04}$ | $\mathbf{1.53}_{\pm0.04}$ | $\mathbf{1.32}_{\pm0.04}$ | $\mathbf{0.56}_{\pm0.03}$ | $\mathbf{0.56}_{\pm0.03}$ | $\mathbf{0.29}_{\pm0.03}$ |
| ConvLNP (E) | $1.40_{\pm0.05}$ | $1.40_{\pm0.05}$ | $0.82_{\pm0.05}$ | $0.06_{\pm0.03}$ | $0.06_{\pm0.03}$ | $-0.62_{\pm0.04}$ |
| ConvLNP (M) | $1.08_{\pm0.06}$ | $1.08_{\pm0.06}$ | $-0.36_{\pm0.03}$ | $0.26_{\pm0.04}$ | $0.26_{\pm0.04}$ | $-0.70_{\pm0.02}$ |
| ConvGNP | $0.79_{\pm0.06}$ | $0.79_{\pm0.06}$ | $-1.03_{\pm0.07}$ | $0.23_{\pm0.04}$ | $0.23_{\pm0.04}$ | $-0.79_{\pm0.02}$ |
| FullConvGNP | $0.71_{\pm0.08}$ | $0.72_{\pm0.06}$ | $-0.20_{\pm0.02}$ | | | |
| ConvCNP | $0.57_{\pm0.07}$ | $0.57_{\pm0.07}$ | $-0.73_{\pm0.02}$ | $0.18_{\pm0.05}$ | $0.18_{\pm0.05}$ | $-0.86_{\pm0.03}$ |
| ACNP (AR) | $0.07_{\pm0.03}$ | $-0.85_{\pm0.03}$ | $-0.84_{\pm0.02}$ | $-0.53_{\pm0.02}$ | $-1.52_{\pm0.10}$ | $-1.51_{\pm0.09}$ |
| AGNP | $-0.31_{\pm0.03}$ | $-1.22_{\pm0.07}$ | $-1.58_{\pm0.11}$ | $-0.55_{\pm0.02}$ | $-0.79_{\pm0.02}$ | $-0.76_{\pm0.03}$ |
| ALNP (E–M) | $-0.33_{\pm0.03}$ | $-0.91_{\pm0.03}$ | $-0.80_{\pm0.04}$ | $-0.67_{\pm0.03}$ | $-1.06_{\pm0.62}$ | $-0.70_{\pm0.03}$ |
| ALNP (E) | $-0.35_{\pm0.03}$ | $-5.00_{\pm0.17}$ | $-3.37_{\pm0.06}$ | $-0.68_{\pm0.03}$ | $-2.59_{\pm7.64}$ | $-0.75_{\pm0.03}$ |
| ALNP (M) | $-0.36_{\pm0.02}$ | $-0.68_{\pm0.02}$ | $-0.68_{\pm0.02}$ | $-0.53_{\pm0.02}$ | $-0.74_{\pm0.04}$ | $-0.69_{\pm0.02}$ |
| GNP | $-0.38_{\pm0.02}$ | F | F | $-0.69_{\pm0.02}$ | $-0.74_{\pm0.04}$ | $-0.70_{\pm0.03}$ |
| LNP (E–M) | $-0.43_{\pm0.02}$ | F | $-3.34_{\pm0.53}$ | $-0.66_{\pm0.02}$ | F | $-0.96_{\pm0.03}$ |
| LNP (E) | $-0.44_{\pm0.02}$ | F | F | $-0.66_{\pm0.02}$ | F | F |
| ACNP | $-0.50_{\pm0.03}$ | $-0.83_{\pm0.03}$ | $-0.85_{\pm0.03}$ | $-0.60_{\pm0.02}$ | $-1.50_{\pm0.10}$ | $-0.73_{\pm0.03}$ |
| LNP (M) | $-0.53_{\pm0.02}$ | $-1.28_{\pm0.05}$ | $-0.80_{\pm0.03}$ | $-0.62_{\pm0.02}$ | $-1.50_{\pm0.11}$ | $-0.75_{\pm0.03}$ |
| CNP (AR) | $-0.65_{\pm0.02}$ | $-1.14_{\pm0.19}$ | $-0.98_{\pm0.06}$ | $-0.69_{\pm0.02}$ | $-1.05_{\pm0.07}$ | $-0.72_{\pm0.03}$ |
| CNP | $-0.68_{\pm0.02}$ | $-0.79_{\pm0.04}$ | $-0.73_{\pm0.03}$ | $-0.69_{\pm0.02}$ | $-1.05_{\pm0.08}$ | $-0.71_{\pm0.03}$ |
| *Trivial* | $-0.82_{\pm0.00}$ | $-0.82_{\pm0.00}$ | $-0.82_{\pm0.00}$ | $-0.82_{\pm0.00}$ | $-0.82_{\pm0.00}$ | $-0.82_{\pm0.00}$ |
| ConvLNP (E–M) | F | F | F | $-0.04_{\pm0.05}$ | $-0.04_{\pm0.05}$ | $-1.47_{\pm0.87}$ |

Table 9: For the EQ synthetic experiments with one-dimensional inputs, average Kullback–Leibler divergences of the posterior prediction map $\pi_y$ with respect to the model normalised by the number of target points. Shows for one-dimensional outputs ($d_y = 1$) and two-dimensional outputs ($d_y = 2$) the performance for interpolation within the range $[-2, 2]$ where the models where trained ("Int."); interpolation within the range $[2, 6]$ which the models have never seen before ("OOID"); and extrapolation from the range $[-2, 2]$ to the range $[2, 6]$ ("Ext."). Models are ordered by interpolation performance. The latent variable models are trained and evaluated with the ELBO objective (E); trained and evaluated with the ML objective (M); and trained with the ELBO objective and evaluated with the ML objective (E–M). Diagonal GP refers to predictions by the ground-truth Gaussian processes without correlations. Trivial refers to predicting the empirical means and standard deviation of the test data. Errors indicate the central 95%-confidence interval. Numbers which are significantly best ($p < 0.05$) are boldfaced. Numbers which are very large are marked as failed with "F". Numbers which are missing could not be run.

| EQ $d_x = 1$, $d_y = 1$ | Int. | OOID | Ext. | EQ $d_x = 1$, $d_y = 2$ | Int. | OOID | Ext. |
|---|---|---|---|---|---|---|---|
| FullConvGNP | **0.00** ±0.00 | **0.00** ±0.00 | **0.00** ±0.00 | FullConvGNP | **0.00** ±0.00 | **0.00** ±0.00 | **0.00** ±0.00 |
| ConvGNP | 0.01 ±0.00 | 0.01 ±0.00 | 3.46 ±0.08 | ConvGNP | 0.01 ±0.00 | 0.01 ±0.00 | 1.73 ±0.05 |
| ConvCNP (AR) | 0.01 ±0.00 | 0.01 ±0.00 | 0.01 ±0.00 | ConvCNP (AR) | 0.01 ±0.00 | 0.01 ±0.00 | 0.01 ±0.00 |
| AGNP | 0.03 ±0.00 | 4.28 ±0.08 | 7.38 ±0.13 | AGNP | 0.04 ±0.00 | 7.71 ±0.10 | 7.87 ±0.10 |
| ConvLNP (E) | 0.06 ±0.00 | 0.06 ±0.00 | 0.11 ±0.01 | ACNP (AR) | 0.07 ±0.00 | 1.31 ±0.01 | 1.09 ±0.01 |
| ACNP (AR) | 0.07 ±0.00 | 1.19 ±0.01 | 0.98 ±0.01 | ConvLNP (E) | 0.08 ±0.00 | 0.08 ±0.00 | 0.13 ±0.00 |
| GNP | 0.08 ±0.00 | F | F | GNP | 0.13 ±0.00 | F | F |
| ConvLNP (M) | 0.25 ±0.01 | 0.25 ±0.01 | 0.67 ±0.01 | ConvLNP (M) | 0.36 ±0.01 | 0.36 ±0.01 | 0.88 ±0.00 |
| CNP (AR) | 0.28 ±0.00 | F | F | ALNP (M) | 0.41 ±0.01 | 1.23 ±0.01 | 0.99 ±0.00 |
| ALNP (M) | 0.31 ±0.01 | 1.04 ±0.01 | 0.84 ±0.01 | CNP (AR) | 0.42 ±0.00 | F | F |
| LNP (E) | 0.34 ±0.01 | F | 1.34 ±0.01 | *Diagonal GP* | 0.47 ±0.01 | 0.47 ±0.01 | 1.06 ±0.00 |
| LNP (E–M) | 0.37 ±0.01 | F | 1.27 ±0.01 | ConvCNP | 0.48 ±0.01 | 0.48 ±0.01 | 1.06 ±0.00 |
| *Diagonal GP* | 0.40 ±0.01 | 0.40 ±0.01 | 0.95 ±0.01 | ACNP | 0.51 ±0.01 | 1.38 ±0.01 | 1.06 ±0.01 |
| ConvCNP | 0.41 ±0.01 | 0.41 ±0.01 | 0.95 ±0.01 | ALNP (E–M) | 0.52 ±0.01 | 1.47 ±0.02 | 1.07 ±0.01 |
| ANP (E–M) | 0.42 ±0.01 | 1.18 ±0.01 | 0.94 ±0.01 | ALNP (E) | 0.53 ±0.01 | 3.79 ±0.05 | 2.84 ±0.02 |
| ANP (E) | 0.44 ±0.01 | 1.32 ±0.01 | 1.25 ±0.01 | LNP (E) | 0.54 ±0.01 | F | 1.47 ±0.01 |
| ACNP | 0.45 ±0.01 | 1.22 ±0.01 | 0.95 ±0.01 | LNP (E–M) | 0.56 ±0.01 | F | 1.42 ±0.01 |
| LNP (M) | 0.49 ±0.01 | 1.54 ±0.01 | 1.45 ±0.01 | LNP (M) | 0.64 ±0.00 | F | 1.52 ±0.00 |
| CNP | 0.54 ±0.01 | F | 1.41 ±0.01 | CNP | 0.66 ±0.01 | F | 1.28 ±0.00 |
| ConvLNP (E–M) | 0.90 ±0.04 | 0.90 ±0.04 | 4.05 ±0.06 | *Trivial* | 1.31 ±0.00 | 1.31 ±0.00 | 1.07 ±0.00 |
| *Trivial* | 1.19 ±0.00 | 1.19 ±0.00 | 0.96 ±0.00 | ConvLNP (E–M) | 2.14 ±0.06 | 2.14 ±0.06 | 9.30 ±0.08 |

Table 10: For the EQ synthetic experiments with two-dimensional inputs, average Kullback–Leibler divergences of the posterior prediction map $\pi_y$ with respect to the model normalised by the number of target points. Shows for one-dimensional outputs ($d_y = 1$) and two-dimensional outputs ($d_y = 2$) the performance for interpolation within the range $[-2, 2]^2$ where the models where trained ("Int."); interpolation within the range $[2, 6]^2$ which the models have never seen before ("OOID"); and extrapolation from the range $[-2, 2]^2$ to the range $[2, 6]^2$ ("Ext."). Models are ordered by interpolation performance. The latent variable models are trained and evaluated with the ELBO objective (E); trained and evaluated with the ML objective (M); and trained with the ELBO objective and evaluated with the ML objective (E–M). Diagonal GP refers to predictions by the ground-truth Gaussian processes without correlations. Trivial refers to predicting the empirical means and standard deviation of the test data. Errors indicate the central 95%-confidence interval. Numbers which are significantly best ($p < 0.05$) are boldfaced. Numbers which are very large are marked as failed with "F". Numbers which are missing could not be run.

| EQ $d_x=2$, $d_y=1$ | Int. | OOID | Ext. |
|---|---|---|---|
| ConvCNP (AR) | **0.01** ±0.00 | **0.01** ±0.00 | **0.01** ±0.00 |
| ConvGNP | 0.08 ±0.00 | 0.08 ±0.00 | 1.92 ±0.02 |
| AGNP | 0.09 ±0.00 | 0.70 ±0.00 | 0.50 ±0.00 |
| ACNP (AR) | 0.09 ±0.00 | 0.72 ±0.00 | 0.51 ±0.00 |
| GNP | 0.19 ±0.00 | 1.01 ±0.00 | 0.80 ±0.00 |
| ConvLNP (M) | 0.34 ±0.00 | 0.34 ±0.00 | 0.47 ±0.00 |
| ALNP (M) | 0.34 ±0.00 | 0.70 ±0.00 | 0.51 ±0.00 |
| *Diagonal GP* | 0.36 ±0.00 | 0.36 ±0.00 | 0.51 ±0.00 |
| ConvCNP | 0.37 ±0.00 | 0.37 ±0.00 | 0.51 ±0.00 |
| ACNP | 0.40 ±0.00 | 0.72 ±0.00 | 0.51 ±0.00 |
| ConvLNP (E) | 0.41 ±0.00 | 0.41 ±0.00 | 0.46 ±0.00 |
| CNP (AR) | 0.41 ±0.00 | 0.90 ±0.00 | 0.71 ±0.00 |
| LNP (E–M) | 0.46 ±0.00 | 0.99 ±0.01 | 0.65 ±0.00 |
| LNP (E) | 0.48 ±0.00 | 1.04 ±0.01 | 0.67 ±0.00 |
| ConvLNP (E–M) | 0.48 ±0.00 | 0.48 ±0.00 | 0.59 ±0.01 |
| ALNP (E–M) | 0.49 ±0.01 | 0.72 ±0.00 | 0.51 ±0.00 |
| ALNP (E) | 0.50 ±0.00 | 0.73 ±0.00 | 0.52 ±0.00 |
| LNP (M) | 0.51 ±0.00 | 0.92 ±0.00 | 0.72 ±0.00 |
| CNP | 0.52 ±0.00 | 0.92 ±0.00 | 0.72 ±0.00 |
| *Trivial* | 0.72 ±0.00 | 0.72 ±0.00 | 0.51 ±0.00 |
| FullConvGNP | | | |

| EQ $d_x=2$, $d_y=2$ | Int. | OOID | Ext. |
|---|---|---|---|
| ConvCNP (AR) | **0.03** ±0.00 | **0.03** ±0.00 | **0.02** ±0.00 |
| ACNP (AR) | 0.11 ±0.00 | 0.79 ±0.00 | 0.56 ±0.00 |
| ConvGNP | 0.19 ±0.00 | 0.19 ±0.00 | 0.74 ±0.01 |
| AGNP | 0.22 ±0.00 | 0.87 ±0.01 | 0.57 ±0.00 |
| GNP | 0.38 ±0.00 | 1.06 ±0.00 | 0.75 ±0.00 |
| ConvNP (M) | 0.39 ±0.00 | 0.39 ±0.00 | 0.52 ±0.00 |
| *Diagonal GP* | 0.40 ±0.00 | 0.40 ±0.00 | 0.56 ±0.00 |
| ConvCNP | 0.41 ±0.00 | 0.41 ±0.00 | 0.56 ±0.00 |
| ANP (M) | 0.42 ±0.00 | 0.79 ±0.00 | 0.54 ±0.00 |
| ACNP | 0.44 ±0.00 | 0.79 ±0.00 | 0.56 ±0.00 |
| CNP (AR) | 0.52 ±0.00 | F | F |
| NP (E–M) | 0.56 ±0.00 | 1.95 ±0.03 | 0.72 ±0.00 |
| ANP (E–M) | 0.56 ±0.00 | 1.90 ±1.72 | 0.55 ±0.00 |
| NP (E) | 0.57 ±0.00 | 1.99 ±0.03 | 0.72 ±0.00 |
| ANP (E) | 0.57 ±0.00 | 3.51 ±3.04 | 0.56 ±0.00 |
| NP (M) | 0.59 ±0.00 | 1.17 ±0.01 | 0.75 ±0.00 |
| CNP | 0.60 ±0.00 | 3.08 ±0.42 | 0.66 ±0.00 |
| *Trivial* | 0.79 ±0.00 | 0.79 ±0.00 | 0.56 ±0.00 |
| ConvNP (E–M) | 0.79 ±0.00 | 0.79 ±0.00 | 0.56 ±0.00 |
| ConvNP (E) | 0.79 ±0.00 | 0.79 ±0.00 | 0.56 ±0.00 |
| FullConvGNP | | | |

Table 11: For the Matérn–$\frac{5}{2}$ synthetic experiments with one-dimensional inputs, average Kullback–Leibler divergences of the posterior prediction map $\pi_y$ with respect to the model normalised by the number of target points. Shows for one-dimensional outputs ($d_y = 1$) and two-dimensional outputs ($d_y = 2$) the performance for interpolation within the range $[-2, 2]$ where the models where trained ("Int."); interpolation within the range $[2, 6]$ which the models have never seen before ("OOID"); and extrapolation from the range $[-2, 2]$ to the range $[2, 6]$ ("Ext."). Models are ordered by interpolation performance. The latent variable models are trained and evaluated with the ELBO objective (E); trained and evaluated with the ML objective (M); and trained with the ELBO objective and evaluated with the ML objective (E–M). Diagonal GP refers to predictions by the ground-truth Gaussian processes without correlations. Trivial refers to predicting the empirical means and standard deviation of the test data. Errors indicate the central 95%-confidence interval. Numbers which are significantly best ($p < 0.05$) are boldfaced. Numbers which are very large are marked as failed with "F". Numbers which are missing could not be run.

| Matérn–$\frac{5}{2}$ $d_x = 1$, $d_y = 1$ | Int. | OOID | Ext. | Matérn–$\frac{5}{2}$ $d_x = 1$, $d_y = 2$ | Int. | OOID | Ext. |
|---|---|---|---|---|---|---|---|
| FullConvGNP | **0.00** ±0.00 | **0.00** ±0.00 | **0.00** ±0.00 | FullConvGNP | **0.00** ±0.00 | **0.00** ±0.00 | **0.00** ±0.00 |
| ConvCNP (AR) | 0.00 ±0.00 | 0.00 ±0.00 | 0.00 ±0.00 | ConvCNP (AR) | 0.01 ±0.00 | 0.01 ±0.00 | 0.01 ±0.00 |
| ConvGNP | 0.01 ±0.00 | 0.01 ±0.00 | 2.32 ±0.06 | ConvGNP | 0.02 ±0.00 | 0.02 ±0.00 | 1.71 ±0.04 |
| AGNP | 0.03 ±0.00 | 4.53 ±0.08 | 7.22 ±0.12 | AGNP | 0.04 ±0.00 | 6.14 ±0.08 | 7.03 ±0.09 |
| ACNP (AR) | 0.04 ±0.00 | 1.08 ±0.01 | 0.87 ±0.01 | ACNP (AR) | 0.05 ±0.00 | 1.18 ±0.01 | 0.96 ±0.01 |
| GNP | 0.09 ±0.00 | F | F | GNP | 0.13 ±0.00 | F | F |
| ConvLNP (E) | 0.13 ±0.00 | 0.13 ±0.00 | 0.31 ±0.02 | ConvLNP (E) | 0.16 ±0.00 | 0.16 ±0.00 | 0.29 ±0.00 |
| ConvLNP (M) | 0.26 ±0.01 | 0.26 ±0.01 | 0.58 ±0.00 | ConvLNP (M) | 0.36 ±0.00 | 0.36 ±0.00 | 0.76 ±0.00 |
| ALNP (M) | 0.30 ±0.00 | 0.98 ±0.01 | 0.78 ±0.01 | ALNP (M) | 0.40 ±0.00 | 1.10 ±0.01 | 0.88 ±0.00 |
| CNP (AR) | 0.34 ±0.01 | 1.81 ±0.04 | 1.32 ±0.02 | CNP (AR) | 0.45 ±0.00 | F | F |
| LNP (E) | 0.36 ±0.00 | F | 1.31 ±0.01 | Diagonal GP | 0.46 ±0.01 | 0.46 ±0.01 | 0.93 ±0.00 |
| LNP (E–M) | 0.37 ±0.01 | F | 1.14 ±0.00 | ConvCNP | 0.46 ±0.01 | 0.46 ±0.01 | 0.93 ±0.00 |
| Diagonal GP | 0.40 ±0.01 | 0.40 ±0.01 | 0.84 ±0.01 | ACNP | 0.49 ±0.01 | 1.23 ±0.01 | 0.93 ±0.00 |
| ConvCNP | 0.40 ±0.01 | 0.40 ±0.01 | 0.84 ±0.01 | ALNP (E–M) | 0.51 ±0.01 | 1.28 ±0.01 | 0.99 ±0.01 |
| ALNP (E–M) | 0.41 ±0.01 | 1.13 ±0.01 | 0.84 ±0.01 | ALNP (E) | 0.51 ±0.01 | 1.43 ±0.02 | 1.10 ±0.01 |
| ACNP | 0.42 ±0.01 | 1.10 ±0.01 | 0.84 ±0.01 | LNP (E–M) | 0.54 ±0.00 | F | 1.24 ±0.00 |
| ALNP (E) | 0.43 ±0.01 | 1.15 ±0.01 | 0.87 ±0.01 | LNP (E) | 0.54 ±0.00 | F | 1.79 ±0.01 |
| LNP (M) | 0.51 ±0.00 | 1.87 ±0.02 | 1.30 ±0.01 | LNP (M) | 0.63 ±0.00 | 2.33 ±0.02 | 1.23 ±0.00 |
| CNP | 0.54 ±0.01 | 1.47 ±0.02 | 1.11 ±0.01 | CNP | 0.65 ±0.00 | 7.72 ±0.69 | 1.23 ±0.00 |
| Trivial | 1.08 ±0.00 | 1.08 ±0.00 | 0.85 ±0.00 | Trivial | 1.18 ±0.00 | 1.18 ±0.00 | 0.94 ±0.00 |
| ConvLNP (E–M) | 1.37 ±0.04 | 1.36 ±0.04 | 4.30 ±0.06 | ConvLNP (E–M) | 3.07 ±0.06 | 3.06 ±0.06 | 9.83 ±0.09 |

Table 12: For the Matérn–$\frac{5}{2}$ synthetic experiments with two-dimensional inputs, average Kullback–Leibler divergences of the posterior prediction map $\pi_y$ with respect to the model normalised by the number of target points. Shows for one-dimensional outputs ($d_y = 1$) and two-dimensional outputs ($d_y = 2$) the performance for interpolation within the range $[-2, 2]^2$ where the models where trained ("Int."); interpolation within the range $[2, 6]^2$ which the models have never seen before ("OOID"); and extrapolation from the range $[-2, 2]^2$ to the range $[2, 6]^2$ ("Ext."). Models are ordered by interpolation performance. Diagonal GP refers to predictions by the ground-truth Gaussian processes without correlations. Trivial refers to predicting the empirical means and standard deviation of the test data. Errors indicate the central 95%-confidence interval. Numbers which are significantly best ($p < 0.05$) are boldfaced. Numbers which are very large are marked as failed with "F". Numbers which are missing could not be run.

| Matérn–$\frac{5}{2}$ $d_x = 2$, $d_y = 1$ | Int. | OOID | Ext. | Matérn–$\frac{5}{2}$ $d_x = 2$, $d_y = 2$ | Int. | OOID | Ext. |
|---|---|---|---|---|---|---|---|
| ConvCNP (AR) | **0.01** ±0.00 | **0.01** ±0.00 | **0.00** ±0.00 | ConvCNP (AR) | **0.01** ±0.00 | **0.01** ±0.00 | **0.01** ±0.00 |
| ACNP (AR) | 0.05 ±0.00 | 0.54 ±0.00 | 0.38 ±0.00 | ACNP (AR) | 0.06 ±0.00 | 0.58 ±0.00 | 0.41 ±0.00 |
| AGNP | 0.08 ±0.00 | 0.83 ±0.01 | 0.37 ±0.00 | ConvGNP | 0.14 ±0.00 | 0.14 ±0.00 | 0.64 ±0.01 |
| ConvGNP | 0.08 ±0.00 | 0.08 ±0.00 | 0.60 ±0.01 | AGNP | 0.17 ±0.00 | 0.58 ±0.00 | 0.40 ±0.00 |
| GNP | 0.16 ±0.00 | 0.90 ±0.00 | 0.75 ±0.00 | GNP | 0.28 ±0.00 | 0.78 ±0.00 | 0.60 ±0.00 |
| ConvLNP (M) | 0.25 ±0.00 | 0.25 ±0.00 | 0.34 ±0.00 | ConvLNP (M) | 0.29 ±0.00 | 0.29 ±0.00 | 0.38 ±0.00 |
| ALNP (M) | 0.26 ±0.00 | 0.51 ±0.00 | 0.37 ±0.00 | ALNP (M) | 0.29 ±0.00 | 0.56 ±0.00 | 0.40 ±0.00 |
| *Diagonal GP* | 0.28 ±0.00 | 0.28 ±0.00 | 0.39 ±0.00 | *Diagonal GP* | 0.30 ±0.00 | 0.30 ±0.00 | 0.41 ±0.00 |
| ConvCNP | 0.28 ±0.00 | 0.28 ±0.00 | 0.39 ±0.00 | ConvCNP | 0.30 ±0.00 | 0.30 ±0.00 | 0.41 ±0.00 |
| ACNP | 0.29 ±0.00 | 0.54 ±0.00 | 0.39 ±0.00 | ACNP | 0.32 ±0.00 | 0.58 ±0.00 | 0.41 ±0.00 |
| CNP (AR) | 0.31 ±0.00 | 0.69 ±0.00 | 0.52 ±0.00 | ConvLNP (E) | 0.36 ±0.00 | 0.36 ±0.00 | 0.37 ±0.00 |
| ConvLNP (E) | 0.32 ±0.00 | 0.32 ±0.00 | 0.30 ±0.00 | CNP (AR) | 0.37 ±0.00 | F | 0.88 ±0.17 |
| LNP (E–M) | 0.34 ±0.00 | 1.07 ±0.01 | 0.69 ±0.00 | LNP (E–M) | 0.41 ±0.00 | 2.29 ±0.05 | 0.59 ±0.00 |
| LNP (E) | 0.35 ±0.00 | 1.25 ±0.01 | 0.72 ±0.00 | LNP (E) | 0.41 ±0.00 | 2.36 ±0.05 | 0.60 ±0.00 |
| ConvLNP (E–M) | 0.36 ±0.00 | 0.36 ±0.00 | 0.43 ±0.00 | ALNP (E–M) | 0.42 ±0.00 | 0.61 ±0.00 | 0.41 ±0.00 |
| LNP (M) | 0.37 ±0.00 | 0.75 ±0.01 | 0.51 ±0.00 | ALNP (E) | 0.42 ±0.00 | 0.61 ±0.00 | 0.41 ±0.00 |
| ALNP (E–M) | 0.39 ±0.00 | 0.65 ±0.01 | 0.43 ±0.00 | LNP (M) | 0.43 ±0.00 | 0.68 ±0.00 | 0.53 ±0.00 |
| CNP | 0.39 ±0.00 | 0.67 ±0.00 | 0.54 ±0.00 | CNP | 0.44 ±0.00 | 0.86 ±0.17 | 0.59 ±0.00 |
| ALNP (E) | 0.41 ±0.01 | 0.67 ±0.01 | 0.44 ±0.00 | ConvLNP (E–M) | 0.49 ±0.00 | 0.49 ±0.00 | 0.61 ±0.00 |
| *Trivial* | 0.55 ±0.00 | 0.55 ±0.00 | 0.39 ±0.00 | *Trivial* | 0.58 ±0.00 | 0.58 ±0.00 | 0.41 ±0.00 |
| FullConvGNP | | | | FullConvGNP | | | |

Table 13: For the weakly periodic synthetic experiments with one-dimensional inputs, average Kullback–Leibler divergences of the posterior prediction map $\pi_y$ with respect to the model normalised by the number of target points. Shows for one-dimensional outputs ($d_y = 1$) and two-dimensional outputs ($d_y = 2$) the performance for interpolation within the range $[-2, 2]$ where the models where trained ("Int."); interpolation within the range $[2, 6]$ which the models have never seen before ("OOID"); and extrapolation from the range $[-2, 2]$ to the range $[2, 6]$ ("Ext."). Models are ordered by interpolation performance. The latent variable models are trained and evaluated with the ELBO objective (E); trained and evaluated with the ML objective (M); and trained with the ELBO objective and evaluated with the ML objective (E–M). Diagonal GP refers to predictions by the ground-truth Gaussian processes without correlations. Trivial refers to predicting the empirical means and standard deviation of the test data. Errors indicate the central 95%-confidence interval. Numbers which are significantly best ($p < 0.05$) are boldfaced. Numbers which are very large are marked as failed with "F". Numbers which are missing could not be run.

| Weakly Periodic $d_x=1,\ d_y=1$ | Int. | OOID | Ext. | Weakly Periodic $d_x=1,\ d_y=2$ | Int. | OOID | Ext. |
|---|---|---|---|---|---|---|---|
| FullConvGNP | **0.02** ±0.00 | **0.02** ±0.00 | **0.00** ±0.00 | FullConvGNP | **0.03** ±0.00 | **0.03** ±0.00 | **0.00** ±0.00 |
| ConvCNP (AR) | 0.05 ±0.00 | 0.05 ±0.00 | 0.04 ±0.00 | ConvCNP (AR) | 0.09 ±0.00 | 0.09 ±0.00 | 0.06 ±0.00 |
| ConvGNP | 0.05 ±0.00 | 0.05 ±0.00 | 0.56 ±0.02 | ConvGNP | 0.12 ±0.00 | 0.12 ±0.00 | 0.72 ±0.01 |
| AGNP | 0.22 ±0.00 | 1.25 ±0.02 | 1.25 ±0.02 | AGNP | 0.25 ±0.00 | 2.17 ±0.02 | 1.95 ±0.02 |
| ConvLNP (M) | 0.28 ±0.00 | 0.28 ±0.00 | 0.43 ±0.00 | ConvLNP (M) | 0.38 ±0.00 | 0.38 ±0.00 | 0.54 ±0.00 |
| ConvLNP (E) | 0.34 ±0.00 | 0.33 ±0.00 | 0.45 ±0.02 | ConvLNP (E) | 0.39 ±0.00 | 0.39 ±0.00 | 0.44 ±0.00 |
| Diagonal GP | 0.38 ±0.01 | 0.38 ±0.01 | 0.59 ±0.01 | Diagonal GP | 0.42 ±0.00 | 0.42 ±0.00 | 0.65 ±0.00 |
| ConvCNP | 0.40 ±0.01 | 0.40 ±0.01 | 0.60 ±0.01 | ConvCNP | 0.46 ±0.00 | 0.46 ±0.00 | 0.65 ±0.00 |
| ALNP (M) | 0.53 ±0.00 | 0.77 ±0.01 | 0.57 ±0.01 | GNP | 0.50 ±0.00 | 1.02 ±0.01 | 0.76 ±0.00 |
| ACNP (AR) | 0.57 ±0.01 | 0.82 ±0.01 | 0.61 ±0.01 | ALNP (M) | 0.62 ±0.00 | 1.04 ±0.01 | 0.64 ±0.00 |
| GNP | 0.59 ±0.01 | 1.31 ±0.02 | 0.62 ±0.01 | ACNP (AR) | 0.63 ±0.00 | 0.89 ±0.01 | 0.66 ±0.00 |
| CNP (AR) | 0.59 ±0.01 | 2.33 ±0.27 | 1.46 ±0.05 | CNP (AR) | 0.67 ±0.00 | 2.52 ±0.07 | 1.21 ±0.01 |
| LNP (E–M) | 0.60 ±0.01 | F | 4.09 ±0.28 | LNP (E–M) | 0.68 ±0.00 | F | 5.18 ±0.18 |
| ALNP (E–M) | 0.60 ±0.01 | 0.78 ±0.01 | 0.59 ±0.01 | LNP (M) | 0.69 ±0.00 | 1.26 ±0.01 | 0.68 ±0.01 |
| LNP (M) | 0.60 ±0.01 | 0.80 ±0.01 | 0.62 ±0.01 | LNP (E) | 0.69 ±0.00 | F | F |
| LNP (E) | 0.61 ±0.01 | F | 9.91 ±0.70 | ALNP (E–M) | 0.70 ±0.00 | 0.85 ±0.01 | 0.64 ±0.00 |
| ALNP (E) | 0.62 ±0.01 | 1.01 ±0.01 | 0.71 ±0.01 | ACNP | 0.71 ±0.00 | 0.89 ±0.01 | 0.66 ±0.00 |
| ACNP | 0.65 ±0.01 | 0.82 ±0.01 | 0.61 ±0.01 | ALNP (E) | 0.72 ±0.00 | 0.93 ±0.01 | 0.72 ±0.00 |
| CNP | 0.67 ±0.01 | 1.45 ±0.03 | 0.68 ±0.01 | CNP | 0.74 ±0.00 | 1.27 ±0.01 | 0.77 ±0.01 |
| Trivial | 0.82 ±0.00 | 0.82 ±0.00 | 0.61 ±0.00 | Trivial | 0.89 ±0.00 | 0.89 ±0.00 | 0.67 ±0.00 |
| ConvLNP (E–M) | 1.58 ±0.03 | 1.57 ±0.03 | 2.85 ±0.04 | ConvLNP (E–M) | 3.02 ±0.03 | 3.02 ±0.03 | 5.40 ±0.04 |

Table 14: For the weakly periodic synthetic experiments with two-dimensional inputs, average Kullback–Leibler divergences of the posterior prediction map $\pi_y$ with respect to the model normalised by the number of target points. Shows for one-dimensional outputs ($d_y = 1$) and two-dimensional outputs ($d_y = 2$) the performance for interpolation within the range $[-2, 2]^2$ where the models where trained ("Int."); interpolation within the range $[2, 6]^2$ which the models have never seen before ("OOID"); and extrapolation from the range $[-2, 2]^2$ to the range $[2, 6]^2$ ("Ext."). Models are ordered by interpolation performance. The latent variable models are trained and evaluated with the ELBO objective (E); trained and evaluated with the ML objective (M); and trained with the ELBO objective and evaluated with the ML objective (E–M). Diagonal GP refers to predictions by the ground-truth Gaussian processes without correlations. Trivial refers to predicting the empirical means and standard deviation of the test data. Errors indicate the central 95%-confidence interval. Numbers which are significantly best ($p < 0.05$) are boldfaced. Numbers which are very large are marked as failed with "F". Numbers which are missing could not be run.

| Weakly Periodic $d_x = 2$, $d_y = 1$ | Int. | OOID | Ext. | Weakly Periodic $d_x = 2$, $d_y = 2$ | Int. | OOID | Ext. |
|---|---|---|---|---|---|---|---|
| ConvCNP (AR) | **0.05** ±0.00 | **0.05** ±0.00 | **0.03** ±0.00 | ConvCNP (AR) | **0.08** ±0.00 | **0.08** ±0.00 | **0.05** ±0.00 |
| ConvGNP | 0.10 ±0.00 | 0.10 ±0.00 | 0.19 ±0.00 | ConvGNP | 0.13 ±0.00 | 0.13 ±0.00 | 0.18 ±0.00 |
| ConvLNP (M) | 0.18 ±0.00 | 0.18 ±0.00 | 0.21 ±0.00 | Diagonal GP | 0.20 ±0.00 | 0.20 ±0.00 | 0.28 ±0.00 |
| Diagonal GP | 0.19 ±0.00 | 0.19 ±0.00 | 0.27 ±0.00 | ConvLNP (M) | 0.21 ±0.00 | 0.21 ±0.00 | 0.25 ±0.00 |
| ConvCNP | 0.20 ±0.00 | 0.20 ±0.00 | 0.27 ±0.00 | ConvCNP | 0.23 ±0.00 | 0.23 ±0.00 | 0.28 ±0.00 |
| ConvLNP (E) | 0.22 ±0.00 | 0.22 ±0.00 | 0.22 ±0.00 | AGNP | 0.24 ±0.00 | 0.39 ±0.00 | 0.27 ±0.00 |
| AGNP | 0.23 ±0.00 | 0.35 ±0.00 | 0.26 ±0.00 | ACNP (AR) | 0.25 ±0.00 | 0.40 ±0.00 | 0.28 ±0.00 |
| ACNP (AR) | 0.23 ±0.00 | 0.37 ±0.00 | 0.26 ±0.00 | GNP | 0.25 ±0.00 | 0.38 ±0.00 | 0.25 ±0.00 |
| ConvLNP (E–M) | 0.23 ±0.00 | 0.23 ±0.00 | 0.29 ±0.00 | ConvLNP (E) | 0.26 ±0.00 | 0.26 ±0.00 | 0.27 ±0.00 |
| GNP | 0.23 ±0.00 | 0.35 ±0.00 | 0.24 ±0.00 | CNP (AR) | 0.27 ±0.00 | 3.21 ±0.14 | 0.77 ±0.02 |
| CNP (AR) | 0.25 ±0.00 | 0.82 ±0.01 | 0.72 ±0.01 | ALNP (M) | 0.28 ±0.00 | 0.39 ±0.00 | 0.28 ±0.00 |
| ALNP (M) | 0.25 ±0.00 | 0.36 ±0.00 | 0.25 ±0.00 | LNP (M) | 0.29 ±0.00 | 0.42 ±0.00 | 0.31 ±0.00 |
| LNP (M) | 0.26 ±0.00 | 1.32 ±0.03 | 0.33 ±0.00 | LNP (E–M) | 0.29 ±0.00 | 1.52 ±0.04 | 0.36 ±0.00 |
| LNP (E–M) | 0.26 ±0.00 | 6.28 ±1.06 | 0.44 ±0.01 | LNP (E) | 0.30 ±0.00 | 1.60 ±0.05 | 0.39 ±0.00 |
| LNP (E) | 0.27 ±0.00 | 9.91 ±1.68 | 0.52 ±0.01 | ConvLNP (E–M) | 0.30 ±0.00 | 0.30 ±0.00 | 0.34 ±0.00 |
| ACNP | 0.28 ±0.00 | 0.37 ±0.00 | 0.27 ±0.00 | ACNP | 0.30 ±0.00 | 0.40 ±0.00 | 0.29 ±0.00 |
| CNP | 0.29 ±0.00 | 0.81 ±0.01 | 0.29 ±0.00 | CNP | 0.31 ±0.00 | 0.63 ±0.01 | 0.33 ±0.00 |
| ALNP (E–M) | 0.31 ±0.00 | 0.39 ±0.00 | 0.26 ±0.00 | ALNP (E–M) | 0.36 ±0.00 | 0.43 ±0.00 | 0.28 ±0.00 |
| ALNP (E) | 0.32 ±0.00 | 0.42 ±0.00 | 0.29 ±0.00 | ALNP (E) | 0.36 ±0.00 | 0.44 ±0.00 | 0.29 ±0.00 |
| Trivial | 0.38 ±0.00 | 0.38 ±0.00 | 0.27 ±0.00 | Trivial | 0.40 ±0.00 | 0.40 ±0.00 | 0.28 ±0.00 |
| FullConvGNP | | | | FullConvGNP | | | |

Table 15: For the sawtooth synthetic experiments with one-dimensional inputs, average log-likelihoods normalised by the number of target points. Shows for one-dimensional outputs ($d_y = 1$) and two-dimensional outputs ($d_y = 2$) the performance for interpolation within the range $[-2, 2]$ where the models where trained ("Int."); interpolation within the range $[2, 6]$ which the models have never seen before ("OOID"); and extrapolation from the range $[-2, 2]$ to the range $[2, 6]$ ("Ext."). Models are ordered by interpolation performance. The latent variable models are trained and evaluated with the ELBO objective (E); trained and evaluated with the ML objective (M); and trained with the ELBO objective and evaluated with the ML objective (E–M). Trivial refers to predicting the empirical means and standard deviation of the test data. Errors indicate the central 95%-confidence interval. Numbers which are significantly best ($p < 0.05$) are boldfaced. Numbers which are very large are marked as failed with "F". Numbers which are missing could not be run.

| Sawtooth $d_x = 1$, $d_y = 1$ | Int. | OOID | Ext. | Sawtooth $d_x = 1$, $d_y = 2$ | Int. | OOID | Ext. |
|---|---|---|---|---|---|---|---|
| ConvCNP (AR) | $\mathbf{3.60}_{\pm 0.01}$ | $\mathbf{3.60}_{\pm 0.01}$ | $\mathbf{3.34}_{\pm 0.01}$ | ConvCNP (AR) | $\mathbf{2.22}_{\pm 0.01}$ | $\mathbf{2.22}_{\pm 0.01}$ | $\mathbf{1.91}_{\pm 0.01}$ |
| ConvLNP (E) | $3.51_{\pm 0.02}$ | $3.52_{\pm 0.02}$ | $2.68_{\pm 0.02}$ | ConvLNP (E) | $2.01_{\pm 0.01}$ | $2.01_{\pm 0.01}$ | $1.47_{\pm 0.01}$ |
| ConvLNP | $3.06_{\pm 0.04}$ | $3.06_{\pm 0.04}$ | $0.64_{\pm 0.01}$ | ConvLNP | $1.73_{\pm 0.03}$ | $1.73_{\pm 0.03}$ | $-0.12_{\pm 0.01}$ |
| ConvGNP | $2.62_{\pm 0.05}$ | $2.61_{\pm 0.08}$ | $-0.04_{\pm 0.01}$ | ACNP (AR) | $1.01_{\pm 0.01}$ | $-0.47_{\pm 0.01}$ | $-0.45_{\pm 0.01}$ |
| ConvCNP | $2.38_{\pm 0.04}$ | $2.37_{\pm 0.04}$ | $-0.00_{\pm 0.01}$ | FullConvGNP | $0.99_{\pm 0.06}$ | $1.04_{\pm 0.03}$ | $0.11_{\pm 0.00}$ |
| FullConvGNP | $2.16_{\pm 0.04}$ | $2.15_{\pm 0.04}$ | $0.18_{\pm 0.01}$ | ConvCNP | $0.83_{\pm 0.03}$ | $0.84_{\pm 0.02}$ | $-0.29_{\pm 0.00}$ |
| ALNP (E–M) | $0.27_{\pm 0.01}$ | $-0.18_{\pm 0.00}$ | $-0.31_{\pm 0.02}$ | ConvGNP | $0.82_{\pm 0.03}$ | $0.82_{\pm 0.03}$ | $-0.29_{\pm 0.00}$ |
| ALNP (E) | $0.27_{\pm 0.01}$ | $-15.96_{\pm 0.17}$ | $-9.14_{\pm 0.04}$ | GNP | $-0.03_{\pm 0.00}$ | F | F |
| ALNP | $0.20_{\pm 0.00}$ | $-0.18_{\pm 0.00}$ | $-0.18_{\pm 0.00}$ | ALNP (E–M) | $-0.06_{\pm 0.01}$ | $-0.75_{\pm 0.01}$ | $-0.35_{\pm 0.00}$ |
| LNP (E–M) | $0.07_{\pm 0.01}$ | F | $-8.67_{\pm 0.53}$ | ALNP (E) | $-0.07_{\pm 0.01}$ | $-0.78_{\pm 0.02}$ | $-0.35_{\pm 0.00}$ |
| LNP (E) | $0.06_{\pm 0.01}$ | F | F | AGNP | $-0.07_{\pm 0.01}$ | $-0.37_{\pm 0.00}$ | $-0.48_{\pm 0.07}$ |
| Trivial | $-0.18_{\pm 0.00}$ | $-0.18_{\pm 0.00}$ | $-0.18_{\pm 0.00}$ | ACNP | $-0.08_{\pm 0.01}$ | $-0.41_{\pm 0.01}$ | $-0.42_{\pm 0.01}$ |
| LNP | $-0.18_{\pm 0.00}$ | $-0.18_{\pm 0.00}$ | $-0.18_{\pm 0.00}$ | LNP (E–M) | $-0.20_{\pm 0.00}$ | F | $-1.33_{\pm 0.00}$ |
| CNP (AR) | $-0.18_{\pm 0.00}$ | $-0.18_{\pm 0.00}$ | $-0.18_{\pm 0.00}$ | LNP (E) | $-0.20_{\pm 0.00}$ | F | $-1.33_{\pm 0.00}$ |
| CNP | $-0.18_{\pm 0.00}$ | $-0.18_{\pm 0.00}$ | $-0.18_{\pm 0.00}$ | ALNP | $-0.28_{\pm 0.00}$ | $-0.33_{\pm 0.00}$ | $-0.33_{\pm 0.00}$ |
| GNP | $-0.18_{\pm 0.00}$ | $-0.18_{\pm 0.00}$ | $-0.18_{\pm 0.00}$ | CNP (AR) | $-0.29_{\pm 0.00}$ | $-1.96_{\pm 0.19}$ | $-1.43_{\pm 0.06}$ |
| AGNP | $-0.18_{\pm 0.00}$ | $-0.18_{\pm 0.00}$ | $-0.18_{\pm 0.00}$ | CNP | $-0.30_{\pm 0.00}$ | $-0.51_{\pm 0.01}$ | $-0.34_{\pm 0.00}$ |
| ACNP | $-0.18_{\pm 0.00}$ | $-0.18_{\pm 0.00}$ | $-0.18_{\pm 0.00}$ | LNP | $-0.32_{\pm 0.00}$ | $-0.32_{\pm 0.00}$ | $-0.32_{\pm 0.00}$ |
| ACNP (AR) | $-0.18_{\pm 0.00}$ | $-0.18_{\pm 0.00}$ | $-0.18_{\pm 0.00}$ | Trivial | $-0.33_{\pm 0.00}$ | $-0.33_{\pm 0.00}$ | $-0.33_{\pm 0.00}$ |
| ConvLNP (E–M) | F | F | F | ConvLNP (E–M) | $-2.98_{\pm 0.10}$ | $-2.98_{\pm 0.10}$ | $-6.74_{\pm 0.04}$ |

Table 16: For the sawtooth synthetic experiments with two-dimensional inputs, average log-likelihoods normalised by the number of target points. Shows for one-dimensional outputs ($d_y = 1$) and two-dimensional outputs ($d_y = 2$) the performance for interpolation within the range $[-2, 2]^2$ where the models where trained ("Int."); interpolation within the range $[2, 6]^2$ which the models have never seen before ("OOID"); and extrapolation from the range $[-2, 2]^2$ to the range $[2, 6]^2$ ("Ext."). Models are ordered by interpolation performance. The latent variable models are trained and evaluated with the ELBO objective (E); trained and evaluated with the ML objective (M); and trained with the ELBO objective and evaluated with the ML objective (E–M). Trivial refers to predicting the empirical means and standard deviation of the test data. Errors indicate the central 95%-confidence interval. Numbers which are significantly best ($p < 0.05$) are boldfaced. Numbers which are very large are marked as failed with "F". Numbers which are missing could not be run.

| Sawtooth $d_x = 2$, $d_y = 1$ | Int. | OOID | Ext. | Sawtooth $d_x = 2$, $d_y = 2$ | Int. | OOID | Ext. |
|---|---|---|---|---|---|---|---|
| ConvCNP (AR) | **2.59** ±0.01 | **2.59** ±0.01 | **2.10** ±0.01 | ConvCNP (AR) | **0.38** ±0.00 | **0.38** ±0.00 | **0.18** ±0.00 |
| ConvLNP (M) | 2.07 ±0.02 | 2.08 ±0.02 | −0.17 ±0.00 | ConvLNP (M) | 0.31 ±0.01 | 0.31 ±0.01 | −0.32 ±0.00 |
| ConvCNP | 1.93 ±0.04 | 1.94 ±0.03 | −0.18 ±0.00 | ConvGNP | 0.26 ±0.01 | 0.26 ±0.01 | −0.33 ±0.00 |
| ConvGNP | 1.90 ±0.04 | 1.91 ±0.03 | −0.18 ±0.00 | ConvCNP | 0.12 ±0.01 | 0.12 ±0.01 | −0.32 ±0.00 |
| ConvLNP (E) | 1.77 ±0.02 | 1.77 ±0.02 | 0.33 ±0.02 | ConvLNP (E) | 0.04 ±0.00 | 0.04 ±0.00 | −0.30 ±0.00 |
| ConvLNP (E–M) | 1.71 ±0.04 | 1.72 ±0.04 | −2.30 ±0.87 | ConvLNP (E–M) | −0.07 ±0.01 | −0.07 ±0.01 | −0.48 ±0.00 |
| *Trivial* | −0.18 ±0.00 | −0.18 ±0.00 | −0.18 ±0.00 | *Trivial* | −0.32 ±0.00 | −0.32 ±0.00 | −0.32 ±0.00 |
| CNP (AR) | −0.18 ±0.00 | −0.18 ±0.00 | −0.18 ±0.00 | ALNP (M) | −0.32 ±0.00 | −0.32 ±0.00 | −0.32 ±0.00 |
| CNP | −0.18 ±0.00 | −0.18 ±0.00 | −0.18 ±0.00 | CNP (AR) | −0.32 ±0.00 | −0.32 ±0.00 | −0.32 ±0.00 |
| GNP | −0.18 ±0.00 | −0.18 ±0.00 | −0.18 ±0.00 | CNP | −0.32 ±0.00 | −0.32 ±0.00 | −0.32 ±0.00 |
| LNP (M) | −0.18 ±0.00 | −0.18 ±0.00 | −0.18 ±0.00 | LNP (M) | −0.32 ±0.00 | −0.32 ±0.00 | −0.32 ±0.00 |
| AGNP | −0.18 ±0.00 | −0.18 ±0.00 | −0.18 ±0.00 | ACNP | −0.32 ±0.00 | −0.32 ±0.00 | −0.32 ±0.00 |
| ALNP (M) | −0.18 ±0.00 | −0.18 ±0.00 | −0.18 ±0.00 | ACNP (AR) | −0.32 ±0.00 | −0.32 ±0.00 | −0.32 ±0.00 |
| ACNP (AR) | −0.18 ±0.00 | −0.18 ±0.00 | −0.18 ±0.00 | GNP | −0.32 ±0.00 | −0.32 ±0.00 | −0.32 ±0.00 |
| ACNP | −0.18 ±0.00 | −0.18 ±0.00 | −0.18 ±0.00 | AGNP | −0.32 ±0.00 | −0.32 ±0.00 | −0.32 ±0.00 |
| LNP (E–M) | −0.19 ±0.00 | F | −0.86 ±0.02 | LNP (E–M) | −0.33 ±0.00 | −0.54 ±0.00 | −0.34 ±0.00 |
| LNP (E) | −0.19 ±0.00 | F | F | LNP (E) | −0.33 ±0.00 | −0.54 ±0.00 | −0.34 ±0.00 |
| ALNP (E–M) | −0.20 ±0.01 | −0.18 ±0.00 | −0.18 ±0.00 | ALNP (E–M) | −0.36 ±0.00 | −0.33 ±0.00 | −0.33 ±0.00 |
| ALNP (E) | −0.20 ±0.01 | −0.71 ±0.00 | −0.33 ±0.00 | ALNP (E) | −0.36 ±0.00 | −0.33 ±0.00 | −0.33 ±0.00 |
| FullConvGNP | | | | FullConvGNP | | | |

Table 17: For the mixture synthetic experiments with one-dimensional inputs, average log-likelihoods normalised by the number of target points. Shows for one-dimensional outputs ($d_y = 1$) and two-dimensional outputs ($d_y = 2$) the performance for interpolation within the range $[-2, 2]$ where the models where trained ("Int."); interpolation within the range $[2, 6]$ which the models have never seen before ("OOID"); and extrapolation from the range $[-2, 2]$ to the range $[2, 6]$ ("Ext."). Models are ordered by interpolation performance. The latent variable models are trained and evaluated with the ELBO objective (E); trained and evaluated with the ML objective (M); and trained with the ELBO objective and evaluated with the ML objective (E–M). Trivial refers to predicting the empirical means and standard deviation of the test data. Errors indicate the central 95%-confidence interval. Numbers which are significantly best ($p < 0.05$) are boldfaced. Numbers which are very large are marked as failed with "F". Numbers which are missing could not be run.

| Mixture $d_x = 1$, $d_y = 1$ | Int. | OOID | Ext. | Mixture $d_x = 1$, $d_y = 2$ | Int. | OOID | Ext. |
|---|---|---|---|---|---|---|---|
| ConvCNP (AR) | $\mathbf{0.45}_{\pm 0.04}$ | $\mathbf{0.45}_{\pm 0.04}$ | $\mathbf{0.30}_{\pm 0.04}$ | ConvLNP (E) | $\mathbf{-0.05}_{\pm 0.03}$ | $\mathbf{-0.05}_{\pm 0.03}$ | $\mathbf{-0.50}_{\pm 0.02}$ |
| ConvLNP (E) | $0.12_{\pm 0.04}$ | $0.12_{\pm 0.04}$ | $-0.37_{\pm 0.03}$ | ConvCNP (AR) | $-0.17_{\pm 0.02}$ | $-0.17_{\pm 0.02}$ | $-0.29_{\pm 0.02}$ |
| FullConvGNP | $-0.05_{\pm 0.03}$ | $-0.05_{\pm 0.03}$ | $-0.49_{\pm 0.01}$ | FullConvGNP | $-0.27_{\pm 0.01}$ | $-0.27_{\pm 0.01}$ | $-0.63_{\pm 0.01}$ |
| ConvLNP | $-0.06_{\pm 0.03}$ | $-0.06_{\pm 0.03}$ | $-0.88_{\pm 0.02}$ | ConvGNP | $-0.29_{\pm 0.02}$ | $-0.29_{\pm 0.02}$ | $-2.59_{\pm 0.06}$ |
| ACNP (AR) | $-0.19_{\pm 0.02}$ | $-1.32_{\pm 0.01}$ | $-1.32_{\pm 0.01}$ | ACNP (AR) | $-0.35_{\pm 0.02}$ | $-1.43_{\pm 0.02}$ | $-1.43_{\pm 0.02}$ |
| ConvCNP | $-0.23_{\pm 0.04}$ | $-0.24_{\pm 0.04}$ | $-1.23_{\pm 0.01}$ | ConvLNP | $-0.39_{\pm 0.03}$ | $-0.40_{\pm 0.03}$ | $-1.10_{\pm 0.02}$ |
| ConvGNP | $-0.24_{\pm 0.02}$ | $-0.23_{\pm 0.02}$ | $-1.00_{\pm 0.02}$ | AGNP | $-0.57_{\pm 0.02}$ | $-3.29_{\pm 0.07}$ | $-3.05_{\pm 0.06}$ |
| AGNP | $-0.41_{\pm 0.02}$ | $-1.03_{\pm 0.03}$ | $-2.61_{\pm 0.06}$ | GNP | $-0.60_{\pm 0.01}$ | $-1.70_{\pm 0.03}$ | $-1.67_{\pm 0.03}$ |
| ALNP | $-0.61_{\pm 0.02}$ | $-1.01_{\pm 0.02}$ | $-1.03_{\pm 0.02}$ | ConvCNP | $-0.68_{\pm 0.02}$ | $-0.68_{\pm 0.02}$ | $-1.39_{\pm 0.01}$ |
| ALNP (E–M) | $-0.63_{\pm 0.02}$ | $-1.24_{\pm 0.01}$ | $-1.10_{\pm 0.02}$ | ALNP | $-0.76_{\pm 0.02}$ | $-1.19_{\pm 0.02}$ | $-1.19_{\pm 0.02}$ |
| ALNP (E) | $-0.67_{\pm 0.02}$ | $-1.52_{\pm 0.02}$ | $-2.25_{\pm 0.03}$ | LNP | $-0.89_{\pm 0.01}$ | $-3.18_{\pm 0.05}$ | $-1.40_{\pm 0.02}$ |
| LNP (E–M) | $-0.68_{\pm 0.01}$ | $-3.05_{\pm 0.04}$ | $-1.75_{\pm 0.01}$ | LNP (E–M) | $-0.89_{\pm 0.01}$ | $-4.43_{\pm 0.07}$ | $-1.62_{\pm 0.02}$ |
| GNP | $-0.70_{\pm 0.02}$ | $-2.22_{\pm 0.05}$ | $-1.62_{\pm 0.04}$ | ALNP (E–M) | $-0.92_{\pm 0.02}$ | $-1.47_{\pm 0.03}$ | $-1.46_{\pm 0.03}$ |
| LNP (E) | $-0.71_{\pm 0.01}$ | $-3.71_{\pm 0.05}$ | $-2.08_{\pm 0.01}$ | LNP (E) | $-0.92_{\pm 0.01}$ | $-4.71_{\pm 0.07}$ | $-1.86_{\pm 0.01}$ |
| LNP | $-0.72_{\pm 0.01}$ | $-1.46_{\pm 0.01}$ | $-1.29_{\pm 0.02}$ | ACNP | $-0.93_{\pm 0.02}$ | $-1.42_{\pm 0.02}$ | $-1.53_{\pm 0.03}$ |
| ACNP | $-0.79_{\pm 0.02}$ | $-1.31_{\pm 0.01}$ | $-1.25_{\pm 0.01}$ | ALNP (E) | $-0.95_{\pm 0.02}$ | $-1.75_{\pm 0.04}$ | $-1.73_{\pm 0.04}$ |
| CNP (AR) | $-1.00_{\pm 0.02}$ | $-1.15_{\pm 0.02}$ | $-1.09_{\pm 0.02}$ | CNP (AR) | $-1.15_{\pm 0.02}$ | $-1.27_{\pm 0.02}$ | $-1.21_{\pm 0.02}$ |
| CNP | $-1.05_{\pm 0.02}$ | $-1.14_{\pm 0.02}$ | $-1.15_{\pm 0.02}$ | CNP | $-1.18_{\pm 0.02}$ | $-1.32_{\pm 0.03}$ | $-1.27_{\pm 0.02}$ |
| ConvLNP (E–M) | $-1.41_{\pm 0.15}$ | $-1.43_{\pm 0.13}$ | $-3.40_{\pm 0.06}$ | ConvLNP (E–M) | $-3.45_{\pm 0.07}$ | $-3.40_{\pm 0.07}$ | $-5.30_{\pm 0.07}$ |
| *Trivial* | $-1.32_{\pm 0.00}$ | $-1.32_{\pm 0.00}$ | $-1.32_{\pm 0.00}$ | *Trivial* | $-1.46_{\pm 0.00}$ | $-1.46_{\pm 0.00}$ | $-1.46_{\pm 0.00}$ |

Table 18: For the mixture synthetic experiments with two-dimensional inputs, average log-likelihoods normalised by the number of target points. Shows for one-dimensional outputs ($d_y = 1$) and two-dimensional outputs ($d_y = 2$) the performance for interpolation within the range $[-2, 2]^2$ where the models where trained ("Int."); interpolation within the range $[2, 6]^2$ which the models have never seen before ("OOID"); and extrapolation from the range $[-2, 2]^2$ to the range $[2, 6]^2$ ("Ext."). Models are ordered by interpolation performance. The latent variable models are trained and evaluated with the ELBO objective (E); trained and evaluated with the ML objective (M); and trained with the ELBO objective and evaluated with the ML objective (E–M). Trivial refers to predicting the empirical means and standard deviation of the test data. Errors indicate the central 95%-confidence interval. Numbers which are significantly best ($p < 0.05$) are boldfaced. Numbers which are very large are marked as failed with "F". Numbers which are missing could not be run.

| Mixture $d_x = 2$, $d_y = 1$ | Int. | OOID | Ext. | Mixture $d_x = 2$, $d_y = 2$ | Int. | OOID | Ext. |
|---|---|---|---|---|---|---|---|
| ConvCNP (AR) | $\mathbf{-0.10}\pm_{0.03}$ | $\mathbf{-0.10}\pm_{0.03}$ | $\mathbf{-0.34}\pm_{0.03}$ | ConvCNP (AR) | $\mathbf{-0.62}\pm_{0.01}$ | $\mathbf{-0.62}\pm_{0.01}$ | $\mathbf{-0.79}\pm_{0.01}$ |
| ConvCNP | $-0.49\pm_{0.03}$ | $-0.49\pm_{0.03}$ | $-1.45\pm_{0.02}$ | ConvGNP | $-0.74\pm_{0.01}$ | $-0.74\pm_{0.01}$ | $-1.43\pm_{0.02}$ |
| ConvGNP | $-0.50\pm_{0.02}$ | $-0.50\pm_{0.02}$ | $-1.24\pm_{0.02}$ | ConvLNP (M) | $-0.78\pm_{0.02}$ | $-0.79\pm_{0.02}$ | $-1.25\pm_{0.02}$ |
| ConvLNP (M) | $-0.57\pm_{0.02}$ | $-0.57\pm_{0.02}$ | $-1.07\pm_{0.02}$ | ConvCNP | $-0.85\pm_{0.01}$ | $-0.85\pm_{0.01}$ | $-1.50\pm_{0.02}$ |
| ConvLNP (E) | $-0.63\pm_{0.02}$ | $-0.63\pm_{0.02}$ | $-1.08\pm_{0.02}$ | ACNP (AR) | $-0.85\pm_{0.01}$ | $-4.30\pm_{0.09}$ | $-4.24\pm_{0.09}$ |
| ALNP (M) | $-0.73\pm_{0.02}$ | $-1.04\pm_{0.02}$ | $-1.05\pm_{0.02}$ | ALNP (M) | $-0.88\pm_{0.01}$ | $-1.41\pm_{0.03}$ | $-1.21\pm_{0.02}$ |
| ConvLNP (E–M) | $-0.76\pm_{0.02}$ | $-0.76\pm_{0.02}$ | $-1.37\pm_{0.02}$ | ConvLNP (E) | $-0.92\pm_{0.01}$ | $-0.92\pm_{0.01}$ | $-1.41\pm_{0.02}$ |
| ACNP (AR) | $-0.77\pm_{0.01}$ | $-1.28\pm_{0.01}$ | $-1.30\pm_{0.01}$ | AGNP | $-0.93\pm_{0.01}$ | $-1.34\pm_{0.01}$ | $-1.21\pm_{0.02}$ |
| AGNP | $-0.78\pm_{0.01}$ | $-1.32\pm_{0.02}$ | $-1.32\pm_{0.02}$ | ACNP | $-0.99\pm_{0.02}$ | $-4.19\pm_{0.10}$ | $-1.27\pm_{0.02}$ |
| ACNP | $-0.91\pm_{0.02}$ | $-1.29\pm_{0.01}$ | $-1.17\pm_{0.02}$ | ConvLNP (E–M) | $-1.05\pm_{0.01}$ | $-1.05\pm_{0.01}$ | $-1.73\pm_{0.03}$ |
| LNP (M) | $-0.91\pm_{0.02}$ | $-1.44\pm_{0.04}$ | $-1.19\pm_{0.02}$ | LNP (M) | $-1.07\pm_{0.01}$ | $-4.04\pm_{0.11}$ | $-1.31\pm_{0.02}$ |
| LNP (E–M) | $-0.92\pm_{0.02}$ | $-1.51\pm_{0.02}$ | $-1.38\pm_{0.02}$ | ALNP (E–M) | $-1.11\pm_{0.02}$ | $-2.65\pm_{0.62}$ | $-1.23\pm_{0.02}$ |
| LNP (E) | $-0.93\pm_{0.02}$ | $-1.74\pm_{0.03}$ | $-1.44\pm_{0.02}$ | ALNP (E) | $-1.12\pm_{0.02}$ | $-8.22\pm_{7.64}$ | $-1.26\pm_{0.02}$ |
| ALNP (E–M) | $-1.00\pm_{0.02}$ | $-1.07\pm_{0.02}$ | $-1.08\pm_{0.02}$ | GNP | $-1.20\pm_{0.02}$ | $-1.39\pm_{0.03}$ | $-1.21\pm_{0.02}$ |
| ALNP (E) | $-1.03\pm_{0.02}$ | $-1.08\pm_{0.02}$ | $-1.09\pm_{0.02}$ | CNP (AR) | $-1.20\pm_{0.02}$ | $-2.61\pm_{0.07}$ | $-1.32\pm_{0.02}$ |
| CNP (AR) | $-1.06\pm_{0.02}$ | $-1.07\pm_{0.02}$ | $-1.08\pm_{0.02}$ | CNP | $-1.20\pm_{0.02}$ | $-2.60\pm_{0.08}$ | $-1.23\pm_{0.02}$ |
| GNP | $-1.06\pm_{0.02}$ | $-1.09\pm_{0.02}$ | $-1.08\pm_{0.02}$ | LNP (E–M) | $-1.20\pm_{0.02}$ | $-3.23\pm_{0.10}$ | $-1.24\pm_{0.02}$ |
| CNP | $-1.07\pm_{0.02}$ | $-1.09\pm_{0.02}$ | $-1.10\pm_{0.02}$ | LNP (E) | $-1.20\pm_{0.02}$ | $-3.25\pm_{0.10}$ | $-1.24\pm_{0.02}$ |
| *Trivial* | $-1.32\pm_{0.00}$ | $-1.32\pm_{0.00}$ | $-1.32\pm_{0.00}$ | *Trivial* | $-1.46\pm_{0.00}$ | $-1.46\pm_{0.00}$ | $-1.46\pm_{0.00}$ |
| FullConvGNP | | | | FullConvGNP | | | |

# I    DETAILS OF SIM-TO-REAL TRANSFER EXPERIMENTS

## I.1    DESCRIPTION OF EXPERIMENT

Our goal will be to make predictions for the famous hare–lynx data set. The hare–lynx data set is a time series from 1845 to 1935 recording yearly population counts of a population of Snowshoe hares and a population of Canadian lynx (MacLulich, 1937). A digital version extracted from the original graph by MacLulich (1937) is available by Hundley.[7] Hundley, the author of this digital source, says that other authors caution that the hare–lynx data is actually a composition of multiple time series, and presents the data with that caution. We, therefore, also present the data with this caution. Figure 12a visualises the hare–lynx data set.

To make predictions for the hare–lynx data set, we use the Lotka–Volterra equations (Lotka, 1910; Volterra, 1926), also called the predator–prey equations. The Lotka–Volterra equations are an idealised mathematical model for the population counts of a prey population and a predator population:

$$\text{prey population:} \quad x'(t) = \alpha x(t) - \beta x(t)y(t), \tag{45}$$

$$\text{predator population:} \quad y'(t) = -\delta y(t) + \gamma x(t)y(t). \tag{46}$$

These differential equations say that the prey population naturally grows exponentially with rate $\alpha$, and that the predator population naturally decays exponentially with rate $\delta$. In addition, the predators hunt the prey. The resulting additional growth in the predator population and the resulting additional decrease in the prey population are both proportional to the product of the densities. In this idealised mathematical form, the population counts converge to a smooth, noiseless limit cycle and then perfectly track this limit cycle ever after. This is unlike real-world predator–prey population counts, which exhibit noisy behaviour and imperfect cycles. We therefore consider a stochastic version of the Lotka–Volterra equations, given by the following two coupled stochastic differential equations:

$$dX_t = \alpha X_t\,dt - \beta Y_t X_t\,dt + \sigma X_t^\nu\,dW_t^{(1)}, \tag{47}$$

$$dY_t = -\gamma X_t\,dt + \delta Y_t X_t\,dt + \sigma Y_t^\nu\,dW_t^{(2)} \tag{48}$$

where $W^{(1)}$ and $W^{(2)}$ are two independent Brownian motions. Compared to the Lotka–Volterra equations, (47) and (48) have two additional terms, $\sigma X_t^\nu\,dW_t^{(1)}$ and $\sigma Y_t^\nu\,dW_t^{(2)}$, which introduce noisy behaviour. In these terms, multiplying by $X_t^\nu$ and $Y_t^\nu$ makes the noise go to zero when $X_t$ and $Y_t$ become small, ensuring that $X_t$ and $Y_t$ remain positive. In addition, we multiply by a parameter $\sigma > 0$ to control the magnitude of the noise, and we raise $X_t$ and $Y_t$ to a power $\nu > 0$ to control how quickly the noise grows as $X_t$ and $Y_t$ grow. Namely, $X_t$ naturally grows exponentially, so, by adding noise of magnitude proportional to $X_t$, we risk large spikes in the prey population. To moderate this behaviour, we choose $\nu$ to be strictly less than one. Finally, to obtain a variety of magnitudes of population counts, we multiply the realisation with a scale $\eta$.

After simulating from (47) and (48) a few times, we settle on $\nu = \frac{1}{6}$. For the remainder of the parameters, we simply manually play around with (47) and (48), settle on parameter ranges that look reasonable, and randomly sample parameters from those intervals. Table 19 summarises the sampling distributions for all parameters of (47) and (48). Figure 12b shows four samples from the proposed stochastic model.

To generate a meta–data set, we simulate (47) and (48) on a dense grid spanning 110 years, throw away the first 10 years, and retain between 150 and 250 data points for $X_t$ and $Y_t$. The numbers of data points and the locations of the data points are sampled separately for $X_t$ and $Y_t$. Hence, whereas the hare–lynx data is regularly spaced and the populations are always simultaneously observed, our simulator generates data at arbitrary and nonsimultaneous points in time. We split these data sets into context and target sets in three different ways. To train the models, for every batch, we randomly choose one of the interpolation, forecasting or reconstruction tasks by rolling a three-sided die. We will also perform these tasks on the real hare–lynx data; in that case, for interpolation, we let the number of target points per output be between one and fifteen. The tasks on simulated and real data are similar, but slightly differ in the number of context and target points.

To deal with the positivity of population counts, we transform the marginals of all models to distributions on $(0, \infty)$ by pushing the marginals through $x \mapsto \log(1 + x)$.

---

[7]See `http://people.whitman.edu/~hundledr/courses/M250F03/LynxHare.txt`.

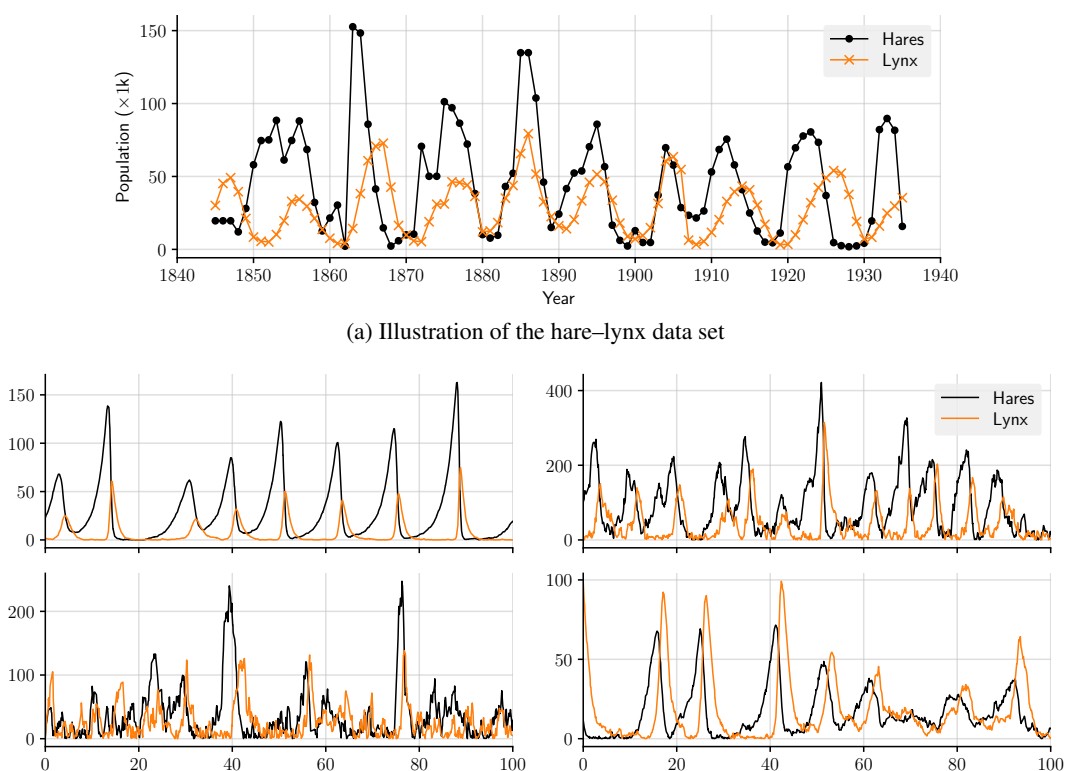

(a) Illustration of the hare–lynx data set

(b) Four samples from the proposed stochastic version of the Lotka–Volterra equations (47) and (48). The parameters of (47) and (48) are sampled according to Table 19.

Figure 12: Hare–lynx data set and proposed stochastic simulator

| Parameter | Distribution |
| --- | --- |
| Initial condition $X_{-10}$ | $\mathrm{Unif}([5, 100])$ |
| Initial condition $Y_{-10}$ | $\mathrm{Unif}([5, 100])$ |
| $\alpha$ | $\mathrm{Unif}([0.2, 0.8])$ |
| $\beta$ | $\mathrm{Unif}([0.04, 0.08])$ |
| $\gamma$ | $\mathrm{Unif}([0.8, 1.2])$ |
| $\delta$ | $\mathrm{Unif}([0.04, 0.08])$ |
| $\nu$ | Fixed to $1/6$ |
| $\sigma$ | $\mathrm{Unif}([0.5, 10])$ |
| $\eta$ | $\mathrm{Unif}([1, 5])$ |

Table 19: Sampling distributions for the parameters of the stochastic version of the Lotka–Volterra equations (47) and (48). These equations are simulated on a dense grid spanning $[-10, 100]$. The table also shows the distribution for the initial conditions at $t = -10$. To not depend too heavily on these initial conditions, the simulation results on $[-10, 0]$ are discarded.

For this experiment, the learning rate is $1 \cdot 10^{-4}$, the margin is $1$, and the points per unit is $4$. We trained the models for 200 epochs.

The convolutional models use a U-Net architecture with seven layers instead of six where, in the first layer, the stride is one instead of two. For the kernel architecture of the FullConvGNP, we reduce the points per unit and the number of channels in the U-Net by a factor two.

| | Int. (S) | For. (S) | Rec. (S) | Int. (R) | For. (R) | Rec. (R) |
|---|---|---|---|---|---|---|
| FullConvGNP | $\mathbf{-3.29}_{\pm 0.02}$ | $\mathbf{-3.46}_{\pm 0.02}$ | $-3.79_{\pm 0.02}$ | $-4.16_{\pm 0.04}$ | $\mathbf{-4.28}_{\pm 0.04}$ | $-4.45_{\pm 0.00}$ |
| ConvCNP (AR) | $\mathbf{-3.30}_{\pm 0.02}$ | $\mathbf{-3.47}_{\pm 0.02}$ | $\mathbf{-3.60}_{\pm 0.02}$ | $\mathbf{-4.10}_{\pm 0.03}$ | $\mathbf{-4.27}_{\pm 0.03}$ | $\mathbf{-4.32}_{\pm 0.01}$ |
| ConvNP (ML) | $-3.41_{\pm 0.02}$ | $-3.84_{\pm 0.02}$ | $-4.44_{\pm 0.02}$ | $\mathbf{-4.13}_{\pm 0.04}$ | $-4.45_{\pm 0.05}$ | $-4.54_{\pm 0.01}$ |
| ConvGNP | $-3.47_{\pm 0.02}$ | $-3.65_{\pm 0.02}$ | $-4.15_{\pm 0.02}$ | $-4.21_{\pm 0.05}$ | $-4.82_{\pm 0.13}$ | $-4.61_{\pm 0.01}$ |
| ConvCNP | $-3.47_{\pm 0.02}$ | $-4.06_{\pm 0.02}$ | $-4.85_{\pm 0.02}$ | $-4.17_{\pm 0.04}$ | $-4.70_{\pm 0.06}$ | $-4.97_{\pm 0.01}$ |
| ConvNP (ELBO) | $-3.77_{\pm 0.02}$ | $-3.83_{\pm 0.02}$ | $-4.12_{\pm 0.02}$ | $-5.45_{\pm 0.05}$ | $-5.47_{\pm 0.07}$ | $-6.39_{\pm 0.05}$ |
| ANP (ML) | $-4.09_{\pm 0.02}$ | $-4.32_{\pm 0.02}$ | $-4.55_{\pm 0.02}$ | $-4.31_{\pm 0.03}$ | $-4.43_{\pm 0.04}$ | $-4.49_{\pm 0.01}$ |
| ANP (ELBO–ML) | $-4.22_{\pm 0.02}$ | $-4.54_{\pm 0.02}$ | $-4.80_{\pm 0.02}$ | $-4.58_{\pm 0.11}$ | $-4.58_{\pm 0.04}$ | $-4.68_{\pm 0.01}$ |
| ACNP (AR) | $-4.23_{\pm 0.02}$ | $-4.44_{\pm 0.02}$ | $-4.58_{\pm 0.02}$ | $-4.40_{\pm 0.03}$ | $-4.55_{\pm 0.04}$ | $-4.59_{\pm 0.02}$ |
| ANP (ELBO) | $-4.32_{\pm 0.03}$ | $-4.58_{\pm 0.02}$ | $-4.82_{\pm 0.02}$ | $-4.71_{\pm 0.15}$ | $-4.63_{\pm 0.05}$ | $-4.70_{\pm 0.01}$ |
| ACNP | $-4.34_{\pm 0.02}$ | $-4.65_{\pm 0.02}$ | $-4.88_{\pm 0.02}$ | $-4.43_{\pm 0.04}$ | $-4.58_{\pm 0.04}$ | $-4.74_{\pm 0.00}$ |
| ConvNP (E.–M.) | $-6.71_{\pm 0.05}$ | $-8.44_{\pm 0.11}$ | F | $-7.20_{\pm 0.31}$ | F | F |

Table 20: Normalised log-likelihoods in the predator–prey experiments. Shows the performance for interpolation ("Int."), forecasting ("For."), and reconstruction ("Rec.") on simulated ("S") and real ("R") data. Models are ordered by interpolation performance on simulated data. The latent variable models are trained and evaluated with the ELBO objective (E); trained and evaluated with the ML objective (M); and trained with the ELBO objective and evaluated with the ML objective (E–M). Errors indicate the central 95%-confidence interval. Numbers which are significantly best ($p < 0.05$) are boldfaced. Numbers which are very large are marked as failed with "F". Numbers which are missing could not be run.

## I.2 FULL RESULTS

In Table 20, we present the full results for the sim-to-real experiments.

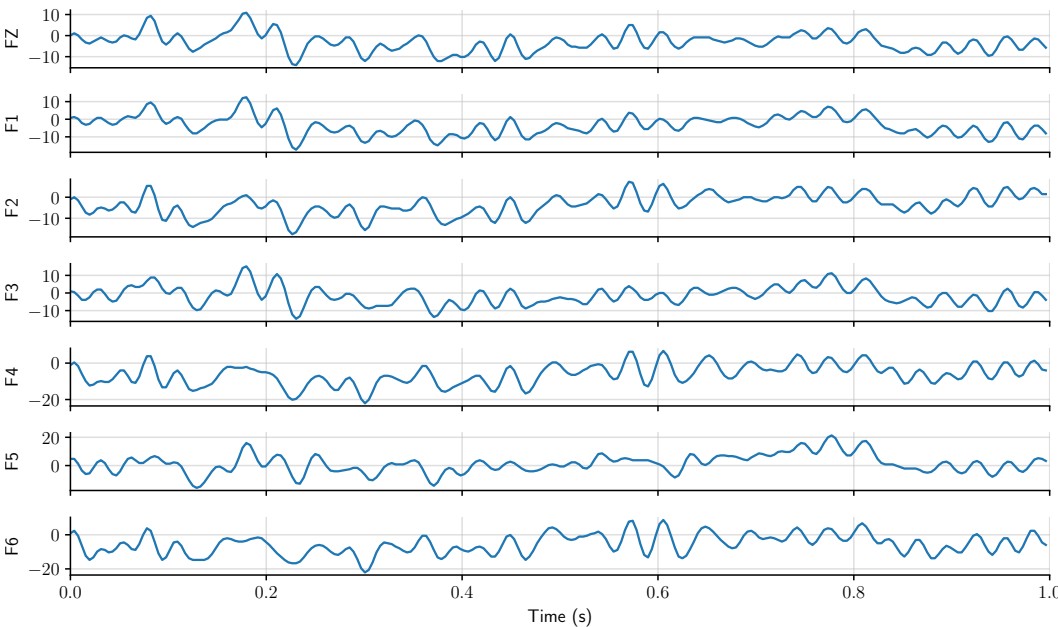

Figure 13: Example of trial in the EEG data set. Note that the signals for all electrodes appear correlated, but are subtly different.

## J  DETAILS OF ELECTROENCEPHALOGRAPHY EXPERIMENTS

We explore an electroencephalography data set collected from 122 subjects (Begleiter, 2022). There are two groups of subjects: alcoholic and control. Every subject was subjected to a single stimulus or two stimuli, and their response was measured with 64 electrodes placed on a subject's scalp. These measurements are in terms of *trials*, where a trial consists of 256 samples of the electrodes spanning one second. The data sets contains up to 120 trials for each subject. The data is available at https://archive.ics.uci.edu/ml/datasets/eeg+database and the collection is described in detail by Zhang et al. (1995). In this experiment, we focus only on seven frontal electrodes: FZ, F1, F2, F3, F4, F5, and F6. Figure 13 illustrates a trial of a subject, showing the samples for these seven electrodes.

We randomly split all subjects into three sets: an evaluation set consisting of ten subjects, a cross-validation set consisting of ten other subjects, and a training set consisting of all remaining subjects. For each of these sets, we create a meta–data set by aggregating the trials for all subjects. We split every trial into a context and target set in the same three ways as for the predator–prey experiment. First, for all seven electrodes separately, randomly designate between 50 and 256 points to be the target points and let the remainder (between 0 and 206) be the context points. This task is called *interpolation* and is the primary measure of performance. Additionally, randomly choose one of the seven electrodes and, for that choice, split the data in two exactly like for forecasting. For all other electrodes, append all data to the context set. This task is called *reconstruction* and measures the model's ability to infer a signal for one electrode from the others. We train all models on the interpolation task, and evaluate the models on the interpolation and reconstruction task.

For this experiment, the learning rate is $5 \cdot 10^{-5}$, the margin is $0.1$, and the points per unit is 256. We trained the models for 1000 epochs. For the FullConvGNP, the learning rate is $2 \cdot 10^{-4}$. The training run for the FullConvGNP was terminated after 84 hours, reaching epoch 127.

The convolutional models use a U-Net architecture where, in the first layer, the stride is one instead of two. In addition, the number of channels are adjusted as follows: the ConvCNP and ConvGNP use 128 channels, the ConvLNP uses 96 channels, and the FullConvGNP uses 64 channels. The length scales of the Gaussian kernels of the convolutional model is initialised to $0.77/256$. To scale to seven outputs, the deep set–based and attentive models reuse the same encoder for every output dimension.

## K    DETAILS OF ENVIRONMENTAL DATA ASSIMILATION EXPERIMENT

In this section we provide further details on the Antarctic cloud cover data assimilation experiment described in Section 4.4.

### K.1    DATA CONSIDERATIONS

**Data sources.** Daily-averaged cloud cover reanalysis data was obtained from ERA5 (Hersbach et al., 2020). An Antarctic land mask and elevation field was obtained from the BedMachine dataset (Morlighem, 2020).

Figure 14 shows an empircal density of the ERA5 cloud cover values calculated over the models' training period of 2000-2013. The spikes at 0 and 1 correspond largely to values of exactly 0 and exactly 1. This motivates the beta-categorical likelihood described in Section 4.4.

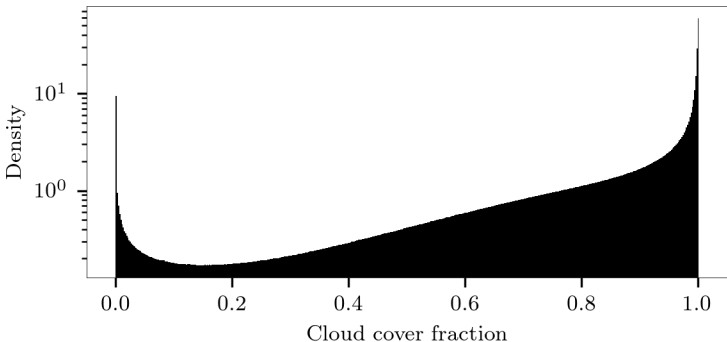

Figure 14: Empirical density of ERA5 cloud cover fraction computed over the period 2000-2013.

**Data preprocessing.** The cloud cover data and land/elevation auxiliary data were regridded from lat/lon to a Southern Hemisphere Equal Area Scalable Earth 2 (EASE2) grid at 25 km resolution and cropping to a size of $280 \times 280$. This centres the data on the South Pole.

**Data normalisation.** We normalised the data before passing it to the convolutional NP models. The cloud cover and land mask data already took appropriate normalised values in $[0, 1]$. The elevation field was normalised from metres to values in $[0, 1]$.

The input coordinates $x$ were normalised from metres to take values in $[-1, 1]$.

### K.2    MODEL CONSIDERATIONS

Here we provide details on the training procedure and architectures for each of the convolutional NP models in the Antarctic data assimilation experiment.

**Generating the training, validation, and test tasks.** Following meta-learning principles, we collect data from day $\tau$ into a task $D_\tau$. Each task $D_\tau$ was generated by first drawing the integer number of simulated cloud cover context points $N^{(c)} \sim \text{Unif}\{1, 2, \ldots 500\}$. Letting $N^{(c)}$ vary encourages the model to learn to deal with both data-sparse and data-rich scenarios. The number of target points $N^{(t)}$ was fixed to a value of 2,000.

Next, the input locations $\mathbf{x}_\tau^{(c)}$ and $\mathbf{x}_\tau^{(t)}$ were sampled uniformly at random across the entire $280 \times 280$ input space and the corresponding $\mathbf{y}_\tau$ values were sampled without observation noise.

For the training dates, the random seed used for generating $D_\tau$ is updated every epoch, allowing for an infinitely growing simulated training data set. In contrast, for the validation and test dates,

the random seeds were held fixed so that metrics computed over the validation and test sets are not stochastic.

**Training procedure.** Each model was trained for 150 epochs on 14 years of data from 2000–2013. An Adam optimiser was used with a learning rate of $5 \times 10^{-5}$ and a batch size of 2. For the loss functions we use a negative log-likelihood loss function for the ConvCNP and ConvGNP. For the ConvLNP we use the ELBO objective and fix the variance of the observation noise to $0.01$ for the first four epochs. Validation data from 2014–2017 was used for checkpointing the model weights using the per-datapoint predictive log-likelihood. The two year period of 2018–2019 data was reserved for the test set.

The time taken to train each model on a Tesla V100 GPU is as follows:

- ConvCNP: 25.0 hours,
- ConvGNP: 27.5 hours,
- ConvLNP: 43.6 hours.

**Architectures.** For each model, the U-Net component of the encoder uses $5 \times 5$ convolutional kernels with the following sequence of channel numbers (d.s. = $2 \times 2$ downsample layer, u.s. = $2 \times 2$ upsample layer): $16 \xrightarrow{\text{d.s.}} 32 \xrightarrow{\text{d.s.}} 64 \xrightarrow{\text{d.s.}} 128 \xrightarrow{\text{u.s.}} 64 \xrightarrow{\text{u.s.}} 32 \xrightarrow{\text{u.s.}} 16$. We use bilinear resize operators for the upsampling layers to fix checkerboard artifacts that we encountered when using standard zero-padding upsampling (Odena et al., 2016). We use a margin of $0.1$ and 150 points per unit for the encoder's internal discretisation. The length scales of the Gaussian kernels for both the encoder and decoder SetConv layers are set to $1/150$ and held fixed during training. These architecture choices result in a receptive field of $0.433$ in normalised input space, or roughly $1.500\,\text{km}$ in raw input space, spanning around 20% of the region in Figure 6 in either dimension.

For the ConvGNP we use 128 basis functions for the low-rank covariance parameterisation described in Markou et al. 2021.

For the ConvLNP we use an 8-dimensional latent variable and evaluate the ML objective (Foong et al., 2020) using 8 latent samples.

The number of learnable parameters for each model is as follows:

- ConvCNP: $618\,\text{k}$,
- ConvGNP: $621\,\text{k}$,
- ConvLNP: $1.234\,\text{k}$ (increase due to second UNet architecture after the latent variable).

The difference in parameters from switching to a beta-categorical likelihood from a Gaussian likelihood is negligible.

**Input data.** Each model receives two context sets as input. The first contains observations of the simulated ERA5 daily-average cloud cover. The second contains a set of 6 gridded auxiliary variables. These are elevation, land mask, $\cos(2\pi \times \text{day of year}/365)$, $\sin(2\pi \times \text{day of year}/365)$, $x_1$, and $x_2$. The elevation and land mask auxiliary fields allow the models to predict spatial non-stationarity. For example, the convolutional filters of the model's encoder could learn how cloud cover around the Antarctic coastline behaves differently to the centre of the continent. The $\cos$ and $\sin$ variables inform the model at what time of year $D_\tau$ corresponds to, helping with learning seasonal variations in the data. The $x_1$ and $x_2$ inputs again help with breaking translation equivariance in the convolutional filters by informing the model where in input space the data corresponds to.

### K.3 ANTARCTIC CLOUD COVER MODEL SAMPLES

Figure 15 gives a detailed breakdown of sample extrapolation ability, showing seven samples from the four Antarctic cloud cover models. The AR ConvCNP samples display remarkable structure and variation while still closely interpolating the context points. The samples also provide interesting scenarios in the gaps between the context points on the left hand side.

In contrast, the ConvCNP samples are incoherent, underestimating the probability of joint events.

The ConvLNP samples were generated by sampling from the latent variable and then computing the mean of the marginal distributions. As is visible in Figure 15, the ConvLNP displays low sample variance with respect to the latent variable. However, to the best of our knowledge we used a

faithful reproduction of the original ConvLNP model, so we leave a more rigorous treatment of this undesirable behaviour to future work.

For all the non-AR models, the limited receptive field size leads to samples on the right hand side becoming independent of the context observations on the left hand side after roughly $750\,\mathrm{km}$ of distance from them. This results in the models defaulting to some mean representation of the data. It is interesting to see that all the non-AR models display similar marginal mean structure, with greater cloud cover towards the centre of the continent and lower cloud cover towards the coastline, followed by increased cloud levels over the Southern Ocean.

The AR samples were drawn on a sparse 70x70 grid spanning the entire input space to save compute time. The ConvCNP model was then conditioned on these AR samples and the predictive mean was computed over the dense 280x280 target space. It took 14 minutes to generate these AR ConvCNP samples on a Tesla V100 GPU.

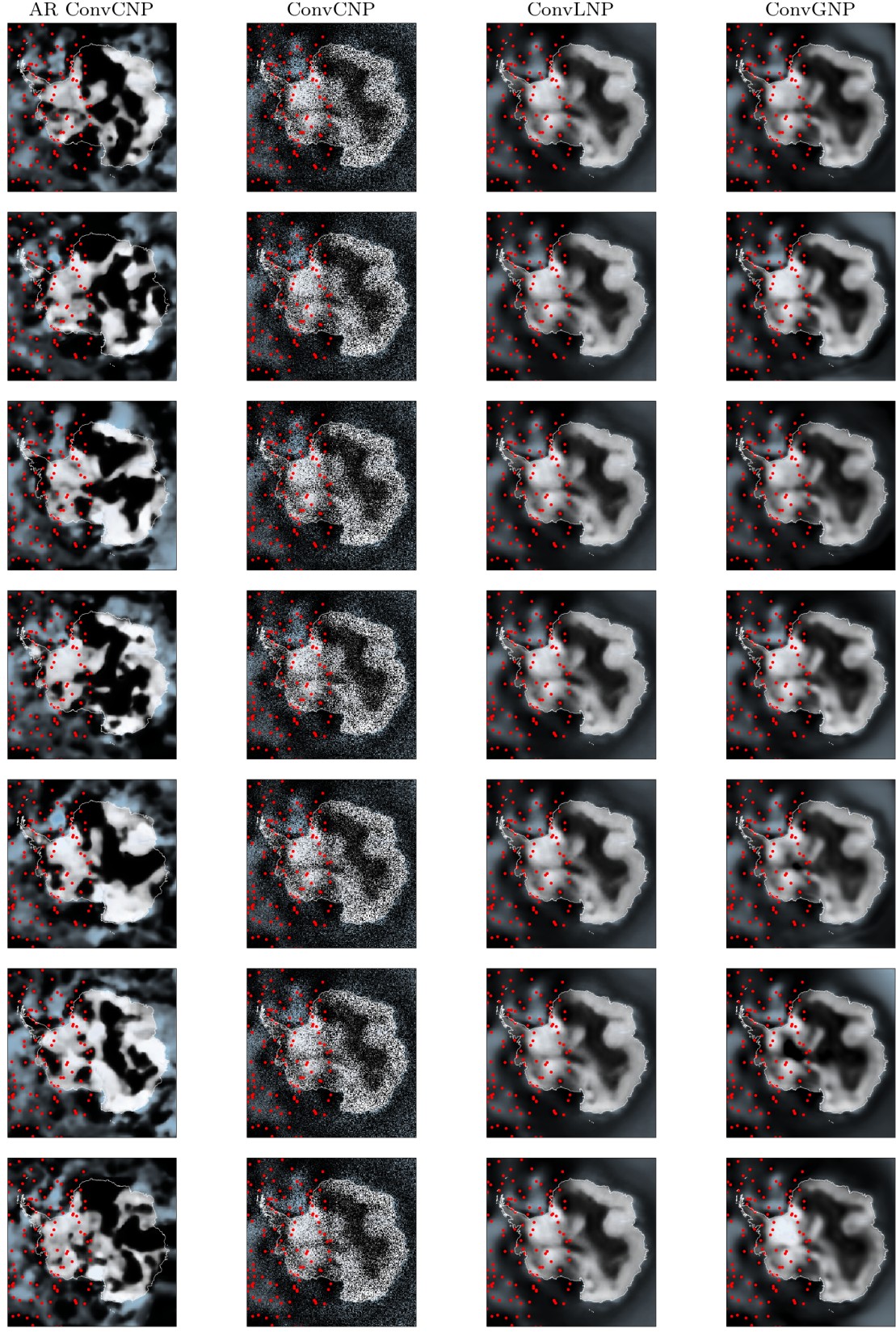

Figure 15: Seven samples from each model in the Antarctic cloud cover sample extrapolation task for 25/06/2018.

## L    DETAILS OF CLIMATE DOWNSCALING EXPERIMENTS

### L.1    DESCRIPTION OF EXPERIMENT

The MLP ConvGNP (Markou et al., 2022) can be used to successfully model dependencies between outputs in a statistical downscaling task, improving log-likelihoods over the MLP ConvCNP (Vaughan et al., 2022) and enabling coherent samples. In this experiment, we demonstrate that the AR ConvCNP can also be used for this purpose.

The goal of this experiment is to estimate the maximum daily temperature at 589 weather stations around Germany. To estimate these temperatures, we follow Vaughan et al. (2022) and use 25 coarse-grained ERA-Interim reanalysis variables (Dee et al., 2011) in combination with $1\,\text{km}$–resolution elevation data (Earth Resources Observation and Science Center, U.S. Geological Survey, U.S. Department of the Interior, 1997). We also consider a second setup where we reveal some of the weather station observations. These revealed weather station observations can be used by the models to aid downscaling performance.

The ERA-Interim reanalysis variables considered are tabulated in Table 21. In contrast to previous downscaling work, which degrade reanalysis data to between $2°$ and $2.5°$, we opt to use the ERA-Reanalysis data at the native $0.75°$ resolution, consistent with the latest high-resolution climate models with horizontal resolution ranging from $0.5°$ to $1.0°$. All variables are spatially subset to between $6°$ to $16°$ longitude and $47°$ to $55°$ latitude, covering Germany. The weather station data are a subselection from of the European Climate Assessment & Dataset (Tank et al., 2002) and are available at `https://www.ecad.eu`; we use the blended data. Like for the ERA-Reanaysis variables, we take the weather stations located within the aforementioned square. The locations of the weather stations around Germany are visualised in Figure 16. The $1\,\text{km}$–resolution elevation data is taken from the United States Geological Survey GTOPO30 elevation data set available at `https://doi.org/10.5066/F7DF6PQS`. This provides global elevation data at 30-arc second resolution, which is approximately $1\,\text{km}$.

Following the VALUE framework (Maraun et al., 2015), we consider all days of the years 1979–2008 and split these years into five folds. We use the first four folds (spanning 1979–2003) for training, holding out the last 1000 days for cross-validation; and use the fifth fold (spanning 2003–2008) for evaluation.

### L.2    MULTISCALE CONVOLUTIONAL ARCHITECTURE

Deploying the AR ConvCNP in this downscaling experiment comes with a significant challenge. Because the elevation data has a fine resolution of $1\,\text{km}$, we expect that predictions by the AR ConvCNP will vary roughly also on this length scale. In the autoregressive sampling procedure (Procedure 2.1), samples from the model will be fed back into the model. Therefore, the AR ConvCNP must handle context data which varies on a $1\,\text{km}$ spatial scale, which means that the discretisation of the AR ConvCNP must roughly be a $1\,\text{km}$–resolution grid. Unfortunately, making the discretisation this fine is prohibitively expensive and imposes prohibitive memory requirements. It is this limitation that prevents us from extending the Vaughan et al. (2022)'s MLP ConvCNP and Markou et al. (2022)'s MLP ConvGNP to include additional weather station observations. We must therefore innovate on the AR ConvCNP design to come up with a convolutional architecture that can handle such a fine discretisation at reasonable computational expense.

The architecture that we propose is a *multiscale architecture* operating on multiple spatial length scales. Let us divide the context set $D = D_{\text{lr}} \cup D_{\text{mr}} \cup D_{\text{hr}}$ into a low-resolution component $D_{\text{lr}}$, a medium-resolution component $D_{\text{mr}}$, and a high-resolution component $D_{\text{hr}}$. Let the low-resolution component $D_{\text{lr}}$ consist of features of the context set that vary on a *long* spatial length scale, the medium-resolution component $D_{\text{mr}}$ of features that vary on a *medium-range* spatial length scale, and the high-resolution component $D_{\text{hr}}$ of features that vary on a *short* spatial length scale. The central assumption of the architecture is that predictions for target points depend on precise short-length-scale details $D_{\text{hr}}$ nearby, but that this dependence weakens as we move away from the target point, starting to depend more on broad-stroke long-length-scale components $D_{\text{lr}}$. For example, predictions might depend on detailed orographic information nearby, but more on general orographic shapes farther away.

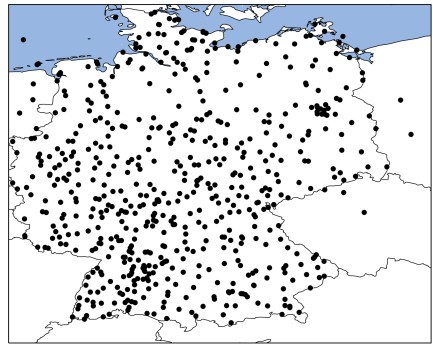

Figure 16: Locations of the 589 weather stations around Germany in the downscaling experiments.

| Variable | Level | Units |
|---|---|---|
| Surface | | |
| Maximum temperature | 2 m | degrees Celsius |
| Mean temperature | 2 m | degrees Celsius |
| Northward wind | 10 m | knots |
| Eastward wind | 10 m | knots |
| Upper atmosphere | | |
| Specific humidity | 850, 700, and 500 hPa | g/kg |
| Mean temperature | 850, 700, and 500 hPa | degrees Celsius |
| Northward wind | 850, 700, and 500 hPa | knots |
| Eastward wind | 850, 700, and 500 hPa | knots |
| Invariant | | |
| Angle of sub-grid-scale orography | surface | |
| Anisotropy of sub-grid-scale orography | surface | |
| Standard deviation of filtered sub-grid-scale orography | surface | |
| Standard deviation of orography | surface | |
| Geopotential | surface | J/kg |
| Longitude | surface | degrees |
| Latitude | surface | degrees |
| Temporal | | |
| Fractional position in the year $t$ transformed with $t \mapsto (\cos(2\pi t), \sin(2\pi t))$ | | |

Table 21: ERA-Interim reanalysis predictors.

Figure 17 depicts the multiscale architecture. The architecture is a cascade of three convolutional deep sets, parametrised by three CNNs; please see the caption. The low-resolution CNN handles the context data $D_{\mathrm{lr}}$ with a long spatial length scale. Because these features have a long spatial length scale, the CNN can get away with a low-resolution discretisation. The output of the low-resolution CNN then feeds into a medium-resolution CNN. The medium-resolution CNN handles the context data $D_{\mathrm{mr}}$ with a medium spatial length scale and has a medium-resolution discretisation. Finally, the output of the medium-resolution CNN feeds into a high-resolution CNN. This CNN handles the context data $D_{\mathrm{hr}}$ with a short spatial length scale and has a high-resolution discretisation.

The key to the computational efficiency of this architecture is that we construct the high-resolution discretisation only locally to the target points: a small square covering $0.25°$ more than the most extremal target points. If the target points are confined to a small region, then the high-resolution grid will also be small, covering only $0.25°$ more than that region. Crucially, the high-resolution

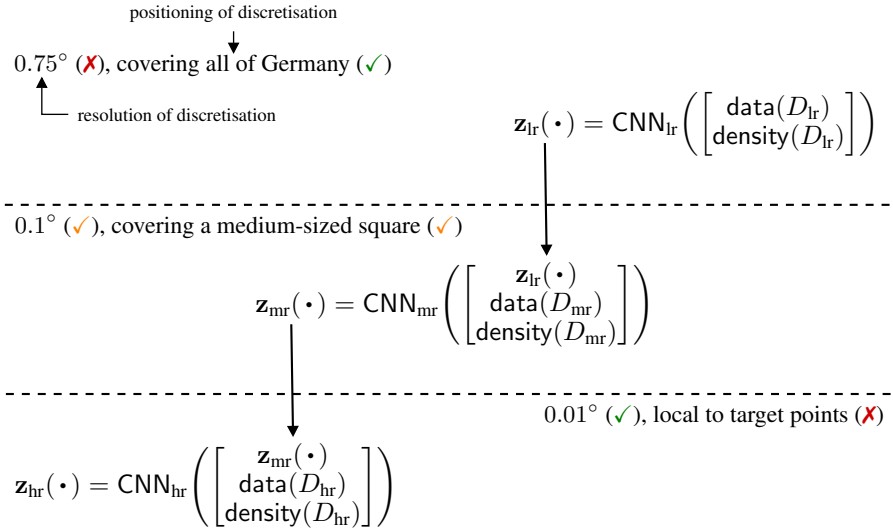

Figure 17: Multiscale architecture for the AR ConvCNP. A cascade of three convolutional deep sets (Gordon et al., 2020) representing a low-resolution, medium-resolution, and high-resolution component. Shows the resolution and positioning of the internal discretisation for every convolutional deep set. The context set $D = D_{\text{lr}} \cup D_{\text{mr}} \cup D_{\text{hr}}$ is also divided into a low-resolution $D_{\text{lr}}$, medium-resolution $D_{\text{mr}}$, and high-resolution component $D_{\text{hr}}$. The low-resolution context data $D_{\text{lr}}$ consists of the 25 coarse-grained ERA-Interim reanalysis variables. The medium-resolution $D_{\text{mr}}$ and high-resolution context data $D_{\text{hr}}$ both consist of the station observations and the $1 \text{ km}$–resolution elevation data. The functions $\text{data}(D)$ and $\text{density}(D)$ produce respectively the data channel and density channel for context data $D$; see Gordon et al. (2020). The variables $\mathbf{z}_{\text{lr}}(\cdot)$, $\mathbf{z}_{\text{mr}}(\cdot)$, and $\mathbf{z}_{\text{hr}}(\cdot)$ represent intermediate representations as continuous functions, and the maps $\text{CNN}_{\text{lr}}$, $\text{CNN}_{\text{mr}}$, and $\text{CNN}_{\text{hr}}$ are translation-equivariant maps between functions on $\mathcal{X}$. Following the construction of the ConvCNP (Gordon et al., 2020), these maps are all implemented with convolutional neural networks (CNN) using a discretisation. For $\text{CNN}_{\text{lr}}$, the internal discretisation is the $0.75°$-resolution grid corresponding to the 25 coarse-grained ERA-Interim reanalysis variables. For $\text{CNN}_{\text{mr}}$, the internal discretisation is a $0.1°$-resolution grid spanning $5°$ more than the most extremal target inputs; the discretisation does *not* depend on the context set. For $\text{CNN}_{\text{hr}}$, the internal discretisation is a $0.01°$-resolution grid spanning $0.25°$ more than the most extremal target inputs; the discretisation also does *not* depend on the context set.

grid will not be constructed over all of Germany, like it would if we were to more naively apply the ConvCNP with a high-resolution discretisation, incurring prohitive computational cost. Even though the high-resolution grid is only constructed locally to the target points, the model can still capture long-range dependencies via the medium-resolution and low-resolution grids. Namely, the medium-resolution grid is a square covering $5°$ more than the most extremal target points, and the low-resolution grid covers all of Germany; see Figure 17. To utilise this computational gain, the target points must be confined to a small region. This perfectly synergises with the autoregressive sampling procedure (Procedure 2.1), because this procedure evaluates the model one target point at a time. The training procedure, however, must be adapted. During training, we subsample the target points to ensure that the target set is always confined to a small square, which is described in Appendix L.4.

During the autoregressive sampling procedure, the AR ConvCNP takes in earlier AR samples from the model. In the architectures of the MLP ConvCNP and MLP ConvGNP, these is no natural context data to which these samples can be appended. Therefore, in addition to the ERA-Interim reanalysis variables and the elevation data, we also let the AR ConvCNP take in observed weather stations as context data. We will append the earlier AR samples to these weather station context data. To have the model make appropriate use of the weather station context set, we must randomly divide the weather stations observations over the context and target set, which we describe in Appendix L.4. We let the low-resolution context data $D_{\text{lr}}$ consist of the 25 coarse-grained ERA-Interim reanalysis variables, and let the medium-resolution $D_{\text{mr}}$ and high-resolution context data $D_{\text{hr}}$ both consist of the weather station observations (and earlier AR samples) and the $1 \text{ km}$–resolution elevation data. When

the $1\,\mathrm{km}$–resolution data is fed to the medium-resolution CNN, the data loses some detail, because the internal discretisation of the medium-resolution CNN is coarser than the data; however, when it is fed to the high-resolution CNN, the data retains its detail. The same holds for the weather station observations (and earlier AR samples).

### L.3 ARCHITECTURES

**MLP ConvCNP and MLP ConvGNP (Vaughan et al., 2022; Markou et al., 2022).** The MLP ConvCNP and MLP ConvGNP are a respectively a ConvCNP and ConvGNP where the decoder $\mathrm{dec}_\theta = \mathrm{fuse}_\theta \circ \mathrm{dec}'_\theta$ is decomposed into a convolutional architecture $\mathrm{dec}'_\theta$ followed by a pointwise MLP $\mathrm{fuse}_\theta$:

$$\mathrm{fuse}_\theta(\mathbf{z}(\cdot)) = \mathrm{MLP}_\theta(\mathbf{z}(\cdot), \mathrm{elevation}(\cdot)). \tag{49}$$

In this architecture, the ERA-Interim variables are incorporated via the convolutional architecture, producing the encoding $\mathbf{z}(\cdot)$. On the other hand, as (49) shows, the $1\,\mathrm{km}$–resolution elevation data is included via the pointwise MLP $\mathrm{fuse}_\theta$.

Parametrise $\mathrm{dec}'_\theta$ with a seven-layer residual convolutional neural network (He et al., 2016). Every residual layer involves one depthwise-separable convolutional filter (Chollet, 2017) with kernel size three followed by a pointwise MLP. Every layer has 128 channels, and the network also outputs 128 channels. The discretisation for $\mathrm{dec}'_\theta$ is the grid of the ERA-Interim reanalysis variables. Parametrise $\mathrm{fuse}_\theta$ with a three-hidden-layer MLP of width 128.

**AR ConvCNP.** The AR ConvCNP does not use the pointwise MLP $\mathrm{fuse}_\theta$ to incorporate the $1\,\mathrm{km}$–resolution elevation data. Instead, it is a normal ConvCNP where the convolutional architecture is implemented by the multi-scale architecture described in Figure 17.

Parametrise $\mathrm{CNN}_{\mathrm{lr}}$ with a depthwise-separable residual convolutional neural network like in the MLP ConvCNP and MLP ConvGNP, but use six layers instead of seven. Let $\mathrm{CNN}_{\mathrm{lr}}$ output 64 channels. The discretisation for $\mathrm{CNN}_{\mathrm{lr}}$ is the grid of the ERA-Interim reanalysis variables. Parametrise $\mathrm{CNN}_{\mathrm{mr}}$ with a U-Net (Ronneberger et al., 2015) using an architecture similar to what we have been using. Before the U-turn, let the U-Net have five convolutional layers with kernel size five, stride one for the first layer and stride two afterwards, 64 output channels for the first three layers and 128 output channels afterwards. After the U-turn, instead of using transposed convolutions, use regular convolutions combined with an upsampling layer using bilinear interpolation. Let $\mathrm{CNN}_{\mathrm{mr}}$ output 64 channels. The receptive field of $\mathrm{CNN}_{\mathrm{mr}}$ is approximately $10°$. The discretisation for $\mathrm{CNN}_{\mathrm{mr}}$ is centred around the target points with margin $5°$. Parametrise $\mathrm{CNN}_{\mathrm{hr}}$ with a U-Net like for $\mathrm{CNN}_{\mathrm{hr}}$, but with four convolutional layers before the U-turn. The receptive field of $\mathrm{CNN}_{\mathrm{hr}}$ is approximately $0.5°$. The discretisation for $\mathrm{CNN}_{\mathrm{hr}}$ is centred around the target points with margin $0.25°$.

### L.4 TRAINING DETAILS

**MLP ConvCNP and MLP ConvGNP.** The MLP ConvCNP and MLP ConvGNP are trained with learning rate $2.5 \cdot 10^{-5}$ for 500 epochs. For the MLP ConvGNP, to encourage the covariance to fit, we fix the variance of the decoder to $10^{-4}\mathbf{I}$ for the first ten epochs.

**AR ConvCNP.** The AR ConvCNP is trained with learning rate $1 \cdot 10^{-5}$ for 500 epochs. During training and cross-validation, the target points are subsampled to lie in a $3° \times 3°$ square. For training, the number of target points is ensured to be at least ten; and for cross-validation, at least one. The size of the cross-validation set is increased ten fold.

**Sampling of data.** For the MLP ConvCNP and MLP ConvGNP, since these architectures cannot take in weather station observations, all weather stations are used as context data. For the AR ConvCNP, a data set is split into a context and target set by randomly selecting $n$ points as context points and letting the remainder be target points. Specifically, the number of context points $n$ is sampled from $p(n) \propto e^{-0.01n}$. This splitting is done after subsampling the $3° \times 3°$ square.

## M ALTERNATE AR PROCEDURE WITH AUXILIARY DATA

We propose an additional procedure which uses autoregressive sampling with auxiliary data to generate more expressive marginal predictives. The input points of the auxiliary data are chosen randomly, and then sampled autoregressively before sampling the target points. Finally, we discard the sampled values for the auxiliary data, but retain the samples for the target points. Adding auxiliary points in this way allows the model to roll out autoregressively with more steps, even if the target set is small (or just a single point). We describe the procedure below:

**Procedure M.1** (Autoregressive application of neural process with auxiliary data). For a neural process $\pi_\theta$, context set $D^{(c)} = (\mathbf{x}^{(c)}, \mathbf{y}^{(c)})$, a target input $x^{(t)}$, a distribution $r$ over $\mathcal{X}$, a number of auxiliary data points $R \in \mathbb{N}$, and a number of trajectories $M \in \mathbb{N}$, let $\text{AuxAR}_{\mathbf{x}^{(t)}}(\pi_\theta, D^{(c)}, r, R, M)$ be the distribution defined as follows. We first autoregressively sample the auxiliary data trajectories at random locations sampled from $r$:

$$\text{for } j = 1, \ldots, M \text{ and } \ell = 1, \ldots, R, \qquad x_\ell^{(\text{aux},j)} \sim r, \tag{50}$$

$$\text{for } j = 1, \ldots, M, \qquad \mathbf{y}^{(\text{aux},j)} \sim \text{AR}_{\mathbf{x}^{(\text{aux},j)}}(\pi_\theta, D^{(c)}). \tag{51}$$

Next, conditioned on the auxiliary data, we sample the target point of interest to make predictions. We then marginalise out the auxiliary data by averaging over the $M$ trajectories:

$$y^{(t)} \sim \frac{1}{M} \sum_{j=1}^{M} P_{x^{(t)}}(\mathbf{x}^{(c)} \oplus \mathbf{x}^{(\text{aux},j)}, \mathbf{y}^{(c)} \oplus \mathbf{y}^{(\text{aux},j)}). \tag{52}$$

This procedure introduces three hyperparameters: the distribution from which to draw inputs $r$, the length of trajectories $R$, and the number of trajectories to sample $M$.

In the following experiments, we set the distribution $r$ to be uniform over the training domain with no dependence on the context set or target point of interest: $r = \text{Unif}([b, h])$, where $b$ and $h$ are the lower and upper bounds of the training domain, respectively. One could experiment with other choices for the distribution $r$. The trajectory length $R$ is chosen between $0^8$ and 8, and the number of trajectories $M$ is chosen between 1 and 128.

### M.1 GENERATED DATA

We create three data generating processes for our experiments: a mixture of functions, random sawtooth functions, and random audio-like functions. The first two experiments have multi-modal true marginals, whereas the last has heavy-tailed marginals.

**Function mixture.** The function mixture data are generated by choosing one of the following three functions, the first with a probability of $\frac{1}{4}$, the second with a probability of $\frac{1}{2}$, and the third with a probability of $\frac{1}{4}$:

$$y = x^2 + \epsilon, \qquad \epsilon \sim \mathcal{N}(0, 0.25), \tag{53}$$

$$y = x + \epsilon, \qquad \epsilon \sim \mathcal{N}(0, 0.0625), \tag{54}$$

$$y = -x + \epsilon, \qquad \epsilon \sim \mathcal{N}(0, 0.25). \tag{55}$$

**Sawtooth.** The sawtooth data are generated from the following function:

$$y(x) = [\omega(dx - \phi)] \bmod 1 \tag{56}$$

We sample the frequency $\omega \sim \text{Unif}([3, 5])$, the direction $d$ as either $-1$ or $1$ with equal probability, and the shift as $\phi \sim \text{Unif}([\frac{1}{\omega}, 1])$.

**Synthetic Audio.** Synthetic audio data are generated by convolving a Dirac comb[9] with a truncated decaying sum of sinusoids:

$$s(t) = \begin{cases} e^{-\frac{t}{\tau}} [\sin(\omega_1 t) + \sin(\omega_2 t)] & \text{for } 0 \leq t < T, \\ 0 & \text{otherwise}, \end{cases} \tag{57}$$

$$f(x) = \text{Comb}_T(x) * s(x), \tag{58}$$

$$y = f(x) + \epsilon \quad \text{where} \quad \epsilon \sim \mathcal{N}(0, 0.001) \tag{59}$$

---

[8] A trajectory length of 0 is equivalent to the standard test-time procedure.

[9] The Dirac comb is defined as $\text{Comb}_T(t) := \sum_{k=-\infty}^{\infty} \delta(t - kT)$ for given period $P$.

| Experiment | ConvCNP | ConvCNP (AuxAR) |
|---|---|---|
| Function Mixture | $-0.63 \pm 0.20$ | $\mathbf{-0.46} \pm 0.15$ |
| Sawtooth | $1.46 \pm 0.30$ | $1.64 \pm 0.28$ |
| Synthetic Audio | $0.12 \pm 0.14$ | $\mathbf{0.55} \pm 0.10$ |

Table 22: Using autoregressive sampling for marginal approximation improves held-out log-likelihoods on all experiments. Values are normalized by the number of target points. Values which are significantly best ($p < 0.05$) are shown in bold.

where
$$\omega_1, \omega_2 \sim \mathrm{Unif}([50, 70]), \quad T \sim \mathrm{Unif}([0.75, 1.25]), \quad \tau \sim \mathrm{Unif}([0.1, 0.3]). \tag{60}$$
We truncate the waves up to the period length, because otherwise the convolution with the Dirac comb would lead to increasing amplitude, resulting in a non-stationary process.

## M.2 TRAINING

For the function mixture experiment, no training is required because we use the analytically derived ideal CNP $\pi_\infty$ as our model. See Section 2.

For the sawtooth and synthetic audio experiments, we train ConvCNP models. We train each model for 100 epochs using 1024 batches per epoch with a batch size of 16. We discretise the encoder by evaluating 64 points per unit. We use a margin of 0.1, and a stride length of 2 for each of the 6 layers of the U-Net. Each layer has 64 channels. The receptive field size from this combination of parameters is 6.953.

During training, we sample a number of context points between uniformly at random from $\{0,\dots,75\}$, and we sample exactly 100 target points. The context points and target points are sampled uniformly from $[-2, 2]$. We use the Adam optimizer with a learning rate of $3 \times 10^{-4}$.

## M.3 RESULTS

| Model | 0 | 1 | 2 | 4 | 8 | 16 |
|---|---|---|---|---|---|---|
| ConvCNP | $-0.47_{\pm 0.07}$ | $-0.45_{\pm 0.18}$ | $-0.24_{\pm 0.12}$ | $-0.12_{\pm 0.12}$ | $0.36_{\pm 0.15}$ | $1.07_{\pm 0.18}$ |
| ConvCNP (AuxAR) | $\mathbf{0.02}_{\pm 0.11}$ | $\mathbf{0.19}_{\pm 0.11}$ | $\mathbf{0.24}_{\pm 0.12}$ | $\mathbf{0.31}_{\pm 0.11}$ | $\mathbf{0.74}_{\pm 0.10}$ | $1.27_{\pm 0.12}$ |

Table 23: Log likelihood values for varying context sizes using ConvCNP and ConvCNP (AuxAR) using the function mixture data generator. Column headers indicate the context set size. Log-likelihoods are shown in bold when they are significantly best ($p < 0.05$). Column headers are context sizes. Errors indicate central 95% confidence interval.

| Model | 0 | 1 | 2 | 4 | 8 | 16 | 32 |
|---|---|---|---|---|---|---|---|
| ConvCNP | $\mathbf{-0.18}_{\pm 0.00}$ | $-0.14_{\pm 0.02}$ | $-0.02_{\pm 0.05}$ | $0.42_{\pm 0.19}$ | $1.89_{\pm 0.23}$ | $3.06_{\pm 0.13}$ | $3.54_{\pm 0.08}$ |
| ConvCNP (AuxAR) | $-0.21_{\pm 0.02}$ | $-0.09_{\pm 0.04}$ | $0.08_{\pm 0.08}$ | $\mathbf{0.95}_{\pm 0.16}$ | $\mathbf{2.37}_{\pm 0.15}$ | $3.03_{\pm 0.13}$ | $3.51_{\pm 0.08}$ |

Table 24: Log likelihood values for varying context sizes using ConvCNP and ConvCNP (AuxAR) using the sawtooth data generator. Column headers indicate the context set size.. Log-likelihoods are shown in bold when they are significantly best ($p < 0.05$). Column headers are context sizes. Errors indicate central 95% confidence interval.

In Table 22, we see that using this procedure improves the held-out log-likelihoods for all of the experiments. We can better understand the utility of this method by observing the performance for

| Model | 0 | 1 | 2 |
|---|---|---|---|
| ConvCNP | $-1.40_{\pm 0.11}$ | $-0.91_{\pm 0.28}$ | $-0.36_{\pm 0.20}$ |
| ConvCNP (AuxAR) | $\mathbf{-1.05}_{\pm 0.11}$ | $-0.63_{\pm 0.23}$ | $-0.29_{\pm 0.17}$ |

Table 25: Log likelihood values for varying context sizes using ConvCNP and ConvCNP (AuxAR) using the function mixture data generator. Column headers indicate the context set size. Log-likelihoods are shown in bold when they are significantly best ($p < 0.05$). Column headers are context sizes. Errors indicate central 95% confidence interval.

different context sizes in Tables 24 and 23. For example, the sawtooth data results in Table 24 show that, for context set sizes 16, the AR method provides no benefits. The increased flexibility of this AR method is not needed in this case — a Gaussian predictive models the marginal sufficiently well. For context set sizes of 4 and 8, on the other hand, we see significant improvements using the AR method. Similarly, for the synthetic audio data results in Table 23, we see improvements using the ConvCNP (AuxAR) for all context set sizes except 16.

The autoregressive sampling with auxiliary data method shown here shows promise for improving modeling of processes with multi-modal and heavy-tailed marginal distributions — all with no changes to the training procedure. The scenarios where this method are most useful are highly contingent upon the context set size, because of its impact how well the marginals are modeled by Gaussians.

