# OpenReview forum: "Autoregressive Conditional Neural Processes"
_ICLR.cc/2023/Conference — ICLR 2023 poster_

### Official Review · Reviewer_WrGZ · 2022-10-21

**Confidence:** 3
**Correctness:** 3
**Technical Novelty And Significance:** 3
**Empirical Novelty And Significance:** 4
**Recommendation:** 8

**Clarity, Quality, Novelty And Reproducibility:**

**Novelty:** The paper fills a blank spot in the literature: as discussed in the paper, AR schemes have been used for visualization of CNP predictions (e.g., [1]), but there exists no thourough investigation of the theoretical and empirical properties of AR-induced predictive distributions.

**Quality/Clarity:** The paper is very well structured and well written. The visualizations are of high quality and help the reader grasp the concepts. I did not check all the math in depth, but the theoretical exposition appears concise and correct. The empirical evaluation is extensive and contains comparisons on a range of synthetic and interesting real-world datasets with the relevant SOTA methods.

**Reproducibility:** The paper contains detailed descriptions of the datasets and experimental settings. Nevertheless, I have some open questions regarding the evaluation methodology, cf. below. Unfortunately, the github links are not valid, so the source code cannot be accessed.

**Open questions:** I ask the authors to elaborate on the following points:

- As discussed in the paper, the AR scheme leads to predictive distributions that are not permutation invariant w.r.t. set of input locations. I would like to see an extended discussion of this point, as I feel that the order can be crucial for the quality of the AR predictive distribution. In particular, (i) how would, e.g., Fig. 3 look like if the targets are fed into the AR-scheme from *right to left* instead of from *left to right*?, (ii) which order was chosen in the experiments?, (iii) how to choose the order in practice?/how to determine an "optimal" order?, (iv) how sensitive is the predictive performance to the order (in the author’s experience)?
- If I understand Sec. E correctly, you sampled new tasks for each training epoch. While I know that this is a standard approach, I would be interested in a comparison between CNP and AR-CNP when trained on a meta dataset with *finitely* many tasks, as is done, e.g., in [1]. Preferably, I would be interested in results on a range of different meta data set sizes in terms of the number of tasks (results on a simple 1D experiment that can be quickly trained would be enough). I think this is an interesting evaluation, because it sheds light on the question whether a lot of training data is required for the resulting model to exhibit good performance when evaluated using the AR scheme.

[1] Volpp et al., "Bayesian Context Aggregation for Neural Processes", ICLR 2021


**Strength And Weaknesses:**

Strengths:

- The paper studies an interesting and relevant problem.
- The paper is very well written and easy to follow.
- The experimental section is very well fleshed out.

Weaknesses: there are only minor open points that I encourage the authors to elaborate on, cf. below.


**Summary Of The Paper:**

The paper investigates an autoregressive (AR) evaluation scheme for Conditional Neural Processes (CNP) and its variants. The work is motivated by the fact that standard CNP does not model correlations between prediction targets. Incorporating these correlations is tackled in prior work by extending the *model*, which typically leads to complex training procedures. In contrast, the proposed method changes the *evaluation procedure* by defining an autoregressive, non-factorized joint distribution over targets, while leaving the model/training procedure of CNPs untouched. In a range of experiments, the paper shows that AR-CNPs show strong performance.

**Summary Of The Review:**

The paper studies an interesting way of constructing expressive predictive distributions for CNP models at test time which will be of interest for the community. Both the theoretical and empirical parts are sound and well fleshed out. I have some open questions that I ask the authors to elaborate on, so I might further increase my score after the discussion period. Nevertheless, I think the paper is ready for publication at ICLR, and, thus, I recommend acceptance.


_________________________________________


Update after rebuttal: I increase my score, cf. my comment below.

---

> ### Author Response · Authors · 2022-11-12
> **Response to Reviewer WrGZ (Part 3)**
>
> # Meta Data–Sets with Finitely Many Tasks
>
> We sample new tasks for every training epoch only in the synthetic experiments (Section 4.1) and in the sim-to-real transfer experiment (Section 4.2). In the real-world data experiments with EEG (Section 4.3) and climate data (Section 4.4), the meta-data sets consist of finitely many tasks.
>
> The reason why we chose to sample new tasks for every training epoch in the synthetic experiments (Section 4.1) is that we intended an equal-footed comparison between the best possible performance of a range of architectures. In particular, we wanted the models to not be limited by training set size. Similarly, in the sim-to-real transfer experiment (Section 4.2), the goal is to train a model as well as possible on the simulator. Since running the simulator is cheap, we obtain the best performance by generating new data every epoch.
>
> For the EEG (Section 4.3) and two climate experiments (Section 4.4), the setting is completely different. In these experiments, we are given finite data sets and are unable to obtain more data, similar to other realistic scenarios. We believe that these experiments are the setting of finitely many tasks that you’re referring to.
>
> Our observation is that, in the experiments that we consider, regardless of whether the CNP is training on infinite or finite data, running the CNP in AR mode consistently and significantly improves performance.
>
> # Conclusion
>
> To recapitulate, our experience is that the order of the target point does not really matter, as long as the number/density of target points does not exceed that at training time. We will include an appendix which compiles all details from the above discussion and also includes an experiment running the model with a random ordering, left to right, and right to left. In addition, whereas the synthetic (Section 4.1) and sim-to-real transfer (Section 4.2) experiments generate new data every training epoch, the EEG (Section 4.3) and two climate experiments (Section 4.4) use meta–data sets with only finitely many tasks.
>
> **If you feel like we have satisfactorily answered your questions, we would invite you to consider increasing our score.**

---

> > ### Comment · Reviewer_WrGZ · 2022-12-02
> > **Thanks for the updates!**
> >
> > Thanks for your clarifications and the updates! I think this further strengthens the submission. While reviewer K56J correctly points out that AR sampling schemes are not new, I believe that this paper still is a valuable contribution to the NP literature as it includes a proper analysis of the theoretical and empirical properties of AR sampling which will be useful for future research and applications of NPs. Thus, I increase my score.

---

> ### Author Response · Authors · 2022-11-12
> **Response to Reviewer WrGZ (Part 2)**
>
> ## (iii) How to choose the order in practice?/how to determine an "optimal" order?
>
> Our experience is that the order of the target point does not really matter, as long as _the number of target points does not exceed the numbers seen at training time_. For CNPs, this means that the _total number of target points_ should not exceed that at training time. For ConvCNPs, this means that the _density of target points_ should not exceed that at training time. In particular, this means that AR ConvCNPs can be run at arbitrarily many target points, many more than the number of target points observed at training time, as long as the density of these points does not significantly exceed the density of the training data.
>
> There is some intuition that can be used to determine how “good” an ordering is. In particular, for a given ordering of the target points, the ordering will produce high quality samples and predictions if the conditional distributions of the AR factorisation match the corresponding conditional distributions of the true posterior. Since the conditionals of the AR CNP are typically  Gaussian by design, this means that the ordering is “good” if the corresponding conditionals of the true posterior are close to Gaussian.
>
> So when is a conditional of the posterior close to Gaussian? Let us assume that the true underlying process is a sum of a non-Gaussian process (constituting epistemic uncertainty) and independent Gaussian noise (constituting aleatoric uncertainty). Generally, a conditional will have both epistemic and aleatoric uncertainty, so a Gaussian will be a bad fit. _However_, as we condition the conditionals of the true generative process on more and more data, the underlying function will be pinned down more and more accurately, meaning that the conditional will consist mostly of aleatoric uncertainty, which is Gaussian. Therefore, as we condition on more and more data, we expect the conditionals to become more and more Gaussian.
>
> Note that this reasoning is in line with our response for (i). There we observed that, in the AR sampling process, the first few predictions may be poor, but, as we perform more and more AR steps, the predictions tend to become more and more non-Gaussian and therefore more and more accurate. To prevent pathological behaviour from a poor ordering, we propose to “average out” the effects from the first few AR steps by using a random ordering, and observe good results in practice.
>
> To summarise: An ordering is “good” if the corresponding conditionals of the true posterior are also close to Gaussian. Under the assumption of a non-Gaussian process with Gaussian noise, conditionals tend to be close to Gaussian if they are conditioned on many data points. As a consequence, the earlier conditionals in the AR factorisation tend to be poor fits to the ground truth posterior, whereas later conditionals tend to produce better fits. Our recommendation is to choose a random ordering to “average out” the effects from the first few AR steps.
>
> ##  (iv) How sensitive is the predictive performance to the order (in the author’s experience)?
>
> In our experience, the predictive performance is insensitive to the order of the target points, as long as the number/density of target points does not exceed that at training time. Please see the previous subsection for more detail.

---

> ### Author Response · Authors · 2022-11-12
> **Response to Reviewer WrGZ (Part 1)**
>
> Thank you for your careful reading and assessment of our submission. We are excited to hear that you think our submission is well written and fills a blank spot in the literature.
>
> Below, we carefully address your two open questions in turn.
>
> On the ordering of the target inputs, we agree that this is an important matter. **Since the main body of our manuscript is already packed with details, we will compile all details from the discussion below into a new appendix dedicated to the effect of the order and number of AR points on predictive performance.** In this appendix, we will include an ablation experiment where we run an AR CNP with three different orderings: random, left to right, and right to left.
>
> # Ordering of Target Inputs
>
> In all experiments, we use random orderings of the target inputs. This is stated in the last sentence of Appendix D (descriptions of the approaches/models). We will add a sentence to the main body for better visibility of this important detail.
>
> We now answer your subquestions (i), (ii), (iii), and (iv).
>
> ## (i) How would, e.g., Fig. 3 look like if the targets are fed into the AR-scheme from _right to left_ instead of from _left to right_?
>
> When evaluating a CNP with Gaussian marginals in AR mode, every conditional prediction in the AR process is Gaussian. Let us consider the process of producing an AR sample. For the first target input $x_1$, we run the CNP forward to obtain a distribution for the corresponding target output $y_1$. Note that, in reality, the true posterior most likely is non-Gaussian, which means that the predictive for the first prediction for the first target point may be poor. Nevertheless, we sample this Gaussian, append the sample $(x_1, y_1)$ to the context set, and run the CNP forward again. Because we now feed the earlier sample $y_1$ through the non-linear network, the marginal predictive for the next target output $y_2$ (having integrated out $y_1$) is non-Gaussian. As we perform more AR steps, the marginal predictions of later points become increasingly non-Gaussian, increasing the model’s flexibility.
>
> We see that, for a given ordering of the target inputs, the prediction for the first target input is likely poor (because it is Gaussian), and (in the best case) the predictions become more and more accurate as we take more AR steps (because they become more and more non-Gaussian). This is exactly what is happening in Figure 3: the left prediction is Gaussian and therefore a poor approximation, and, as we go to the right and take more and more AR steps, the prediction becomes more and more non-Gaussian and therefore more accurate.
>
> If we were to feed the target inputs in right to left, then the same phenomenon would happen. The right prediction would be a Gaussian and a very poor approximation, and, as we go to the left and take more AR steps, the prediction would become more and more non-Gaussian and therefore more accurate.
>
> This behaviour confirms a general trend that we have observed throughout all our experiments: the first few AR samples tend to be poor, but, after a few AR steps, the predictions become non-Gaussian and form a good approximation. With a random ordering for every sample, this means that the poor first few AR steps are at different and random x locations for every sample; after the first few AR steps, all remaining AR steps tend to form a good approximation.
>
> ## (ii) Which order was chosen in the experiments?
>
> As mentioned above, we used random orderings for all experiments.

---

> ### Author Response · Authors · 2022-11-15
> **Reminder and updates to our paper**
>
> Dear Reviewer WrGZ,
>
> This is a brief message to remind you that we're nearing the end of the discussion period. We are eager to engage into a discussion, so we would very much appreciate it if you would reply to our response.
>
> We would also like to let you know that, in addition to the earlier amendments, we have inserted two new appendices: Appendix D and E. Appendix D illustrates the AR sampling process and the procedure the obtain smooth samples. Appendix E describes, analyses, and justifies our choice of ordering of the target points.
>
> Thank you again for taking the time to review our submission.

---

> ### Author Response · Authors · 2022-11-19
> **A further update to the paper**
>
> Dear Reviewer WrGZ,
>
> We just wanted to let you know that we have added a further section in our appendix which discusses how sensitive the predictive performance of AR CNPs is to the AR ordering. Please see Figure 9 and the accompanying discussion in Appendix D.4.
>
> Thank you again for your time and effort in reviewing our submission. We look forward to engaging in some good discussion about the points you have raised.

---

### Official Review · Reviewer_K56J · 2022-10-25

**Confidence:** 3
**Correctness:** 4
**Technical Novelty And Significance:** 2
**Empirical Novelty And Significance:** 3
**Recommendation:** 5

**Clarity, Quality, Novelty And Reproducibility:**

The clarity is reasonable, and the results look very much reproducible.
The novelty is questionable, as discussed above.

**Strength And Weaknesses:**

*strengths*
- related work section is comprehensive (maybe also worth including [this](https://arxiv.org/abs/2206.03992))
- paper is overall well written, with thoughtful discussion
- a variety of experimental results

*weaknesses*
- the idea of unrolling autoregressively seems to be already well known at this point, in many recent works, and in some of the works already cited (e.g. esp [[1]](https://arxiv.org/abs/2207.04179) cf C.1, and also [[2]](https://arxiv.org/abs/2102.04426) [[3]](https://arxiv.org/abs/2110.02037) [[4]](https://arxiv.org/abs/2205.13554)). Therefore, I'm not sure that the ideas of this paper are novel enough to be a useful contribution to the community.
  - e.g. [[3]](https://arxiv.org/abs/2110.02037) Section 3.1 has a more in-depth discussion about parallelizing blocks, and presents a dynamic programming method from [[5]](https://arxiv.org/abs/2106.03802) for choosing how to parallelize. It's not in the stochastic process framework, but it does seem very similar.

**Summary Of The Paper:**

The paper presents Autoregressive Conditional Neural Processes, a way to use conditional neural processes in an autoregressive way, by feeding newly sampled points back into the CNP and re-evaluating. The paper vouches that using CNPs in this autoregressive way gives better predictive performance, since there are no independence assumptions. Furthermore, there are no changes that need to be done to the training or architecture, since this process only affects the inference routine.

The paper notes that this process does give up the property of consistency, since the predictions can depend on the different permutations chosen during inference. Also, sampling can be done in parallel using blocks of size N, to trade off conditional dependence and speed.


**Summary Of The Review:**

In summary, the paper proposes to augment conditional neural processes with an autoregressive inference procedure, by feeding samples back into the model. This method improves predictive performance without requiring retraining of the model.

Based on my personal opinion, I think this (or similar) techniques have already been used in many recent works, so I'm not sure that it is novel enough to be of interest. However, I'm not against acceptance if other reviewers find the idea novel, since otherwise the paper is well written with reasonable experimentation.


=============

I have read the other reviews and the author's response.

---

> ### Author Response · Authors · 2022-11-12
> **Response to Reviewer K56J (Part 4)**
>
> # Conclusion
>
> In conclusion, our finding is that, without modifications to the model or training procedure, AR can be applied to any CNP, and dramatically improves performance. We believe that this finding, backed up by both theoretical results and extensive experimental evidence, is of significant value to the community. We believe that this surprising result will impact the current practice of NPs, which does not consider running existing CNPs in AR.
>
> **If our discussion and amendments address your concerns to a satisfactory extent, we hope that you might consider increasing your score.**

---

> ### Author Response · Authors · 2022-11-12
> **Response to Reviewer K56J (Part 3)**
>
> # Novel contributions of our submission
>
> In addition to the above, our work presents various novel contributions which we believe are of value to the community:
>
> 1. We perform an extensive and varied range of experiments on synthetic and real data that act as a strong reference for the case of CNPs in AR mode. A notable result is the observation that AR CNPs can recover the ground truth process in various cases. We also perform experiments with environmental data which are of tangible value to the climate science community in their own right. We further comment on this below.
> 2. We identify the infinite-data limit of CNPs and GNPs and use this identification to formulate a theoretical result (Proposition 1) which states that AR CNPs outperform Gaussian Neural Processes (GNPs). We also formulate Proposition 2, which inspires an approach to separate aleatoric and epistemic uncertainty. Separating these uncertainties is crucial to all applications that require smooth samples, e.g. the climate experiments.
> 3. We provide a detailed and comprehensive exposition of related methods (Section 3, Figure 5). This positions AR CNPs as a yet unexplored method and reveals further gaps in the AR NDE toolbox and opportunities for further work.
>
>
> # Strength and range of experimental results
>
> While you acknowledge that the variety of experiments in our manuscript is a strength, we feel that the extensiveness of our experiments may not be well reflected in your assessment. In order to support our theoretical result (Proposition 1), which gives guarantees in the infinite-data limit, we have conducted a wide range of careful and rigorous experiments with both synthetic and real data. These experiments clearly demonstrate that AR CNPs provide a dramatic improvement over standard CNPs, GNPs, as well as LNPs, in practice and not just in theory.
>
> Specifically, we have conducted an extensive evaluation involving twelve experiments with synthetic Gaussian data and eight experiments with synthetic non-Gaussian data (see Tables 7 to 18 in Appendix F.3 for the full results), as well as a sim-to-real non-Gaussian task with predator-prey data. We have taken care to use fair and reproducible conditions in our experiments, to compare against an extensive range of CNP, LNP and GNP baselines with various architectures, including fully connected, attentive and convolutional networks. We also evaluated all models across a range of regimes, namely interpolation, extrapolation and out-of-distribution prediction. Beyond demonstrating the strength of AR CNPs over alternative models, this extensive evaluation demonstrates that our findings are robust across a range of settings, a contribution which we believe is of value to the NP and NDE literature.
>
> Lastly, we evaluated AR CNPs on several experiments with real data including physiological data from electroencephalograms as well as environmental temperature and cloud cover data, demonstrating the effectiveness of AR CNPs. In particular, the temperature downscaling and data fusion tasks which we tackle (Table 5) are of particular importance in the Climate Science literature. In these experiments, we demonstrate that the AR ConvCNP outperforms the ConvGNP, which previously yielded state-of-the-art results for downscaling. Further, our experiments with cloud cover over the Antarctic highlight further benefits of AR modelling. For example, AR works out-of-the-box with arbitrary likelihood functions, such as our beta-categorical mixture, while still modelling statistical dependencies in the output. Furthermore, as we point out, AR sampling also increases the effective receptive field of the model, enabling rich sample structure far away from observed data (Fig. 6b, Fig. 12) without any additional changes to the CNN. The combination of modelling dependencies, non-Gaussianity, and extrapolating far from observations while leveraging the data fusion capabilities of ConvCNPs is of great interest to the environmental science community, particularly for estimating the risk of compound events in remote regions.
>
> We hope that this discussion clarifies the significance of the experimental section.

---

> ### Author Response · Authors · 2022-11-12
> **Response to Reviewer K56J (Part 2)**
>
> # Distinguishing features of our work
>
> The central finding which we demonstrate in our work is that AR can be readily applied to _any existing trained CNP model_. Even though CNPs were not designed to be run in AR mode, we demonstrate that doing so improves performance dramatically and outperforms more sophisticated neural process variants, without the need for any changes to the model architecture or training procedure. In our opinion, this is a surprising, remarkable, and valuable finding.
>
> We believe that our work features various aspects that significantly differentiates it from existing methods in both the NP and the AR NDE literatures. This belief is reflected by the review of WrGz, commenting
>
> > The paper fills a blank spot in the literature [... since…] there exists no thourough [sic] investigation of the theoretical and empirical properties of AR-induced predictive distributions.
>
> while reviewer AdCH also comments that
>
> > The work seems to distinguish itself well from the prior works.
>
> While we have made an effort to describe these differences in our related work section (Section 3), we will amend our manuscript accordingly to better articulate them.
>
> ## AR CNPs versus Transformer Neural Processes
>
> Transformer Neural Processes (TNPs) [1] are a particular neural process architecture that propose two innovations. First, TNPs include an attention backbone with a causal masking procedure. Second, the output of this backbone is used to directly compute an autoregressive loss.
>
> This means that TNPs are a specific neural process designed to support an AR loss. In contrast, we propose to take any existing trained CNP, and, at test time, to run it in AR mode.
> - TNPs are a **specific** architecture. AR CNPs can be constructed from **any** existing CNP.
> - TNPs train with a bespoke **AR loss**. The CNP in an AR CNP is trained **as usual**, with a **non-AR loss**.
>
> In addition, the AR sampling procedure proposed in Appendix C.1 is specific for TNP-A. Our key finding is that this works remarkably well for any CNP.
> - **Only TNP-A** produces AR samples. **Any AR CNP** produces AR samples.
>
> **We therefore see a major difference in the positioning of TNPs and our submission.** TNPs propose a new architecture with a bespoke capability to do AR, whereas our proposal is that _any existing CNP_ can be run in AR mode. The value of our work therefore lies in (1) communicating this finding, (2) providing substantial experimental evidence to back it up, and (3) providing theoretical and methodological improvements that put AR CNPs on solid grounding. TNPs do no not address any of (1), (2), or (3). **We have made an amendment to the manuscript to better articulate these differences.**
>
> Finally, we stress the importance of building on existing CNPs. Practitioners have carefully designed CNPs for a plethora of applications, e.g. by incorporating symmetries into the architecture or by other advanced improvements. Our finding means that they can take these existing models and obtain a free performance win simply by running them in AR mode.  **Again, we will amend the manuscript to highlight this.**
>
> ## AR CNPs versus AR NDEs
>
> AR CNPs are built from _continuous_ stochastic processes and, as such, they naturally handle (a) continuous, irregularly sampled inputs and (b) missing data. These two features are crucial for real world applications such as healthcare or climate science. Handling these features with AR NDEs such as the diffusion- and energy-based models of [2], [3], [4] and [5] is not straightforward. Secondly, and arguably more importantly, another benefit of combining AR with the continuous stochastic process framework is the ability to leverage continuous symmetries, such as translation equivariance. In many problems, it is crucial to exploit these inductive biases to achieve reasonable performance.
>
> As a concrete but important example, the two real-world climate experiments (Section 4.4) involve very carefully designed neural processes in a complicated setup operating on various sources of data. _It would not have been possible to perform these experiments with AR NDEs because they cannot handle multiple, continuous data sources, as is essential in climate science_. On the other hand, turning these existing CNPs into AR CNPs was a straightforward change.
>
> **In our opinion, AR CNPs fill a very specific but definite gap left open by the current AR NE literature.**

---

> ### Author Response · Authors · 2022-11-12
> **Response to Reviewer K56J (Part 1)**
>
> We would like to thank you for your review and your comments on our manuscript. We are glad you found our manuscript well written and our discussion thoughtful and comprehensive.
>
> From your review, it appears that your main point of criticism is that autoregressive (AR) modelling has already been explored at length and that our work may not be novel enough to be of use to the community. While we agree that AR modelling is already a popular and promising method in the Neural Density Estimator (NDE) literature, running existing CNPs in AR mode has not been explored in previous work. In particular, the existing works which you brought up in your review are significantly different to the approach we present in several ways. In addition, we feel that your assessment might have overlooked central aspects of our work which present substantial novelty and value for the research community, backed up by an extensive range of experiments.

---

> ### Author Response · Authors · 2022-11-15
> **Reminder and updates to our paper**
>
> Dear Reviewer K56J,
>
> This is a brief message to remind you that we're nearing the end of the discussion period. We are eager to engage into a discussion, so we would very much appreciate it if you would reply to our response.
>
> We would also like to let you know that, in addition to the earlier amendments, we have inserted two new appendices: Appendix D and E. Appendix D illustrates the AR sampling process and the procedure the obtain smooth samples. Appendix E describes, analyses, and justifies our choice of ordering of the target points. We have also modified our discussion in the related work section (Section 3) to better reflect the differences of our work to Transformer Neural Processes.
>
> Thank you again for taking the time to review our submission.

---

### Official Review · Reviewer_AdCH · 2022-10-27

**Confidence:** 2
**Correctness:** 3
**Technical Novelty And Significance:** 3
**Empirical Novelty And Significance:** 3
**Recommendation:** 6

**Clarity, Quality, Novelty And Reproducibility:**

**Novelty.** The work seems to distinguish itself well from the prior works. It is motivated by the following limitations of the prior works: i) Standard CNPs assume a diagonal covariance among the target points — a limitation of CNP resolved by this work. ii) GNPs can model non-diagonal covariance structure but are restricted to Gaussian processes — a limitation of GNP but not of the proposed approach. iii) Latent-variable NPs (LNP) can also model non-diagonal covariance structure but are trained via an approximation of the likelihood objective — a limitation of LNP but not of the proposed approach.

**Reproducibility.** The work is reproducible since the CNP code is available and this work simply proposes sampling target points auto-regressively.

**Clarity and Quality.** The writing seems error-free. But given that the idea is simple and perhaps impactful, the writing and the presentation of math seem to make the idea a bit inaccessible to the first-time reader. After my first reading of the paper, I missed catching the key idea (CNP in AR mode) that I was only able to realize after some re-reads.

**Strength And Weaknesses:**

### Pros

1. Paper adopts the simplest class, CNP, to demonstrate their key idea — thus their main result seems to be free from the nuances of a complex architecture.
2. Paper shows that CNPs in AR mode are a more flexible class than GNP.
3. The paper clearly mentions the limitations such as the lack of consistency under marginalization and permutation. Also, care may be taken when feeding more points into the context set than shown during training. It is good for practitioners to know about these limitations and exercise caution when deploying it.
4. Paper also proposes how one can trade off speed and expressiveness in CNPs in AR mode by sampling target points block-wise.
5. Empirically, they show that CNP in AR mode outperforms the usual CNP that comes with diagonal covariance over target points. It also tends to do better/comparable to other baselines e.g. GNP or LNP. The fact it does so despite carrying the simplicity of CNPs seems like a good takeaway.

### Weaknesses/Questions

1. I do not fully understand the procedure for obtaining the smooth samples on the bottom part of page 4.
2. Paper does not analyze how the performance is affected if we auto-regressively sample more target points than shown during training. How quickly does it get worse?
3. Another important question not analyzed is whether TNP (a recent state-of-the-art) trained in diagonal-covariance-mode (i.e. TNP-D) would benefit similarly at test-time to what is shown in this paper for CNPs.
4. When trying to sample $N$ target points in the AR model, we sample the points $1, 2, \ldots, N$ sequentially. It would be good to show whether prediction error worsens going from point $1$ to point $N$ due to the accumulation of prediction errors in the context set.

**Summary Of The Paper:**

The paper shows that a standard CNP (trained with the diagonal covariance assumption) can be used, at test time, to sample target values capturing their non-diagonal covariance structure. This is done by sampling one target point at a time from the CNP and feeding that predicted target point back into the context set of the CNP. This is termed as running ‘CNP in auto-regressive (AR) mode’.

**Summary Of The Review:**

Several results (e.g., equation 7) rely on mathematical proofs which I think I (the reviewer) do not have the right expertise to review properly. Besides that, I think the paper is simple and clever. But due to my lack of expertise in judging the math and the possibility that there is some paper that already shows some part of what the paper shows (e.g. because it is so simple), I will exercise caution and rate it as 6 with confidence 2.

---

> ### Author Response · Authors · 2022-11-12
> **Response to Reviewer AdCH (Part 4)**
>
> ## 4. Accumulation of errors during AR sampling
>
> By accumulation of prediction errors in the context set, our interpretation is that you’re referring to the idea that, during the AR sampling procedure, we produce biased samples and then feed those biased samples back into the model to produce potentially even more biased samples. If we understand correctly, the risk is that this error/bias might accumulate and, after sufficiently many AR steps, potentially result in a catastrophic breakdown of the predictions. This is a very interesting question that requires a nuanced answer.
>
> To begin with, although the idea of error accumulating until a breakdown of the predictions sounds reasonable, Proposition 1 tells us that this cannot happen. In particular, Proposition 1 says that predictions by a perfectly trained AR CNP will never be worse than predictions by a perfectly trained GNP. Therefore, although bias might certainly accumulate, it will never influence the predictions in such a way that they become worse than the GNP. In particular, a catastrophic breakdown cannot happen.
>
> In fact, it may sound unintuitive, but in practice we have observed _the opposite phenomenon_. More specifically, as you sample more and more points in an AR fashion, under reasonable assumptions, the conditional distributions from which you sample become closer and closer to the ground truth. (To not digress from the main point too much here, we also refer you to “Ordering of Target Inputs” in our response to WrGZ, where we argue this in more detail.) Therefore, one could reasonably suspect that the samples might become more _accurate_ as you perform more and more AR steps. Indeed, with successive AR applications of the model, the predictions become more non-Gaussian and thus increase in modelling capacity, so this does not sound unreasonable.
>
> This behaviour is exactly what we observe in practice, and this is illustrated in Figure 3. On the left, the first AR step, the prediction is poor, because it is Gaussian. However, as we take more and more AR steps and go to the right, the prediction becomes increasingly non-Gaussian and converges to the ground truth - a surprising result! For a more detailed analysis of Figure 3, please see “(i) How would, e.g., Fig. 3 look (...) _left to right_?” in our response to WrGZ.
>
> Finally, there is one more aspect to consider. If, during the AR sampling process, the model is evaluated at more points than the number of context points seen at training time, then the predictions will break down, simply because the neural networks are then out-of-distribution (OOD) and will likely not generalise. For the AR ConvCNP, the model breaks down when the _density_ of the points is higher than that seen at training time; hence, the AR ConvCNP can be evaluated at _arbitrarily many points_, as long as the _density_ does not exceed that at training time.
>
> We find this discussion very interesting. We hope to further investigate it and perhaps even mathematically formalise it in future work.
>
> # Mathematical exposition in the paper
>
> You commented that you found our mathematical exposition in Section 2 somewhat difficult to follow. We appreciate that the notation in this section is denser than that typically encountered in NP papers, and might make the section less easy to read. However, we would like to highlight that adopting this slightly more detailed notation is necessary for our theoretical treatment of AR CNPs, and stating/proving Proposition 1. That being said, we agree with you that the clarity of this section can be improved, without impacting its rigour. **To this end, to make our central contribution easier to grasp on a first read, we have added a plain English description and simple mathematical example to Section 2, alongside other minor modifications. Thank you for raising this point and helping us to improve the paper’s clarity.**
>
> # Conclusion
>
> We are pleased that you found our paper simple, clever and novel. You raised a number of valuable points about things which we will add to our submission to improve it, including a clarification on our procedure for obtaining noiseless samples, an analysis of the impact of AR target points, and an analysis of the accumulation of errors in AR. We have made and will make more amendments to our manuscript to address these issues. In addition, we intend to include an evaluation of the TNP-D in AR mode in our camera-ready submission. Thank you again for your valuable insights that will help us make our paper even stronger.
>
> **If our discussion and changes to our manuscript address your concerns, we would invite you to consider improving our score.**

---

> > ### Comment · Reviewer_AdCH · 2022-12-07
> > **Great Response. Thank You!**
> >
> > I really appreciate this great response! It definitely addresses a lot of my concerns that I could follow without knowing the details of the proofs. The non-Gaussian sampling emerging naturally by simply changing the order of the target inputs is very interesting indeed.
> >
> > As I said, the main issue is not your paper but my own expertise in judging the proofs correctly which is the reason I will maintain my ratings for now.

---

> ### Author Response · Authors · 2022-11-12
> **Response to Reviewer AdCH (Part 3)**
>
> ## 2. Analysis on the number of AR target points
>
> To begin with, when sampling more points than shown during training, one can always use the trick to generate fine samples as just mentioned in “Satisfying the assumptions of Proposition 2” above. However, suppose that we really perform the AR sampling procedure with more target points than seen during training. There are now two cases to consider. For deep–set based models, such as the CNP, presenting the models with more points than seen during training presents the model with an out-of-distribution situation. What happens then comes entirely down to how well the neural networks generalise. Our experience is that the predictions quickly start to break down.
>
> For convolutional–deep–set based models, such as the ConvCNP, because of the translation equivariance property of convolutional deep sets, it is not the _number of points_ that matters but the _density of the points_. In particular, the AR ConvCNP can be run at arbitrarily many target points, many more than the number target points observed at training time, as long as the density of these points does not significantly exceed the density of the training data. Once the density exceeds the density of the training data, the model is presented with an out-of-distribution situation, and what happens then again comes down to how well neural networks generalise.
>
> **We will include a more detailed analysis on the effect of the number and order of AR points in a dedicated appendix, and modify the main text of the paper to highlight this.This appendix will also include the discussion in our response to WrGZ. Thank you for your valuable comment.**
>
> ## 3. Transformer Neural Processes
>
> You mentioned Transformer Neural Processes (TNPs), which are a particular CNP that use a transformer architecture with a causal mask to obtain good performance across a range of tasks. We have already made reference to TNPs in our related work section (Section 2), but we agree with you that it would be interesting to compare how TNP-D performs when run in AR mode at test time. By Proposition 1, we expect that running TNP-D in AR mode will generally improve its performance. **While the time constraints of the discussion period do not allow us to implement and carefully test the TNP-D, we would like to include this model in our experiments in our camera-ready version. Thank you for suggesting this addition.**

---

> ### Author Response · Authors · 2022-11-12
> **Response to Reviewer AdCH (Part 2)**
>
> ## 1. Procedure for obtaining smooth samples
>
> We would like to clarify the procedure we use to obtain smooth samples at the bottom of page 4. This procedure consists of two steps. In the first step, we produce a noisy sample with the AR sampling procedure. In the second, we remove the noise to produce the noiseless sample. **Step 1:** Suppose we are given a context set $(\mathbf{x}^{(c)}, \mathbf{y}^{(c)})$, and wish to draw function samples from the predictive of an AR CNP at some target points $\mathbf{x}\_{1:N}$. We follow Procedure 2.1 to sequentially sample corresponding noisy output values, $\mathbf{y}\_{1:N}$, one by one. Specifically, we first sample $y_1$ from the predictive distribution of the CNP with context set $(\mathbf{x}^{(c)}, \mathbf{y}^{(c)})$ and target input $x_1$, namely
>
> $$ y_1 \sim P_{x_1} \pi_\theta(\mathbf{x}^{(c)}, \mathbf{y}^{(c)}). $$
>
> After drawing the target output $y_1$, we append $(x_1, y_1)$ to the context set, to obtain a new context set $(\mathbf{x}^{(c)} \oplus x_1, \mathbf{y}^{(c)} \oplus y_1)$, and then repeat this procedure for sampling $y_2$, namely
>
> $$ y_2 \sim P_{x_2} \pi_\theta(\mathbf{x}^{(c)} \oplus x_1, \mathbf{y}^{(c)} \oplus y_1). $$
>
> We proceed in this way to obtain noisy samples $\mathbf{y}\_{1:N}$. The problem is that these samples also include sampled observation noise. This comes from the fact that the CNP also models the i.i.d. observation noise present in the data (aleatoric noise). **Step 2:** To eliminate this noise, following Equation 8, we pass the final context set, with all AR samples appended, once more through the model and take the mean prediction of the model as the noiseless sample:
>
> $$  \mathcal{N}(\mu\_{1:N}, \mathbf{D}) = P\_{\mathbf{x}\_{1:N}} \pi\_\theta(\mathbf{x}^{(c)} \oplus \mathbf{x}\_{1:N}, \mathbf{y}^{(c)} \oplus \mathbf{y}\_{1:N})$$
>
> where $\mu\_{1:N}$ denotes the mean of the output and where $\mathbf{D}$ denotes a diagonal covariance matrix. This mean, $\mu\_{1:N}$, now constitutes the sample with aleatoric noise removed. We emphasise that this procedure is justified by Proposition 2.
>
> **Satisfying the assumptions of Proposition 2:** To use Proposition 2, we should choose $\mathbf{x}\_{1:N}$ to be as dense as possible, in order to remove all epistemic uncertainty after conditioning on the noisy samples $\mathbf{y}\_{1:N}$. However, this grid cannot be chosen too densely, because it might subsequently present the model with a number of inputs greater than seen during training, i.e. an out-of-distribution (OOD) set of inputs (a point that you highlighted in your review, which we will expand upon in the next section). In practice, as long as the number of target points does not exceed that during training, we found that the models are fairly robust to this $N$ and give sensible samples (as shown in Figure 4). _However_, this does _not_ mean that we cannot produce fine AR samples. In particular, in step 2, we are free to compute
>
> $$  \mathcal{N}(\mathbf{\mu’}\_{1:N’}, \mathbf{D}) = P\_{\mathbf{x’}\_{1:N’}} \pi\_\theta(\mathbf{x}^{(c)} \oplus \mathbf{x}\_{1:N}, \mathbf{y}^{(c)} \oplus \mathbf{y}\_{1:N})$$
>
> where now $\mathbf{x’}\_{1:N’}$ is a very fine grid. In other words, to produce fine samples, we first produce as many AR points as we can without resulting in an OOD context set, and then query the resulting mean function at arbitrarily many inputs. We will further expand on the issue choosing $N$ further in the subsequent section, in response to question (2) which you raised.
>
> We hope this clarifies the procedure for obtaining noiseless samples from the model. **We have made amendments to our text to clarify this procedure and we will introduce another figure (possibly in the appendix) which depicts the sampling procedure visually. Thank you for raising this point.**

---

> ### Author Response · Authors · 2022-11-12
> **Response to Reviewer AdCH (Part 1)**
>
> We thank you for your valuable time taken to provide detailed comments on our paper. We are glad you see the simplicity and improved performance of our approach as strengths of our work. We are pleased that you recognise how our work distinguishes itself well from prior works and that you appreciate the insights from detailed discussion on our method such as consistency and block-wise sampling.
>
> In your review, you have raised some questions, which we would like to address here. You pointed out that the key idea (CNP in AR mode) was possible to miss on a first-time read. We have made changes to the manuscript to make the reading in Section 2 easier to follow. Please see the revised PDFs. Specifically, the revised PDF includes a simple example of AR sampling with a CNP that complements the more technical presentation in Section 2.
>
> Our motivation for the more technical approach of our writing is to present NPs in a mathematically rigorous way. This more rigorous approach, in turn, enables a theoretical analysis, eventually leading to Proposition 1. The theoretical analysis, specifically Proposition 1, is one of the central contributions of our work, providing insights for a better understanding of AR CNPs.
>
> **Responses to your questions:** Below, we will respond to your questions in the order you posed them.

---

> ### Author Response · Authors · 2022-11-15
> **Reminder and updates to our paper**
>
> Dear Reviewer AdCH,
>
> This is a brief message to remind you that we're nearing the end of the discussion period. We are eager to engage into a discussion, so we would very much appreciate it if you would reply to our response.
>
> We would also like to let you know that, in addition to the earlier amendments, we have inserted two new appendices: Appendix D and E. Appendix D illustrates the AR sampling process and the procedure the obtain smooth samples. Appendix E describes, analyses, and justifies our choice of ordering of the target points. We have also amended our mathematical exposition in Section 2, to address your concerns on the readability of this section.
>
> Thank you again for taking the time to review our submission.

---

> ### Author Response · Authors · 2022-12-07
> **Last week of discussion period**
>
> Dear AdCH,
>
> This week is the last week of the discussion period. We have put a lot effort in our response below and additional revisions to incorporate your feedback, so we would greatly appreciate it if you would take the time to reply before the discussion period ends.
>
> We would like to point out that K56J and WrGZ have both increased their scores. If you believe that we have addressed your concerns to a satisfactory extent, we hope that you might help our chances of acceptance by also increasing your score.

---

### Author Response · Authors · 2022-11-12
**Reply to all Reviewers and Area Chairs**

We would like to thank all reviewers for the time spent assessing our paper and for providing helpful comments.

The main contribution of our submission is the finding that existing neural process models, _without modification to the model or training procedure_, can be deployed autoregressively (AR) to dramatically improve performance. This observation is significant and novel, because, even though AR models have been investigated at length in previous work, the immediate and substantial benefits of running existing CNPs in AR mode have not.

Importantly, we believe that NP practitioners can take existing CNPs and run them in AR mode to obtain immediate improvements in performance! The purpose of our submission is to communicate this message to the wider community and to back it up with extensive reproducible experimental evidence across an array of modelling scenarios.

The work which is perhaps most comparable to our submission is TNPs (Nguyen & Glover, 2022). However, as we explain in our response to WrGZ, both the positioning and the contributions of TNPs are completely different to our work.

In addition to the observation that AR CNPs present dramatic performance gains, our work features an analysis that enables a new theoretical understanding of CNPs, GNPs and AR CNPs and proposes several methodological improvements that are key to making AR CNPs work well.

For the convenience of the reviewers and the AC, we here summarise the main positive and negative points raised by the reviewers:

1. AdCH “think[s] the paper is simple and clever,” and “distinguishes well from prior works”. AdCH has four questions. We answer these questions in detail in our reply to AdCH.
2. K56J thinks the paper is “well written”, “very much reproducible”, and includes a “variety of experimental results”. However, the idea of unrolling autoregressively is not novel. Although AR models have already been explored at length, running existing CNPs in AR mode has, at best, only been cursorily mentioned in prior work. To the best of our knowledge, there exists no prior work that investigates AR CNPs in a range of careful, reproducible experimental settings.
3. WrGZ thinks that the paper is “very well written” and that the “experimental section is very well fleshed out”. WrGZ mentions only two minor open points.

---

### Decision · Program_Chairs · 2023-01-20

**Decision:**

Accept: poster

**Justification For Why Not Higher Score:**

(1) Their autoregressive approach sacrifices key theoretical guarantees of standard CNPs. (2) Lack of comparison with TNPs which is the most relevant autoregressive CNP work here.

**Justification For Why Not Lower Score:**

Neat and clever trick that is worth future study. Good experiments even though baselines are limited.

**Metareview: Summary, Strengths And Weaknesses:**

The paper proposes an algorithmic technique to convert CNPs with factorized Gaussian likelihoods with diagonal covariance into autoregressive likelihoods. The technique is conceptually simple: augment the model predictions into the context set of a CNP before making the next prediction. There are some subtle important details here, such as removing the noise in the prediction before augmentation and picking a right order of target points for prediction -- there are no "clean" answers here but the authors do a good job at discussing the potential issues that might arise. There are also theoretical limitations in this scheme, such as the lack of consistency under marginalization and permutation, which have been acknowledged by the authors. Nevertheless, I think there is empirical merit in the scheme, as is also evident by the experiments done by the authors comparing their autoregressive CNPs with the default CNPs.

One outstanding concern with the current work is the performance of their approach with TNPs (Nguyen&Grover, ICML 2022). TNPs also study both diagonal and autoregressive factorizations of their approach and are the current SOTA in the NP space. While the current work is distinct in converting a diagonal CNP into an autoregressive one post-training, the reviewers are curious if the performance benefits surpass TNPs. This is a valid question from both a practical and conceptual perspective and one that I encourage the authors to study empirically in their final version.



**Note From Pc:**

if the above contains the word "oral" or "spotlight" please see: "oral" presentation means -> notable-top-5% and "spotlight" means -> notable-top-25%. As stated in our emails, we are disassociating presentation type from AC recommendations